# Test-Time Reinforcement Learning for Flow Matching

Jili Chen [1 2]  Changqin Huang [3 1]  Qionghao Huang [1]  Yaxin Tu [3]  Zhonglong Zheng [2]  Xiaodi Huang [4]

## Abstract

Flow-matching has emerged as a leading framework for high-fidelity text-to-image generation. However, its alignment with human preferences through RL is often hindered by substantial computational overhead. In this paper, we introduce Flow-TTRL, the first test-time reinforcement learning framework that achieves alignment on the fly. Our approach reinterprets intermediate latent representations as an implicit policy and utilizes SDE-based rollouts to explore high-reward trajectories within the learned vector field. Specifically, we propose a two-stage optimization strategy: Proximal Reward Difference Prediction (PRDP) ensures structural stability in high-noise regimes through pairwise reward regression, while Group Relative Policy Optimization (GRPO) refines fine-grained aesthetic details by maximizing relative advantages within sampled candidate groups. Experimental results show that Flow-TTRL significantly boosts aesthetic quality, text-image alignment, and human preference across diverse backbones. On the GenEval benchmark, Flow-TTRL elevates the accuracy of SD 3.5-Medium from 63% to 87% and Flux.1 Dev from 66% to 83%. Furthermore, our framework achieves an average gain of 15% to 20% across T2I-CompBench metrics, delivering performance comparable to state-of-the-art RL-based fine-tuning methods without the need for additional fine-tuning. Our code is available at https://github.com/TheShy-Dream/Flow-TTRL.

[1]Zhejiang Key Laboratory of Intelligent Education Technology and Application, Zhejiang Normal University, Jinhua, 321004, China [2]School of Computer Science and Technology, Zhejiang Normal University, Jinhua, 321004, China [3]College of Education, Zhejiang University, Hangzhou, 310063, China [4]School of Computing, Mathematics and Engineering, Charles Sturt University, Albury, NSW 2640, Australia. Correspondence to: Changqin Huang <cqhuang@zju.edu.cn>, Qionghao Huang <qhhuang@m.scnu.edu.cn>.

*Proceedings of the $43^{rd}$ International Conference on Machine Learning*, Seoul, South Korea. PMLR 306, 2026. Copyright 2026 by the author(s).

## 1. Introduction

Flow-matching (FM) (Lipman et al., 2022; Esser et al., 2024; Lei et al., 2023; Ma et al., 2025; Labs et al., 2025) has recently emerged as a dominant paradigm in high-fidelity visual generation. To align these models with complex human preferences, integrating FM with Group Relative Policy Optimization (GRPO) (Shao et al., 2024) has emerged as a potent strategy. Such improvements rely on fine-tuning model parameters on samples generated through SDE sampling to align the model with target reward signals (Xue et al., 2025; Liu et al., 2025a; Wang et al., 2025a; He et al., 2025; Wang et al., 2025b).

Despite these advancements, this heavy reliance on fine-tuning introduces significant practical limitations (Figure 1) (Fan et al., 2023; Black et al., 2023; Yang et al., 2024). The substantial training overhead and data dependency often restrict such methods to industrial-scale compute environments. Moreover, the resulting policies are inherently rigid, as they are tightly coupled to the reward models and specific configs used during training. The inflexibility makes it challenging to adapt a finalized model to personalized user preferences or diverse downstream tasks (Huang et al., 2025b; Wei et al., 2024; Dunlop et al., 2025). These bottlenecks are exacerbated in large-scale models, where the computational cost of fine-tuning becomes prohibitive. Such limitations underscore the urgent need for a more efficient RL paradigm that enables optimization while bypassing the need for computationally expensive parameter updates.

To address these challenges, a fundamental question naturally emerges: **How to design a reinforcement learning mechanism that dynamically navigates the latent manifold to locate high-reward regions during inference?** Motivated by this insight, we propose **Flow-TTRL**, a novel Test-Time Reinforcement Learning framework designed for flow-matching models. Specifically, Flow-TTRL reinterprets intermediate latent representations as an implicit policy, as illustrated in the right part of Figure 1. To explore the high-reward action space, we employ SDE-based rollouts at each denoising step to generate a group of candidate trajectories. The latent state is then optimized by minimizing reinforcement learning objectives calculated from the relative rewards of these candidates. This procedure performs action rectification through a coarse-to-fine strategy,

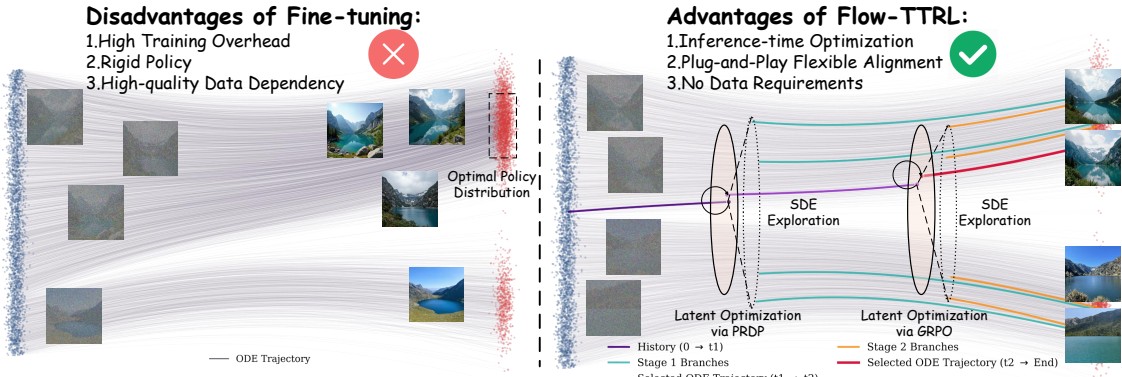

*Figure 1.* **Comparison between conventional RL fine-tuning and our proposed Flow-TTRL.** Left: Traditional fine-tuning modifies network weights to shift the generative distribution. This paradigm is often hindered by significant disadvantages, including high training overhead, rigid policies, and a heavy dependency on massive, high-quality training data. Right: Flow-TTRL keeps the backbone frozen and performs test-time latent optimization. Flow-TTRL achieves efficient, plug-and-play alignment by dynamically searching for high-reward regions during inference. By operating within the pre-existing latent manifold, it successfully circumvents the fundamental bottlenecks of fine-tuning.

enabling the latent to dynamically switch between trajectories within the learned vector field toward high-reward regions. Specifically, in the high-noise early stages, Proximal Reward Difference Prediction (PRDP) is used to achieve structural alignment by regressing on pairwise reward differences, mitigating the fragility of early denoising. As the structure stabilizes, we transition to Group Relative Policy Optimization (GRPO), leveraging group-wise advantages to excavate and polish fine-grained aesthetic details. By maintaining a parallel reference trajectory as a manifold anchor with an integrated KL-divergence constraint, Flow-TTRL efficiently searches for high-reward latent regions to achieve plug-and-play alignment within a single inference pass without modifying model parameters. Furthermore, Flow-TTRL requires no additional training data and enables flexible control over alignment intensity by adjusting reward weight configurations.

We evaluate the effectiveness of Flow-TTRL through extensive experiments on two widely adopted flow-matching backbones, including SD 3.5-Medium (Esser et al., 2024) and Flux.1 Dev (Labs et al., 2025). On the GenEval benchmark (Ghosh et al., 2023), Flow-TTRL significantly elevates the accuracy of SD3.5-M from 63% to 87% and Flux.1 Dev from 66% to 83%. On the T2I-CompBench (Huang et al., 2023; 2025a) benchmark, our framework achieves a 15% to 20% average performance gain over native backbones, outperforming state-of-the-art RL-based fine-tuning methods across the vast majority of tasks. These results, bolstered by comprehensive quantitative and qualitative analyses on five datasets, validate Flow-TTRL in achieving high-fidelity alignment while remaining fully accessible on consumer-grade hardware. Our contributions can be summarized as follows:

- **The First Test-Time Reinforcement Learning Framework**: We introduce the first test-time reinforcement learning framework for flow matching, which reinterprets latent trajectories as an implicit policy to enable flexible alignment without the need for traditional fine-tuning or curated training data.

- **Coarse-to-fine Trajectory Optimization:** We design a coarse-to-fine strategy (PRDP followed by GRPO) that balances structural stability with precise aesthetic refinement during the trajectory optimization process.

- **Competitive Training-Free Performance:** We demonstrate that Flow-TTRL achieves highly competitive results on GenEval and T2I-CompBench, yielding performance comparable to established RL-based fine-tuning methods and proprietary models. Furthermore, it consistently enhances human preference alignment and image fidelity across all five evaluated datasets.

## 2. Related Work

**Diffusion and Flow Matching.** Diffusion models (Sohl-Dickstein et al., 2015; Ho et al., 2020; Song et al., 2020a;b) simulate a thermodynamic diffusion process using SDEs to progressively perturb data into Gaussian noise, and then train a neural network for denoising reconstruction. By comparison, flow matching (Lipman et al., 2022; Liu et al., 2022; Albergo & Vanden-Eijnden, 2023) constructs optimal transport paths by directly regressing the conditional vector field between source and target distributions, enabling more efficient and stable sample generation through straighter ODE trajectories. Recent research (Albergo et al., 2023; Ma et al., 2024) demonstrates that diffusion and flow models can be unified under an SDE/ODE framework. They

are already dominant methods for image (Rombach et al., 2022; Podell et al., 2023; Peebles & Xie, 2023; Esser et al., 2024) and video generation (Ho et al., 2022; Blattmann et al., 2023; Lei et al., 2023; Polyak et al., 2024). Building on previous theoretical and applied foundations, our work further extends the practical utility of flow matching models through test-time reinforcement learning.

**Reinforcement Learning in Diffusion.** Reinforcement learning, such as RLHF or RLAIF (Christiano et al., 2017; Lee et al., 2023a), guides diffusion models to align their outputs with human preferences or desired objectives, improving quality, controllability, and safety. The main approaches can be summarized as follows: (1) some methods directly fine-tune diffusion models using differentiable rewards (Prabhudesai et al., 2023; Xu et al., 2023; Clark et al., 2023; Wu et al., 2024; Guo et al., 2025). (2) RWR-based methods reweight sampled trajectories to guide the models to increase the likelihood of high-reward trajectories (Peters & Schaal, 2007; Lee et al., 2023b; Dong et al., 2023; Fan et al., 2025). (3) PPO-based methods treat the denoising process as a Markovian dynamic process and optimize the policy to maximize expected rewards (Schulman et al., 2017; Fan et al., 2023; Black et al., 2023; Zhang et al., 2024a; Li et al., 2024; Hu et al., 2025a; Chen & Kuo, 2025). (4) DPO-based methods bypass reward models by learning from paired comparisons of preferred and less-preferred samples (Rafailov et al., 2023; Wallace et al., 2024; Yang et al., 2024; Zhu et al., 2025; Shen et al., 2025; Hu et al., 2025b; Liang et al., 2025). (5) GRPO-based methods perform value-free preference optimization for memory-efficient alignment (Xue et al., 2025; Liu et al., 2025a; Wang et al., 2025a; He et al., 2025; Wang et al., 2025b; Luo et al., 2025). Traditional RL methods can better align models with human preferences, but they typically require additional modules such as LoRA (Hu et al., 2022), carefully curated datasets, complex training procedures, and significant computational resources to achieve satisfactory performance. Moreover, they struggle to provide personalized adaptation (Dang et al., 2025). To address these challenges, we optimize sampling trajectories during inference, thereby greatly reducing computational overhead.

**Test-Time Alignment in Diffusion.** Test-time alignment allows the model to incur some additional computation during inference, achieving better preference alignment without requiring extra training. The optimization targets include the sampling process (Bai et al., 2024; 2025), initial or intermediate noise (Eyring et al., 2024; Guo et al., 2024; Tang et al., 2024; Zhou et al., 2025), and latent representations (Chefer et al., 2023; Rassin et al., 2023; Zhang et al., 2024b; Simon et al., 2025) during the reverse diffusion process, with supervision signals provided by external reward models or internal attention scores. While traditional test-time optimization methods typically view diffusion as a pure denoising ODE

and often require differentiable rewards, our Flow-TTRL provides a training-free alternative by reformulating the unified ODE+SDE process into a Markov Decision Process. This perspective not only enables compatibility with general, non-differentiable rewards but also ensures KL-regularized alignment, making the process more robust against reward hacking compared to standard test-time optimization approaches.

# 3. Preliminaries

**Rectified Flow.** Rectified Flow (Liu et al., 2022) learns a transport map between a target data distribution $\pi_0$ and a source distribution $\pi_1$ (e.g., standard Gaussian) by solving an ODE. It constructs a linear interpolation path $\mathbf{x}_t = (1 - t)\mathbf{x}_0 + t\mathbf{x}_1$ for $t \in [0, 1]$, which corresponds to a straight line connecting a pair $(\mathbf{x}_0, \mathbf{x}_1)$. The neural velocity field $v_\theta$ is optimized to match this direction by minimizing the following objective:

$$\mathcal{L}(\theta) = \mathbb{E}_{\mathbf{x}_0 \sim \pi_0, \mathbf{x}_1 \sim \pi_1, t \sim \mathcal{U}[0,1]}\Big[\big\|v_\theta(\mathbf{x}_t, t) - (\mathbf{x}_1 - \mathbf{x}_0)\big\|^2\Big]. \tag{1}$$

During inference, samples are generated by numerically integrating the learned field $v_\theta(\mathbf{x}_t, t)$ starting from $\mathbf{x}_1 \sim \pi_1$ to $t = 0$.

**Denoising as an MDP.** The iterative denoising process in the diffusion model can be formulated as a Markov Decision Process (MDP) tuple $(\mathcal{S}, \mathcal{A}, \rho_0, \mathcal{P}, \mathcal{R})$ (Black et al., 2023). The state at step $t$ is defined as $s_t \triangleq (\mathbf{c}, t, \mathbf{x}_t)$, comprising the context, timestep, and noisy latent. The policy $\pi(\mathbf{a}_t|s_t) \triangleq p_\theta(\mathbf{x}_{t-1}|\mathbf{x}_t, \mathbf{c})$ predicts the action $\mathbf{a}_t \triangleq \mathbf{x}_{t-1}$ which corresponds to the denoised sample for the next step. The transition kernel is defined using Dirac delta distributions: $P(s_{t+1}|s_t, \mathbf{a}_t) \triangleq (\delta_c, \delta_{t-1}, \delta_{\mathbf{x}_{t-1}})$, where $\delta_y$ denotes a point mass centered at $y$. The process initializes from $\rho_0(s_0) \triangleq (p(\mathbf{c}), \delta_T, \mathcal{N}(\mathbf{0}, \mathbf{I}))$ and operates with a reward signal, assigned only at the terminal step: $R(s_t, \mathbf{a}_t) \triangleq r(\mathbf{x}_0, \mathbf{c})$.

# 4. Flow-TTRL

In this section, we propose **Flow-TTRL**, a test-time reinforcement learning framework designed to enhance human-preference alignment during inference. We begin by reformulating the trajectory optimization into the optimization of latent representations (4.1). Then, we introduce a coarse-to-fine optimization strategy consisting of a warm-up stage based on Proximal Reward Difference Prediction (4.2), followed by a trajectory refinement stage using GRPO (4.3). Finally, we detail our mechanism for noise weighting and latent optimization (4.4). The overall procedure of Flow-TTRL is depicted in Figure 2.

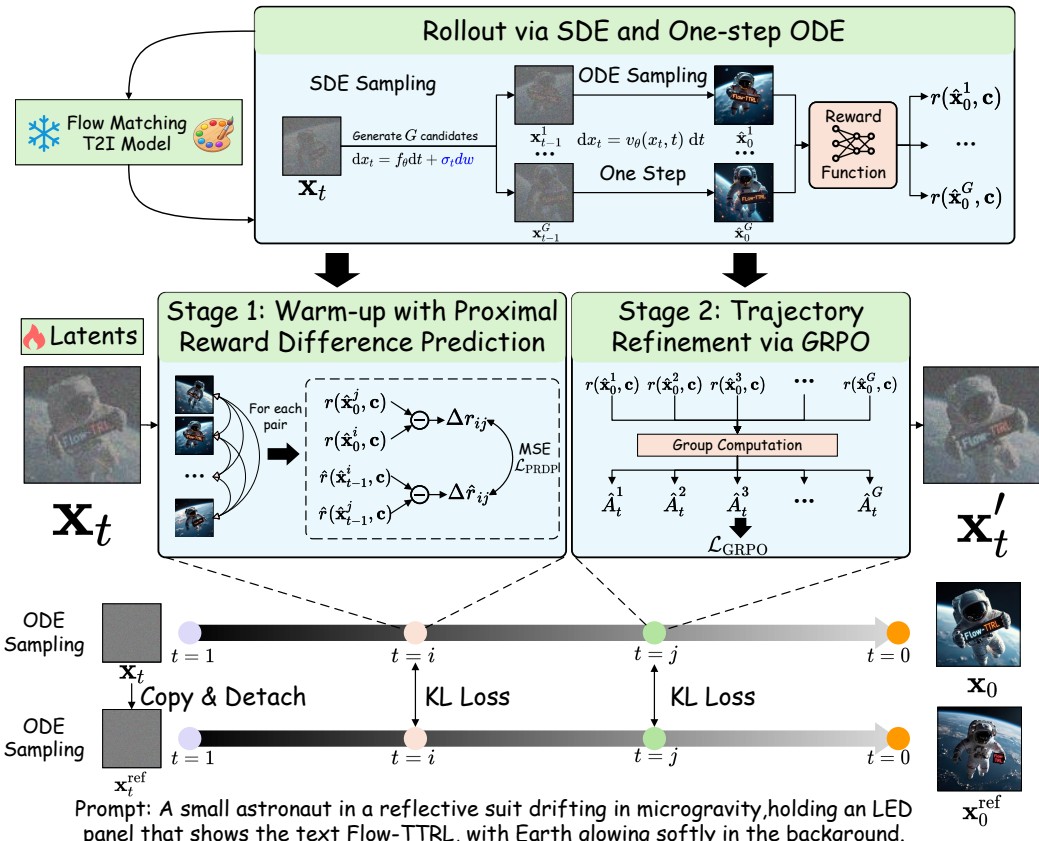

*Figure 2.* **Overview of the Flow-TTRL framework.** To achieve TTRL, we design a coarse-to-fine strategy that transitions from PRDP warm-up to GRPO refinement. Throughout this process, a parallel reference trajectory is maintained as a global anchor to regularize the latent updates. Notably, the entire procedure optimizes only the intermediate latents, requiring no additional parameters or model fine-tuning.

## 4.1. Latent Representations as an Implicit Policy

**Action Rectification through Latent Optimization.** Unlike fixed environmental states, the latent state $\mathbf{x}_t$ in flow matching models exhibits a dual role within the sampling process. On one hand, $\mathbf{x}_t$ represents the instantiated action $\mathbf{a}_t$ at the current step; on the other hand, it serves as the essential state component $s_t$ that conditions the subsequent vector field estimation. This dual role implies that $\mathbf{x}_t$ is not merely a passive observation, but a mutable variable within a fully differentiable transition chain. As the policy network $v_\theta$ governs the evolution of the trajectory, the denoising "action" at step $t$ is uniquely dictated by the latent state $\mathbf{x}_t$. Consequently, modifying $\mathbf{x}_t$ inherently alters the sampled vector field, thereby steering the generation trajectory. Formally, let the transition be modeled by an ODE solver $\Phi$. The latent state at the next step is a function of the current state and the network output:

$$\mathbf{x}_{t-\Delta t} = \Phi(\mathbf{x}_t, v_\theta(\mathbf{x}_t, t)). \qquad (2)$$

We propose that the latent $\mathbf{x}_t$ functions as an Implicit Policy. By introducing a learnable perturbation $\delta_t$ to $\mathbf{x}_t$, we can modulate the effective vector field output, which can be estimated through a first-order Taylor expansion via the gradient of the field with respect to the input $\mathbf{x}_t$:

$$v_\theta(\mathbf{x}_t + \delta_t, t) \approx v_\theta(\mathbf{x}_t, t) + \underbrace{\nabla_{\mathbf{x}_t} v_\theta(\mathbf{x}_t, t) \cdot \delta_t}_{\text{Action Rectification}}. \qquad (3)$$

Eq. 3 demonstrates that optimizing the latent space is mathematically equivalent to rectifying the policy's action. This allows us to maximize the reward $R$ by backpropagating gradients directly to the latent representation:

$$\mathbf{x}_t^* = \mathbf{x}_t + \eta \nabla_{\mathbf{x}_t} R(\Phi_\theta(\mathbf{x}_{T:0}, \mathbf{c})), \qquad (4)$$

where $\eta$ is the step size. This formulation bypasses the need to update the massive parameters of the backbone network $\theta$, enabling trajectory optimization during inference.

**Reference Latent.** In standard RL, a reference policy $\pi_{\text{ref}}$ is essential to constrain the optimization landscape, typically via KL-divergence regularization. In Flow-TTRL, since the latent representation functions as the policy, we introduce the reference latent $\mathbf{x}_t^{\text{ref}}$, to serve as the regularization anchor for the online latent $\mathbf{x}_t$. We maintain a parallel trajectory

generated purely by starting from the same noise $\mathbf{x}_1$. Specifically, $\mathbf{x}_t^{\text{ref}}$ evolves according to the vanilla ODE trajectory defined by the solver $\Phi$: $\mathbf{x}_{t-\Delta t}^{\text{ref}} = \Phi(\mathbf{x}_t^{\text{ref}}, v_\theta(\mathbf{x}_t^{\text{ref}}, t))$, serving as a global anchor to the original data manifold. Consequently, the optimization paradigm transforms from the traditional global parameter update:

$$\max_{\pi_\theta} \; \mathbb{E}_{\mathbf{x}_0, \mathbf{c}} \Big[ r(\mathbf{x}_0, \mathbf{c}) - \beta \, \text{KL}\big(\pi_\theta(\mathbf{x}_0|\mathbf{c}) \, \| \, \pi_{\text{ref}}(\mathbf{x}_0|\mathbf{c})\big) \Big],$$
(5)

to our proposed latent update:

$$\max_{\mathbf{x}_t} \; \mathbb{E}_{\mathbf{x}_{t-1} \sim \pi(\mathbf{x}_{t-1}|\mathbf{x}_t, \mathbf{c})} \Big[ r\big(\Phi_\theta(\mathbf{x}_{t-1:0}, \mathbf{c})\big) \Big]$$
$$- \beta \, \text{KL}\Big( \pi(\cdot|\mathbf{x}_t, \mathbf{c}) \, \| \, \pi(\cdot|\mathbf{x}_t^{\text{ref}}, \mathbf{c}) \Big). \quad (6)$$

**Rollout and Optimization with the RL Objective.** Optimizing this RL objective requires exploring the action space. However, the standard probability flow ODE is deterministic. Inspired by Flow-GRPO (Liu et al., 2025a), we adopt the SDE formulation:

$$\boldsymbol{x}_{t+\Delta t} = \boldsymbol{x}_t + [\boldsymbol{v}_\theta(\boldsymbol{x}_t, t) + \frac{\sigma_t^2}{2t}(\boldsymbol{x}_t + (1-t)\boldsymbol{v}_\theta(\boldsymbol{x}_t, t))]\Delta t$$
$$+ \sigma_t \sqrt{\Delta t} \epsilon,$$
(7)

where $\epsilon \sim \mathcal{N}(0, I)$ injects stochasticity, and $\sigma_t = \alpha\sqrt{\frac{t}{1-t}}$ controls the noise intensity, $\alpha$ is a hyper-parameter. Thus, we can generate a batch of candidate latents $\{\mathbf{x}_{t-1}^i\}_{i=1}^G$ from the current state $\mathbf{x}_t$, $G$ is the number of candidates. For each candidate $\mathbf{x}_{t-1}^i$, we approximate the terminal state $\hat{\mathbf{x}}_0^i$ via a single-step projection $\hat{\mathbf{x}}_0^i = \mathbf{x}_{t-1}^i - (t-1) \cdot v_\theta(\mathbf{x}_{t-1}^i, t-1)$. This approximation is analogous to Tweedie's formula (Robbins, 1992; Efron, 2011), a widely adopted practice in test-time optimization and reinforcement learning (Yu et al., 2023; Bansal et al., 2024; Li et al., 2025; Clark et al., 2023) to simplify the computation graph and obtain dense supervision signals (The error analysis is provided in Appendix A.2). The reward is then obtained by decoding the projected latent $r^i\big(\Phi_\theta(\mathbf{x}_{t-1:0}, \mathbf{c})\big) \approx R\big(\mathcal{D}(\hat{\mathbf{x}}_0^i), \mathbf{c}\big)$, where $\mathcal{D}$ denotes the VAE decoder. Consequently, the variation in rewards can be explicitly attributed to the stochasticity introduced by the SDE, providing a dense supervision signal for optimizing the action space for each step.

## 4.2. Stage 1: Warm-up with Proximal Reward Difference Prediction

The early phase of the reverse diffusion process is decisive for establishing global semantics, such as object quantity, spatial layout, and structural composition (Kulikov et al., 2025). While Eq. 6 theoretically unifies RL with latent optimization, directly applying it during the early denoising stages is perilous. In this high-noise regime, the latent $\mathbf{x}_t$ becomes highly unstable due to its inherent fragility, the

high variance of reinforcement learning updates, and the strong stochasticity in early reverse-diffusion steps. These factors will cause trajectory collapse, where the generation deteriorates into noise.

We mitigate this instability by formulating the optimization as a regression problem on Proximal Reward Difference Prediction (Deng et al., 2024). Instead of maximizing absolute rewards, we align the difference in transition probabilities with the difference in rewards between sample pairs. Inspired by DPO (Rafailov et al., 2023), we can derive an optimal solution to:

$$\pi(\mathbf{x}_{t-1}|\mathbf{x}_t^*, \mathbf{c}) = \frac{1}{\mathcal{Z}(c)} \pi(\mathbf{x}_{t-1}|\mathbf{x}_t^{\text{ref}}, \mathbf{c}) \exp\left(\frac{r(\hat{\mathbf{x}}_0, \mathbf{c})}{\beta}\right),$$
(8)

where $\mathcal{Z}(\mathbf{c}) = \int \pi(\mathbf{x}_{t-1}|\mathbf{x}_t^{\text{ref}}, \mathbf{c}) \exp\left(\frac{r(\hat{\mathbf{x}}_0, \mathbf{c})}{\beta}\right) d\mathbf{x}_{t-1}$ is the partition function. The detailed proof is provided in Appendix A.1. Thus, the optimal latent $\mathbf{x}_t^*$ must satisfy:

$$\log \frac{\pi(\mathbf{x}_{t-1}|\mathbf{x}_t^*, \mathbf{c})}{\pi(\mathbf{x}_{t-1}|\mathbf{x}_t^{\text{ref}}, \mathbf{c})} = \frac{1}{\beta} r(\hat{\mathbf{x}}_0, \mathbf{c}) - \log \mathcal{Z}(\mathbf{c}), \quad (9)$$

To streamline the notation, we define $\hat{r}(\mathbf{x}_{t-1}, \mathbf{c}) \triangleq \log \frac{\pi(\mathbf{x}_{t-1}|\mathbf{x}_t, \mathbf{c})}{\pi(\mathbf{x}_{t-1}|\mathbf{x}_t^{\text{ref}}, \mathbf{c})}$. Then, we can eliminate the intractable term $\mathcal{Z}(\mathbf{c})$ by considering the difference between the trajectories evolved from any pair of candidate latents within the sampled group $\{\mathbf{x}_{t-1}^k\}_{k=1}^G$, indexed by $i$ and $j$:

$$r(\hat{\mathbf{x}}_0^i, c) - r(\hat{\mathbf{x}}_0^j, \mathbf{c}) = \beta\big(\hat{r}(\mathbf{x}_{t-1}^i, \mathbf{c}) - \hat{r}(\mathbf{x}_{t-1}^j, \mathbf{c})\big), \quad (10)$$

Define $\Delta r_{ij} = r(\hat{\mathbf{x}}_0^i, \mathbf{c}) - r(\hat{\mathbf{x}}_0^j, \mathbf{c})$ and $\Delta\hat{r}_{ij} = \hat{r}(\mathbf{x}_{t-1}^i, \mathbf{c}) - \hat{r}(\mathbf{x}_{t-1}^j, \mathbf{c})$. Inspired by PPO (Schulman et al., 2017), we adopt a clipped objective; the final loss is:

$$\mathcal{L}_{\text{PRDP}}(\mathbf{x}_t) = \mathbb{E}_{i,j} \Bigg[ \max\Bigg( \Big(\Delta\hat{r}_{ij} - \frac{\Delta r_{ij}}{\beta}\Big)^2, $$
$$\Big(\text{clip}\big(\Delta\hat{r}_{ij}, \Delta\hat{r}_{ij}^{\text{old}} - \varepsilon, \Delta\hat{r}_{ij}^{\text{old}} + \varepsilon\big) - \frac{\Delta r_{ij}}{\beta}\Big)^2 \Bigg) \Bigg]$$
(11)

where $\Delta\hat{r}_{ij}^{\text{old}}$ represents the implicit reward difference computed from the previous latent policies and $\varepsilon$ is a hyper-parameter for the clip threshold, and $\beta$ controls the KL regularization strength. It is worth noting that PRDP helps mitigate one-step estimation errors during the early reverse diffusion process.

## 4.3. Stage 2: Trajectory Refinement via GRPO

Thanks to the warm-up phase, the $\mathbf{x}_t$ has undergone a coarse alignment with human preferences, and the single-step projection $\hat{x}_0 = \mathbf{x}_t - t \cdot v_\theta(\mathbf{x}_t)$ can provide a statistically reliable approximation of the final image. The stabilized global structure and reduced latent variance create an ideal window for fine-grained aesthetic preference alignment. However,

since PRDP optimizes reward differences rather than maximizing absolute rewards, it is limited in polishing subtle textures or discovering novel high-reward features. Consequently, we transition to GRPO (Shao et al., 2024), leveraging group-wise comparisons to excavate fine-grained preference details. We compute the advantage $\hat{A}_t^i$ for the $i$-th candidate by normalizing its projected reward against the group statistics:

$$\hat{A}_t^i = \frac{r(\hat{\mathbf{x}}_0^i, \mathbf{c}) - \text{mean}(\{r(\hat{\mathbf{x}}_0^k, \mathbf{c})\}_{k=1}^G)}{\text{std}(\{r(\hat{\mathbf{x}}_0^k, \mathbf{c})\}_{k=1}^G) + \epsilon}. \quad (12)$$

Let $\rho_t^i = \frac{\pi(\mathbf{x}_{t-1}^i | \mathbf{x}_t, \mathbf{c})}{\pi(\mathbf{x}_{t-1}^i | \mathbf{x}_t^{\text{old}}, \mathbf{c})}$ be the probability ratio between the current and previous latent policies. The loss function is defined as:

$$\mathcal{L}_{\text{GRPO}} = \mathbb{E}_{i \sim G} \left[ \max \left( -\hat{A}_t^i \rho_t^i, -\hat{A}_t^i \cdot \text{clip}(\rho_t^i, 1 - \varepsilon, 1 + \varepsilon) \right) \right.$$
$$\left. + \beta \frac{\|\bar{\mathbf{x}}_{t-1}^i - \bar{\mathbf{x}}_{t-1}^{\text{ref}}\|^2}{2\sigma_t^2} \right]. \quad (13)$$

where $\varepsilon$ is the clipping threshold, and $\beta$ controls the regularization strength. The $\bar{\mathbf{x}}_{t-1}^i$ evolve from the candidate latent $\mathbf{x}_t^i$, and $\bar{\mathbf{x}}_{t-1}^{\text{ref}}$ evolve from the reference latent $\mathbf{x}_t^{\text{ref}}$.

### 4.4. Noise Reweighting for Latent Optimization

To encourage early global exploration, we design the diffusion dynamics with decreasing noise levels to transition from structural exploration to fine-grained refinement, and reweight the loss function based on the noise intensity and step size $\sigma_t\sqrt{\Delta t}$ (He et al., 2025). Formally, the latent is updated via $\mathbf{x}_t' \leftarrow \mathbf{x}_t - \eta\sigma_t\sqrt{\Delta t}\nabla_{\mathbf{x}_t}\mathcal{L}$. The pipeline of Flow-TTRL is illustrated in Algorithm 1. The detailed convergence analysis is provided in Appendix A.3.

## 5. Experiments

### 5.1. Experimental Setup

We evaluate Flow-TTRL on two representative flow matching models: Stable Diffusion 3.5 (SD 3.5-M) (Esser et al., 2024) and Flux.1 Dev (Labs et al., 2025). Our method is applied strictly at inference time, requiring no parameter updates to the pre-trained backbones.

We evaluate performance across four key dimensions. **Compositional Image Generation:** We employ GenEval (Ghosh et al., 2023) and T2I-CompBench (Huang et al., 2023; 2025a) to comprehensively assess structural integrity. GenEval utilizes pre-trained object detectors to verify object existence, attributes, and spatial relations, while T2I-CompBench leverages BLIP-based VQA to quantify complex attribute binding and object relationships. For other evaluations, we randomly sample 400 prompts from

---

**Algorithm 1** Flow-TTRL: Test-Time Reinforcement Learning for Flow Matching

1: **Input:** Pre-trained flow network $v_\theta$, Prompt $\mathbf{c}$, Inference steps $T$, Opt. steps $K$, Group size $G$, Learning rate $\eta$, Warm-up threshold $T_{\text{warm}}$, Stop threshold $T_{\text{end}}$.
2: **Output:** Generated image $\mathbf{x}_0$.
3: **Define:** Time step size $\Delta t = 1/T$.
4: **Initialize:** Sample latent $\mathbf{x}_T \sim \mathcal{N}(\mathbf{0}, \mathbf{I})$.
5: **Initialize Reference:** $\mathbf{x}_T^{\text{ref}} \leftarrow \text{sg}(\mathbf{x}_T)$.
6: **for** $t = T$ **down to** 1 **do**
7:     Let continuous time $\tau = t/T$.
8:     **if** $t > T_{\text{end}}$ **then**
9:         *// 1. Preparation*
10:         **Old Policy:** $\mathbf{x}_t^{\text{old}} \leftarrow \text{sg}(\mathbf{x}_t)$
11:         Determine noise intensity $\sigma_t$.
12:         *// 2. Rollout (Sample from Old Policy)*
13:         Sample $G$ candidate latents $\{\mathbf{x}_{t-1}^i\}_{i=1}^G$ from $\pi(\cdot|\mathbf{x}_t^{\text{old}}, \mathbf{c})$ using SDE sampling with noise $\sigma_t$.
14:         **for** $i = 1$ **to** $G$ **do**
15:             Project $\hat{\mathbf{x}}_0^i \leftarrow \mathbf{x}_{t-1}^i - (\tau - \Delta t) \cdot v_\theta(\mathbf{x}_{t-1}^i, \tau - \Delta t)$.
16:             Compute reward $r^i \leftarrow r(\hat{\mathbf{x}}_0^i, \mathbf{c})$.
17:         **end for**
18:         *// 3. Latent Optimization*
19:         **for** $k = 1$ **to** $K$ **do**
20:             **if** $t > T_{\text{warm}}$ **then**
21:                 Compute loss $\mathcal{L} \leftarrow \mathcal{L}_{\text{PRDP}}$ via Eq. 11
22:             **else**
23:                 Compute probability ratio $\rho_t^i \leftarrow \pi(\mathbf{x}_{t-1}^i|\mathbf{x}_t, \mathbf{c})/\pi(\mathbf{x}_{t-1}^i|\mathbf{x}_t^{\text{old}}, \mathbf{c})$.
24:                 Compute advantages $\hat{A}_t^i$ via Eq. 12.
25:                 Compute loss $\mathcal{L} \leftarrow \mathcal{L}_{\text{GRPO}}$ via Eq. 13
26:             **end if**
27:             Reweight loss: $\mathcal{L} \leftarrow \sigma_t\sqrt{\Delta t} \cdot \mathcal{L}$.
28:             Update latent: $\mathbf{x}_t \leftarrow \mathbf{x}_t - \eta\nabla_{\mathbf{x}_t}\mathcal{L}$.
29:         **end for**
30:     **end if**
31:     *// 4. Transition (Evolve both Trajectories)*
32:     Evolve latent: $\mathbf{x}_{t-1} \leftarrow \Phi(\mathbf{x}_t, v_\theta(\mathbf{x}_t, \tau))$.
33:     Evolve reference: $\mathbf{x}_{t-1}^{\text{ref}} \leftarrow \Phi(\mathbf{x}_t^{\text{ref}}, v_\theta(\mathbf{x}_t^{\text{ref}}, \tau))$.
34: **end for**
35: **Return** $\mathcal{D}(\mathbf{x}_0)$.

---

the PartiPrompts, Pick-a-Pic (Kirstain et al., 2023), and Drawbench (Saharia et al., 2022) datasets, respectively. **Human Preference Alignment:** To measure how well the generated images align with general human judgment, we report PickScore (Kirstain et al., 2023), HPS v2 (Wu et al., 2023), and ImageReward (Xu et al., 2023). **Aesthetic Quality:** Visual appeal and artistic quality are evaluated using the LAION-Aesthetics score (AES). **Text-Image Alignment:** To quantify how accurately the generated images reflect the

input text prompts, we report the CLIP Score (Radford et al., 2021). To mitigate randomness, we generate 16 images for each prompt using a fixed set of random seeds for all models and report the averaged results. Detailed hyperparameter configurations and quality metrics are provided in Appendix B.1 and Appendix B.2, respectively.

## 5.2. Main Results

| Model | Overall↑ | Single↑ | Two↑ | Counting↑ | Colors↑ | Position↑ | Color Attrib.↑ |
|---|---|---|---|---|---|---|---|
| *Diffusion Models* | | | | | | | |
| DALL-E 2 (Ramesh et al., 2022) | 0.52 | 0.94 | 0.66 | 0.49 | 0.77 | 0.10 | 0.19 |
| DALL-E 3 (Betker et al., 2023) | 0.67 | 0.96 | 0.87 | 0.47 | 0.83 | 0.43 | 0.45 |
| *Autoregressive Models* | | | | | | | |
| Emu3 (Wang et al., 2024) | 0.54 | 0.98 | 0.71 | 0.34 | 0.81 | 0.17 | 0.21 |
| Show-o (Xie et al., 2024) | 0.53 | 0.95 | 0.52 | 0.49 | 0.82 | 0.11 | 0.28 |
| GPT-4o (Hurst et al., 2024) | 0.84 | 0.99 | 0.92 | 0.85 | 0.92 | 0.75 | 0.61 |
| JanusFlow (Ma et al., 2025) | 0.63 | 0.97 | 0.59 | 0.45 | 0.83 | 0.53 | 0.42 |
| Janus-Pro-7B (Chen et al., 2025) | 0.80 | 0.99 | 0.89 | 0.59 | 0.90 | 0.79 | 0.66 |
| *Flow Matching Models* | | | | | | | |
| Flux.1 Dev (Labs et al., 2025) | 0.66 | 0.98 | 0.81 | 0.74 | 0.79 | 0.22 | 0.45 |
| SD 3.5-M (Esser et al., 2024) | 0.63 | 0.98 | 0.78 | 0.50 | 0.81 | 0.24 | 0.52 |
| SANA-1.5 4.8B (Xie et al., 2025) | 0.81 | 0.99 | 0.93 | 0.86 | 0.84 | 0.59 | 0.65 |
| SD3.5-L (Esser et al., 2024) | 0.71 | 0.98 | 0.89 | 0.73 | 0.83 | 0.34 | 0.47 |
| *Test-time Optimization Methods* | | | | | | | |
| SD 3.5-M + DAS (Kim et al., 2025) | 0.83 | **1.00** | 0.98 | 0.89 | 0.86 | 0.57 | 0.67 |
| SD 3.5-M + TITAN-Guide (Simon et al., 2025) | 0.74 | **1.00** | 0.91 | 0.76 | 0.85 | 0.32 | 0.60 |
| Flux.1 Dev + DAS (Kim et al., 2025) | 0.80 | **1.00** | 0.96 | 0.81 | 0.90 | 0.37 | 0.73 |
| Flux.1 Dev + TITAN-Guide (Simon et al., 2025) | 0.76 | 0.99 | 0.90 | 0.83 | 0.88 | 0.34 | 0.64 |
| *Flow-RL based Methods* | | | | | | | |
| SD 3.5-M + Flow-GRPO | 0.95 | **1.00** | 0.99 | 0.95 | 0.92 | **0.99** | 0.86 |
| SD 3.5-M + TempFlow-GRPO | **0.97** | **1.00** | **1.00** | **0.96** | **0.95** | **0.99** | **0.91** |
| Flux.1 Dev + Flow-TTRL | 0.83 | **1.00** | 0.97 | 0.94 | 0.90 | 0.41 | 0.77 |
| SD 3.5-M + Flow-TTRL | 0.87 | **1.00** | 0.99 | 0.99 | 0.90 | 0.55 | 0.85 |

*Table 1.* **Experimental results on GenEval.** The **Top-5** results are highlighted with different shades of blue, where darker colors indicate higher rankings. The best results are highlighted in **bold**.

| Model | Attribute Binding | | | Object Relationship | | Complex |
|---|---|---|---|---|---|---|
| | Color↑ | Shape↑ | Texture↑ | 2D-Spatial↑ | Non-Spatial↑ | |
| DALL-E 3 (Betker et al., 2023) | 0.7785 | 0.6205 | 0.7036 | 0.2865 | 0.3003 | 0.3773 |
| Janus-Pro-7B (Chen et al., 2025) | 0.5145 | 0.3323 | 0.4069 | 0.1566 | 0.3137 | — |
| Emu3 (Wang et al., 2024) | 0.7913 | 0.5846 | 0.7422 | — | — | — |
| Flux.1 Dev (Labs et al., 2025) | 0.7407 | 0.5718 | 0.6922 | 0.2863 | 0.3127 | 0.3703 |
| SD 3.5-M (Esser et al., 2024) | 0.7994 | 0.5669 | 0.7338 | 0.2850 | 0.3146 | 0.3542 |
| SD 3.5-M + Flow-GRPO (Liu et al., 2025a) | 0.8379 | 0.6130 | 0.7236 | 0.5447 | 0.3195 | — |
| SD 3.5-M + DAS (Kim et al., 2025) | 0.8561 | 0.6482 | 0.7717 | 0.3679 | 0.3294 | 0.3847 |
| SD 3.5-M + TITAN-Guide (Simon et al., 2025) | 0.8468 | 0.6235 | 0.7689 | 0.3725 | 0.3276 | 0.3791 |
| Flux.1 Dev + DAS (Kim et al., 2025) | 0.8371 | 0.6779 | 0.7689 | 0.3662 | 0.3161 | 0.3951 |
| Flux.1 Dev + TITAN-Guide (Simon et al., 2025) | 0.8217 | 0.6511 | 0.7673 | 0.3786 | 0.3154 | 0.3853 |
| Flux.1 Dev + Flow-TTRL | 0.8804 | 0.6717 | 0.7958 | 0.4390 | 0.3229 | **0.4179** |
| SD 3.5-M + Flow-TTRL | **0.9042** | **0.7361** | **0.8261** | 0.4414 | **0.3319** | 0.4045 |

*Table 2.* **Results on T2I-CompBench.** The **Top-3** results are highlighted in different shades of blue, where darker colors indicate higher rankings. Markers follow Table 1.

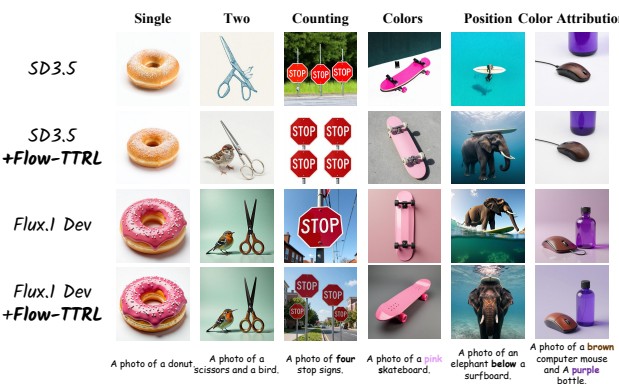

*Figure 3.* Qualitative Comparison on the GenEval Datasets.

As reported in Table 1 and Table 2, our experimental results demonstrate that Flow-TTRL bridges the gap between open-source flow models and top-tier commercial systems. Remarkably, despite the absence of parameter updates, our method rivals the performance of established RL-based fine-tuning, even demonstrating superior results across specific aesthetic and alignment metrics. Furthermore, it consistently outperforms representative test-time optimization approaches, such as DAS and TITAN-Guide. This validates the effectiveness of test-time scaling, suggesting that increasing inference compute is a highly efficient alternative to model training for maximizing generation quality. Figure 3 further illustrates this capability qualitatively: for simple prompts, Flow-TTRL enhances detail expressiveness and refinement, while for complex prompts, it achieves superior aesthetic quality without compromising semantic alignment. Computational efficiency comparisons, extended experiments on PartiPrompts, Pick-a-Pic, and DrawBench, and additional qualitative results are detailed in Appendix D.1, D.2, and D.11, respectively. In Appendix D.8, we conduct comparative experiments with other latent selection methods (Best-of-N and Beam Search) to demonstrate the significance of latent space optimization.

### 5.3. Analysis

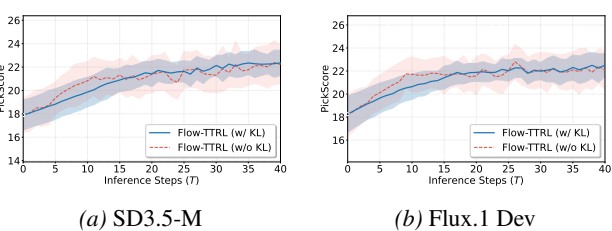

*(a)* SD3.5-M   *(b)* Flux.1 Dev

*Figure 4.* **Effect of KL in mitigating reward hacking.** The KL constraint facilitates a stable and consistent increase in PickScore during the Flow-TTRL process, preventing the optimization from collapsing into reward hacking.

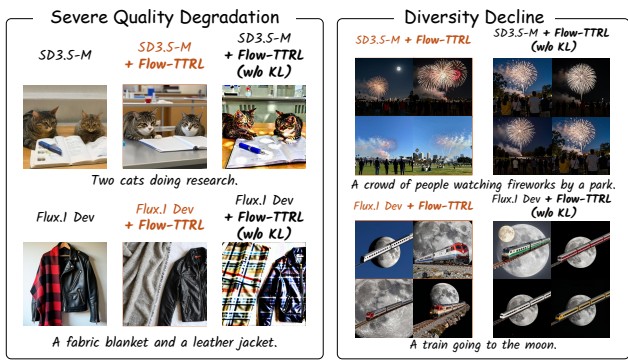

*Figure 5.* **Qualitative comparison of KL effects.** The KL constraint effectively prevents quality degradation and diversity decline.

**Reward Hacking.** Reward hacking refers to a phenomenon

| Model | GenEval↑ | Complex↑ | Human Preference Alignment | | | Aesthetic Quality |
| | | | PickScore↑ | HPS v2↑ | ImageReward↑ | Aesthetic↑ |
|---|---|---|---|---|---|---|
| SD3.5-M + Flow-TTRL | **0.87** | **0.4045** | **22.24** | **0.299** | 1.187 | 5.733 |
| SD3.5-M + Flow-TTRL (w/o KL) | 0.71 | 0.3569 | 22.11 | 0.287 | 0.934 | 5.328 |
| SD3.5-M + Flow-TTRL (w/o PRDP) | 0.81 | 0.3792 | 22.17 | 0.291 | 1.221 | 5.553 |
| SD3.5-M + Flow-TTRL (w/o GRPO) | 0.72 | 0.3612 | 22.07 | 0.289 | 0.896 | 5.721 |
| SD3.5-M + Flow-TTRL (w/o NR) | 0.85 | 0.3895 | 22.13 | 0.294 | **1.243** | **5.892** |
| SD3.5-M | 0.63 | 0.3542 | 21.89 | 0.285 | 0.727 | 5.631 |
| Flux.1 Dev + Flow-TTRL | **0.83** | **0.4179** | 22.67 | **0.322** | **1.340** | 6.185 |
| Flux.1 Dev + Flow-TTRL (w/o KL) | 0.69 | 0.3697 | 22.39 | 0.304 | 1.114 | 5.893 |
| Flux.1 Dev + Flow-TTRL (w/o PRDP) | 0.76 | 0.3861 | 22.61 | 0.302 | 1.307 | 5.981 |
| Flux.1 Dev + Flow-TTRL (w/o GRPO) | 0.70 | 0.3765 | 22.41 | 0.317 | 0.897 | 6.034 |
| Flux.1 Dev + Flow-TTRL (w/o NR) | 0.82 | 0.3997 | **22.69** | 0.316 | 1.329 | **6.235** |
| Flux.1 Dev | 0.66 | 0.3703 | 22.35 | 0.316 | 0.929 | 6.086 |

*Table 3.* **Ablation results on Compositional Image Generation, Human Preference Alignment, and Aesthetic Quality.** We evaluate performance on GenEval, the Complex task of T2I-CompBench, and the Pick-a-Pic benchmark. NR denotes noise reweighting. The best results are highlighted in **bold**.

where a model exploits loopholes in the reward function to achieve high scores without fulfilling the true objective. To mitigate this, we incorporate a KL constraint relative to the reference latent. Our ablation studies in Table 3 demonstrate that removing this KL constraint leads to a significant degradation in image generation quality. Furthermore, the reward trajectories in Figure 4 show that our KL constraint ensures a stable and consistent PickScore increase during the Flow-TTRL process, effectively preventing the reward oscillations and hacking behavior observed in the unconstrained baseline. Furthermore, we provide qualitative comparisons in Figure 5, which reveal that reward hacking causes severe quality degradation and a sharp decline in diversity. The model tends to collapse into a few high-scoring but distorted patterns, destroying the generative prior. Please refer to Appendix D.3, D.4, D.5, D.9, and D.10 for further details on the KL analysis, verifiable reward experiments, iteration number effects, multiple reward analysis, and attention patterns visualization, respectively.

**Effect of Two Stage Procedure.** The ablation results in Table 3 and visual examples in Figure 6 demonstrate the synergy of our two-stage strategy. Since the quantity and layout of objects are primarily determined during the early denoising stages, the PRDP warm-up (Stage 1) provides a stable structural foundation. By optimizing reward differences in this high-noise regime, PRDP ensures the presence and correct positioning of objects, effectively preventing trajectory collapse or omission. Subsequently, GRPO refinement (Stage 2) leverages group-wise comparisons to polish subtle textures and aesthetic details once the global structure is stabilized. Ablation studies confirm that PRDP is essential for structural fidelity, while GRPO is critical for human preference alignment. For a comprehensive discussion on the rationale behind our two-stage PRDP and GRPO approach, please refer to Appendix C.

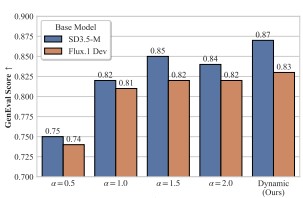 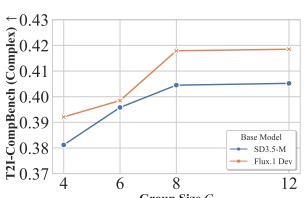

*(a)* Ablation of noise level $\alpha$ on GenEval.

*(b)* Effect of group size $G$ on Complex task.

*Figure 7.* **Ablation studies on noise reweighting and group size.** (a) Dynamic noise reweighting outperforms all fixed noise levels. (b) Performance scales with group size and saturates at $G = 8$.

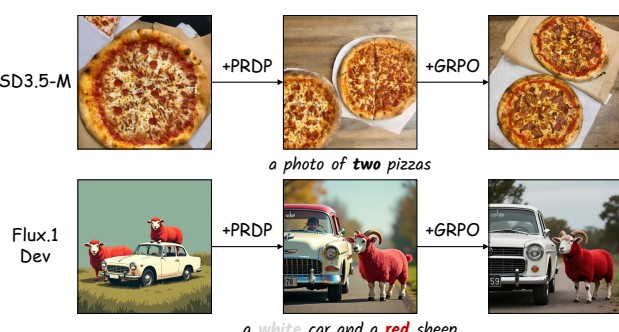

*a photo of **two** pizzas*

*a **white** car and a **red** sheep*

*Figure 6.* **Visual comparison of the effects of PRDP and GRPO.** PRDP ensures the correctness of object rendering and spatial composition, preventing the loss of key elements. GRPO further refines the image by optimizing subtle textures and aesthetic preferences.

**Effect of Noise Reweighting.** As shown in Table 3 and Figure 7a, we investigate the impact of different noise levels $\alpha$ on GenEval performance. For both SD3.5 and Flux.1 Dev, we observe that the optimal static noise level lies within $[1.0, 2.0]$. Extreme values of $\alpha$ degrade performance: an excessively small $\alpha$ restricts the exploration space, while an

overly large $\alpha$ introduces instability into the optimization trajectory. Notably, our dynamic noise reweighting strategy, which adaptively modulates weights according to the reverse-diffusion stages, consistently outperforms all static configurations. This confirms that a time-dependent noise schedule is superior in balancing exploration and stability throughout the generation process. We provide intuitive examples regarding the effect of stochasticity levels on the Flow-TTRL optimization process in Appendix D.6. Furthermore, we provide the experimental results of different $\alpha$ scheduling strategies in Appendix D.7.

**Effect of Group Size.** The impact of group size $G$ on the T2I-CompBench (Complex) task is illustrated in Figure 7b. It is worth noting that $G$ simultaneously dictates the execution efficiency of both PRDP and GRPO. For both models, generation quality scales with the number of candidates, as a larger $G$ provides more accurate advantage estimates for RL. However, the performance gain plateaus significantly after $G = 8$. Empirically, $G = 8$ provides an ideal tradeoff between computational efficiency and alignment accuracy.

## 6. Conclusion

In this work, we introduced Flow-TTRL, a novel framework to leverage test-time reinforcement learning to flow-matching models. By conceptualizing latent trajectories as an implicit policy, Flow-TTRL facilitates precise alignment through dynamic trajectory steering within the learned vector field while preserving the integrity of pre-trained weights. Our two-stage strategy successfully reconciles the fundamental requirements of structural stability and fine-grained preference refinement. Extensive empirical evaluations demonstrate that Flow-TTRL yields substantial improvements in compositional image generation, aesthetic quality, and human preference alignment. Flow-TTRL establishes a versatile and efficient paradigm for the alignment of large-scale generative models.

## Acknowledgements

This work was supported by the National Natural Science Foundation of China (Grant No. 62337001), "the Pioneer" and "Leading Goose" R&D Program of Zhejiang (Grant No. 2025C02022), and the Major Program of the Natural Science Foundation of Zhejiang Province, China (Grant No. LD26F020003).

## Impact Statement

This paper presents work whose goal is to advance the field of Machine Learning. There are many potential societal consequences of our work, none which we feel must be specifically highlighted here.

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

# Appendix: Table of Contents

# A. Proofs

## A.1. The Optimal Latent in PRDP Objective in Flow-TTRL

In this section, we provide the derivation for the equivalence between maximizing the RL objective and minimizing the KL divergence to the optimal Gibbs distribution.

We define the objective function $\mathcal{J}(\mathbf{x}_t)$ as proposed in the main text:

$$\mathcal{J}(\mathbf{x}_t) = \mathbb{E}_{\mathbf{x}_{t-\Delta t} \sim \pi(\cdot|\mathbf{x}_t, \mathbf{c})} \left[ r(\mathbf{x}_0, \mathbf{c}) - \beta \log \frac{\pi(\mathbf{x}_{t-\Delta t}|\mathbf{x}_t, \mathbf{c})}{\pi(\mathbf{x}_{t-\Delta t}|\mathbf{x}_t^{\mathrm{ref}}, \mathbf{c})} \right]. \tag{14}$$

We compare the current policy distribution $\pi(\mathbf{x}_{t-\Delta t}|\mathbf{x}_t, \mathbf{c})$ with the optimal Gibbs distribution $q^*(\mathbf{x}_{t-\Delta t})$, defined as:

$$q^*(\mathbf{x}_{t-\Delta t}) = \frac{1}{\mathcal{Z}} \pi(\mathbf{x}_{t-\Delta t}|\mathbf{x}_t^{\mathrm{ref}}, \mathbf{c}) \exp\left( \frac{r(\mathbf{x}_0, \mathbf{c})}{\beta} \right), \tag{15}$$

where $\mathcal{Z} = \int \pi(\mathbf{x}_{t-\Delta t}|\mathbf{x}_t^{\mathrm{ref}}, \mathbf{c}) \exp\left( \frac{r(\mathbf{x}_0, \mathbf{c})}{\beta} \right) d\mathbf{x}_{t-\Delta t}$ is the partition function, which is constant with respect to $\mathbf{x}_t$.

The KL divergence from $\pi(\cdot|\mathbf{x}_t, \mathbf{c})$ to $q^*$ is given by:

$$\begin{aligned}
\mathrm{KL}\big(\pi(\cdot|\mathbf{x}_t, \mathbf{c}) \,\|\, q^*\big) &= \mathbb{E}_{\mathbf{x}_{t-\Delta t} \sim \pi(\cdot|\mathbf{x}_t, \mathbf{c})} \left[ \log \frac{\pi(\mathbf{x}_{t-\Delta t}|\mathbf{x}_t, \mathbf{c})}{q^*(\mathbf{x}_{t-\Delta t})} \right] \\
&= \mathbb{E}_{\mathbf{x}_{t-\Delta t} \sim \pi(\cdot|\mathbf{x}_t, \mathbf{c})} \left[ \log \pi(\mathbf{x}_{t-\Delta t}|\mathbf{x}_t, \mathbf{c}) - \log\left( \frac{1}{\mathcal{Z}} \pi(\mathbf{x}_{t-\Delta t}|\mathbf{x}_t^{\mathrm{ref}}, \mathbf{c}) e^{\frac{r(\mathbf{x}_0, \mathbf{c})}{\beta}} \right) \right] \\
&= \mathbb{E}_{\mathbf{x}_{t-\Delta t} \sim \pi(\cdot|\mathbf{x}_t, \mathbf{c})} \left[ \log \pi(\mathbf{x}_{t-\Delta t}|\mathbf{x}_t, \mathbf{c}) - \log \pi(\mathbf{x}_{t-\Delta t}|\mathbf{x}_t^{\mathrm{ref}}, \mathbf{c}) - \frac{1}{\beta} r(\mathbf{x}_0, \mathbf{c}) + \log \mathcal{Z} \right].
\end{aligned} \tag{16}$$

Multiplying both sides by $-\beta$ and rearranging the terms:

$$
\begin{aligned}
-\beta \cdot \mathrm{KL}\big(\pi(\cdot|\mathbf{x}_t, \mathbf{c}) \,\|\, q^*\big) &= \mathbb{E}_{\mathbf{x}_{t-\Delta t} \sim \pi(\cdot|\mathbf{x}_t, \mathbf{c})} \left[ r(\mathbf{x}_0, \mathbf{c}) - \beta \left( \log \pi(\mathbf{x}_{t-\Delta t}|\mathbf{x}_t, \mathbf{c}) - \log \pi(\mathbf{x}_{t-\Delta t}|\mathbf{x}_t^{\mathrm{ref}}, \mathbf{c}) \right) \right] - \beta \log \mathcal{Z} \\
&= \mathbb{E}_{\mathbf{x}_{t-\Delta t} \sim \pi(\cdot|\mathbf{x}_t, \mathbf{c})} \left[ r(\mathbf{x}_0, \mathbf{c}) - \beta \log \frac{\pi(\mathbf{x}_{t-\Delta t}|\mathbf{x}_t, \mathbf{c})}{\pi(\mathbf{x}_{t-\Delta t}|\mathbf{x}_t^{\mathrm{ref}}, \mathbf{c})} \right] - \mathrm{const} \\
&= \mathcal{J}(\mathbf{x}_t) - \mathrm{const}.
\end{aligned}
\tag{17}
$$

Since $\beta > 0$ and $\log \mathcal{Z}$ is independent of $\mathbf{x}_t$, we conclude that:

$$
\arg\max_{\mathbf{x}_t} \mathcal{J}(\mathbf{x}_t) = \arg\min_{\mathbf{x}_t} \mathrm{KL}\big(\pi(\cdot|\mathbf{x}_t, \mathbf{c}) \,\|\, q^*\big).
\tag{18}
$$

Therefore, the optimal latent state $\mathbf{x}_t^*$ is reached when the induced local transition matches the target distribution $q^*(\mathbf{x}_{t-\Delta t}) \propto \pi(\mathbf{x}_{t-\Delta t}|\mathbf{x}_t^{\mathrm{ref}}, \mathbf{c}) \exp\left(r(\mathbf{x}_0, \mathbf{c})/\beta\right)$.

### A.2. Bounded Error of Single-Step Projection

In this section, we provide a detailed proof of the bounded error of the single-step projection. The proof is analogous to (Na et al., 2025) (Appendix A.4).

**Proposition 1 (Error Bound of Conditional Mean Approximation for General Reward Models).**

Let the text-conditioned reward function used in Test-Time Reinforcement Learning be defined as a composition $h(\mathbf{x}_0; \mathbf{c}) = g(f(\mathbf{x}_0); \mathbf{c})$, where $f : \mathcal{X} \to \mathbb{R}^d$ is a feature extractor and $g : \mathbb{R}^d \times \mathcal{Y} \to \mathbb{R}$ is a scoring head. Assume the following regularity conditions hold:

1. The scoring function $g(\cdot; \mathbf{c})$ is $L_g$-Lipschitz continuous with respect to the feature embeddings, meaning $|g(z_1; \mathbf{c}) - g(z_2; \mathbf{c})| \leq L_g \|z_1 - z_2\|$ for some constant $L_g > 0$.

2. The feature extractor $f$ has bounded gradients on the support of the conditional distribution, such that $\max_{\mathbf{x}} \|\nabla_x f(\mathbf{x})\| \leq M_f$.

Then, the approximation error of evaluating the reward on the conditional mean $\bar{\mathbf{x}}_0$ instead of the true conditional expectation is strictly upper-bounded by:

$$
\left| \mathbb{E}_{\mathbf{x}_0 \sim p(\mathbf{x}_0|\mathbf{x}_t, \mathbf{c})}[h(\mathbf{x}_0; \mathbf{c})] - h(\bar{\mathbf{x}}_0; \mathbf{c}) \right| \leq L_g \cdot M_f \cdot m_1^{flow},
\tag{19}
$$

where $m_1^{flow} := \mathbb{E}_{\mathbf{x}_0 \sim p(\mathbf{x}_0|\mathbf{x}_t, \mathbf{c})}[\|\mathbf{x}_0 - \bar{\mathbf{x}}_0\|]$ represents the mean absolute deviation of the target data conditional distribution.

**Proof.** We begin by expressing the absolute error between the expected reward and the reward evaluated at the conditional mean:

$$
\mathrm{Error} = \left| \mathbb{E}_{\mathbf{x}_0 \sim p(\mathbf{x}_0|\mathbf{x}_t, \mathbf{c})}[h(\mathbf{x}_0; \mathbf{c})] - h(\bar{\mathbf{x}}_0; \mathbf{c}) \right|.
\tag{20}
$$

By the linearity of expectation, we can rewrite this as the absolute value of the expected difference:

$$
\mathrm{Error} = \left| \mathbb{E}_{\mathbf{x}_0 \sim p(\mathbf{x}_0|\mathbf{x}_t, \mathbf{c})} \left[ g(f(\mathbf{x}_0); \mathbf{c}) - g(f(\bar{\mathbf{x}}_0); \mathbf{c}) \right] \right|
\tag{21}
$$

Applying the triangle inequality for expectations (Jensen's inequality), we bound the absolute value of the expectation by the expectation of the absolute value:

$$
\mathrm{Error} \leq \mathbb{E}_{\mathbf{x}_0 \sim p(\mathbf{x}_0|\mathbf{x}_t, \mathbf{c})} \left[ |g(f(\mathbf{x}_0); \mathbf{c}) - g(f(\bar{\mathbf{x}}_0); \mathbf{c})| \right].
\tag{22}
$$

Given the assumption that the scoring head $g$ is $L_g$-Lipschitz continuous, we can bound the reward difference by the distance in the latent feature space:

$$
\mathrm{Error} \leq \mathbb{E}_{\mathbf{x}_0 \sim p(\mathbf{x}_0|\mathbf{x}_t, \mathbf{c})} \left[ L_g \|f(\mathbf{x}_0) - f(\bar{\mathbf{x}}_0)\| \right].
\tag{23}
$$

Applying the mean value inequality to the continuously differentiable feature extractor $f$, the distance in the feature space is bounded by the maximum Jacobian norm multiplied by the distance in the data space:

$$
\|f(\mathbf{x}_0) - f(\bar{\mathbf{x}}_0)\| \leq \max_x \|\nabla_x f(\mathbf{x})\| \cdot \|\mathbf{x}_0 - \bar{\mathbf{x}}_0\| \leq M_f \cdot \|\mathbf{x}_0 - \bar{\mathbf{x}}_0\|
\tag{24}
$$

Substituting this deterministic bound back into the expectation yields:

$$\text{Error} \leq \mathbb{E}_{\mathbf{x}_0 \sim p(\mathbf{x}_0|\mathbf{x}_t,\mathbf{c})} \left[ L_g \cdot M_f \cdot ||\mathbf{x}_0 - \bar{\mathbf{x}}_0|| \right]. \tag{25}$$

Factoring out the constants $L_g$ and $M_f$, and substituting the definition of the conditional mean absolute deviation $m_1^{flow}$, we arrive at the final bound:

$$|\mathbb{E}_{\mathbf{x}_0 \sim p(\mathbf{x}_0|\mathbf{x}_t,\mathbf{c})}[h(\mathbf{x}_0;\mathbf{c})] - h(\bar{\mathbf{x}}_0;\mathbf{c})| \leq L_g \cdot M_f \cdot m_1^{flow}. \tag{26}$$

### A.3. Convergence Analysis

In this section, we provide a theoretical analysis of the convergence properties of the latent optimization process in Flow-TTRL. Our analysis is analogous to (Tang et al., 2024) (Appendix B).

We define the optimization objective at each denoising step $t$ as $\mathcal{J}(x_t)$, which corresponds to the objective function in Eq. 6:

$$\mathcal{J}(x_t) = \mathbb{E}_{xt-1 \sim \pi(x_{t-1}|x_t,c)}[r(\Phi_\theta(x_{t-1:0},c))] - \beta \text{KL}(\pi(\cdot|x_t,c)|\pi(\cdot|x_t^{\text{ref}},c)). \tag{27}$$

**Assumption 1** (*L*-smoothness)**.** We assume that the objective function $\mathcal{J}(x_t)$ is $L$-smooth with respect to the latent variable $x_t$ in the latent manifold. That is, for any $x_t^{(1)}, x_t^{(2)}$, its gradient $\nabla \mathcal{J}$ is $L$-Lipschitz continuous:

$$|\nabla \mathcal{J}(x_t^{(1)}) - \nabla \mathcal{J}(x_t^{(2)})| \leq L|x_t^{(1)} - x_t^{(2)}|. \tag{28}$$

This smoothness assumption is widely adopted in the analysis of latent space optimizations for diffusion models and has been empirically validated as reasonable.

Under Assumption 1, the *Descent Lemma* (Bertsekas, 1997) ensures that for any two successive optimization steps $x_t^{(k)}$ and $x_t^{(k+1)}$, the objective value satisfies:

$$\mathcal{J}(x_t^{(k+1)}) \geq \mathcal{J}(x_t^{(k)}) + \langle \nabla \mathcal{J}(x_t^{(k)}), x_t^{(k+1)} - x_t^{(k)} \rangle - \frac{L}{2}|x_t^{(k+1)} - x_t^{(k)}|^2. \tag{29}$$

In Flow-TTRL, the latent variable is updated using gradient ascent with a learning rate $\eta$:

$$x_t^{(k+1)} = x_t^{(k)} + \eta \nabla \mathcal{J}(x_t^{(k)}). \tag{30}$$

By substituting the update rule (30) into the inequality (29), we obtain:

$$\mathcal{J}(x_t^{(k+1)}) - \mathcal{J}(x_t^{(k)}) \geq \langle \nabla \mathcal{J}(x_t^{(k)}), \eta \nabla \mathcal{J}(x_t^{(k)}) \rangle - \frac{L}{2}|\eta \nabla \mathcal{J}(x_t^{(k)})|^2 \tag{31}$$

$$= \eta |\nabla \mathcal{J}(x_t^{(k)})|^2 - \frac{L\eta^2}{2}|\nabla \mathcal{J}(x_t^{(k)})|^2 \tag{32}$$

$$= \eta \left(1 - \frac{L\eta}{2}\right)|\nabla \mathcal{J}(x_t^{(k)})|^2. \tag{33}$$

Provided that the step size satisfies $0 < \eta < \frac{2}{L}$, the term $\eta(1 - \frac{L\eta}{2})$ remains strictly positive. Consequently, the objective value $\mathcal{J}(x_t)$ is guaranteed to be monotonically non-decreasing at each iteration $k$:

$$\mathcal{J}(x_t^{(k+1)}) \geq \mathcal{J}(x_t^{(k)}). \tag{34}$$

## B. Implementation Details

### B.1. Hyperparameter Configuration

For the inference configuration, we set the total number of sampling steps $T = 40$ for both Flux.1 Dev and SD 3.5-M, with the classifier-free guidance (CFG) scale fixed at 3.5. To balance exploration and stability, we implement a linear decay schedule for the noise intensity $\sigma_t$ in the SDE (Eq. 7), scaling from 1.5 at the start of inference down to 0.5. The optimization process is divided into three distinct phases:

- **Phase 1 (Warm-up):** In the first $15\%$ of timesteps (high-noise regime), we employ the PRDP objective with a KL-regularization coefficient $\beta = 0.0002$ to prevent early trajectory collapse.

- **Phase 2 (Refinement):** For the subsequent $35\%$ of timesteps, we transition to the GRPO objective. Here, we relax the regularization to $\beta = 0.00015$.

- **Phase 3 (Standard Evolution):** In the final $50\%$ of timesteps, optimization is halted ($T_{\text{end}}$), and the trajectory evolves via the standard ODE solver.

Across optimization steps, we set the number of gradient updates per timestep $K = 2$ and the clipping threshold $\varepsilon = 0.0002$. Regarding the learning rate $\eta$, we adopt model-specific values to accommodate the distinct ratio distributions and gradient magnitudes inherent to each architecture. Specifically, we set $\eta = 750$ for Flux.1 Dev and $\eta = 500$ for SD 3.5-M. Following (Xue et al., 2025), our method incorporates a multi-reward function for the final reward calculation, assigning weights of 0.7 to HPS v2 and 0.3 to CLIP Score. The group size is set to $G = 8$. Our method is computationally efficient. All experiments can be performed on a single NVIDIA A800 (40GB) GPU. It is worth noting that, when combined with 4-bit quantization and attention slicing, Flow-TTRL can operate within 24GB of VRAM, making it more accessible for diverse hardware setups.

To ensure a consistent and reproducible evaluation, all experimental results reported in this paper are based on this fixed configuration, unless otherwise specified. However, it is worth noting that the optimal hyperparameters for various inference-time enhancement methods may fluctuate depending on the specific characteristics and complexity of individual prompts. We provide a detailed discussion and qualitative evidence in Appendix D.12, illustrating how these hyperparameters can be manually calibrated to correct suboptimal generations.

### B.2. Quality Metrics

The metrics are listed as follows:

- **CLIPScore** (Radford et al., 2021): a reference-free metric that quantifies the semantic alignment between text and image via the cosine similarity of their embeddings.

- **Aesthetic Score**: a CLIP-based linear predictor trained on the LAION-Aesthetics dataset to evaluate visual appeal and artistic quality.

- **PickScore** (Kirstain et al., 2023): a preference model trained on large-scale user interactions from the Pick-a-Pic dataset to effectively predict human choices.

- **ImageReward** (Xu et al., 2023): a general-purpose human preference reward model trained via RLHF to capture text–image alignment, visual fidelity, and harmlessness.

- **HPS v2** (Wu et al., 2023): a metric trained on large-scale human annotations that aligns better with human aesthetic and structural preferences.

## C. Rationale for Two-Stage PRDP and GRPO

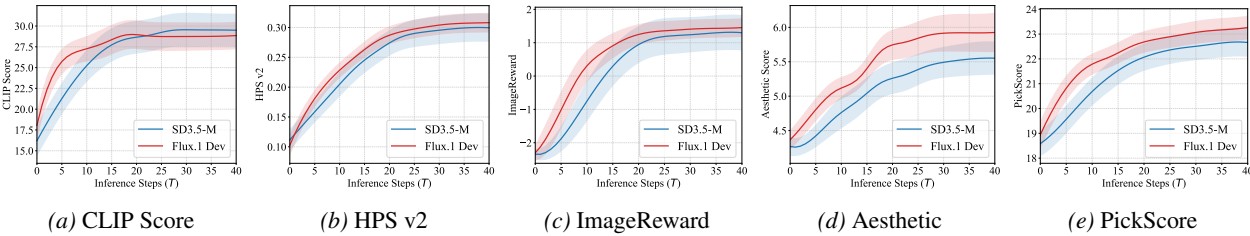

*(a) CLIP Score*  *(b) HPS v2*  *(c) ImageReward*  *(d) Aesthetic*  *(e) PickScore*

*Figure 8.* **Reward trajectories during Flow-TTRL.** Results are averaged over a subset of 200 prompts from PartiPrompts. The steep initial climb confirms that early-to-mid stages are critical for the denoising process.

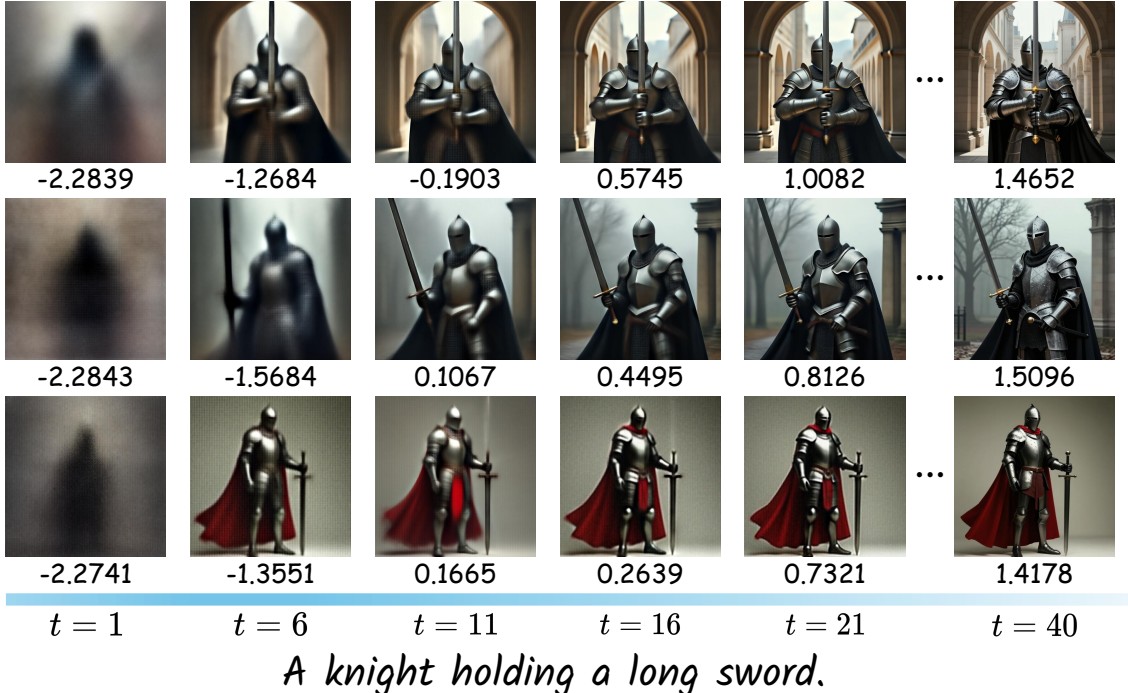

| -2.2839 | -1.2684 | -0.1903 | 0.5745 | 1.0082 | 1.4652 |
| -2.2843 | -1.5684 | 0.1067 | 0.4495 | 0.8126 | 1.5096 |
| -2.2741 | -1.3551 | 0.1665 | 0.2639 | 0.7321 | 1.4178 |

$t = 1$    $t = 6$    $t = 11$    $t = 16$    $t = 21$    $t = 40$

*A knight holding a long sword.*

*Figure 9.* **Evolution of visual features during diffusion.** Based on ImageReward scores, early steps primarily determine global structures like positioning and lighting. Middle steps refine fine-grained textures and aesthetic details. This semantic transition justifies switching from the coarse, stability-focused PRDP to the fine, aesthetics-focused GRPO.

This section details the theoretical underpinnings of our two-stage optimization strategy, validated on a subset of 200 prompts from the PartiPrompts benchmark. The reward trajectories across Aesthetic, PickScore, HPS v2, ImageReward, and CLIPScore (Figure 8) consistently show that the primary reward gains occur within the early-to-mid diffusion steps. **Stage 1: Structural Alignment via PRDP.** Qualitative analysis in Figure 9 reveals that the earliest denoising steps determine global semantics, such as object positioning and lighting. However, optimization in this high-noise regime is perilous: reward models are inaccurate, absolute scores exhibit low variance, and the latent state $\mathbf{x}_t$ is fragile. PRDP mitigates this by regressing on the pairwise reward differences. This relative contrast mechanism, combined with MSE-based optimization, enables the model to focus on substantial structural improvements while filtering out noise-induced micro-variations, thereby preventing trajectory collapse. **Stage 2: Aesthetic Refinement via GRPO.** As the trajectory stabilizes in the mid-diffusion phase, the single-step ODE projection $\hat{\mathbf{x}}_0$ becomes a statistically reliable approximation. At this juncture, the optimization goal shifts from "what" to "how"—refining textures, materials, and fine-grained details (Figure 9). GRPO excels here by leveraging group-wise advantage normalization to excavate subtle aesthetic preferences that simple difference regression might overlook.

In summary, the early diffusion phase is decisive for establishing the **object quantity, spatial layout, and global structure**, whereas subsequent stages focus on refining **fine-grained aesthetic details and texture fidelity**. Consequently, we employ PRDP in the early stage to provide a stable structural warm-up, ensuring compositional correctness under high uncertainty. This is followed by GRPO to excavate subtle human preferences once the denoising trajectory is stabilized. This two-stage transition from "coarse structural alignment" to "fine-grained preference excavation" is essential for generating images that are both semantically accurate and aesthetically superior.

## D. Extended Experimental Results

### D.1. Computation Efficiency

To evaluate the practicality of Flow-TTRL, we analyze the trade-off between generation quality and computational cost in Figure 10, comparing our method against base models, Best-of-N (BoN), and fine-tuning models (Flow-GRPO, TempFlow-GRPO). While BoN relies on brute-force sampling with a fixed budget of approximately 8 minutes per prompt, Flow-TTRL

employs a heuristic search strategy that achieves superior performance with significantly reduced latency. For instance, our method surpasses the BoN baseline in just ∼2.2 minutes (N=40). Interestingly, Flow-TTRL remains effective even with minimal inference steps ($N = 10$), suggesting that extensive search is not a prerequisite for quality gains. We hypothesize this is due to an implicit search for clean projections. Specifically, since reward models naturally favor latent trajectories that produce cleaner and more recognizable $\hat{x}_0$ projections, Flow-TTRL performs a targeted search for such "clean" paths. This explains why the model maintains high-fidelity generation even in sparse-step regimes. Consequently, Flow-TTRL offers a dynamic balance between efficiency and quality, providing a flexible, training-free alternative to the computationally expensive offline training (e.g., >1k GPU hours) required by methods like Flow-GRPO. Furthermore, Flow-TTRL demonstrates a superior efficiency-performance trade-off compared to other existing test-time optimization methods (DAS, TITAN-Guide).

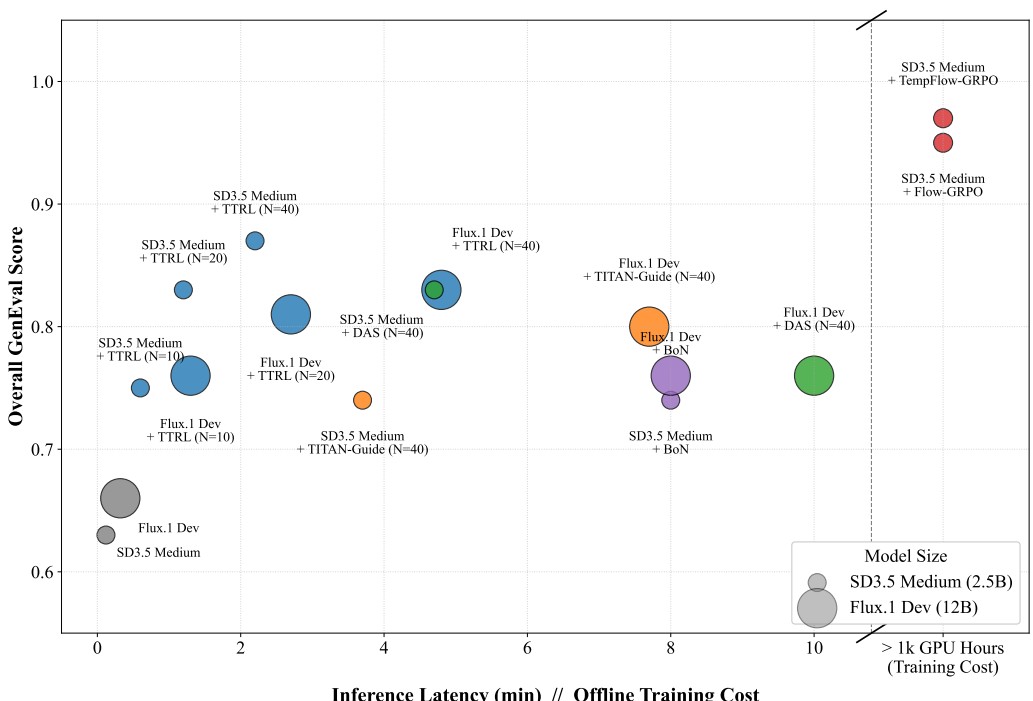

*Figure 10.* **Computation efficiency comparison on GenEval score.** The dashed line separates inference latency (left) from fine-tuning training cost (right). Flow-TTRL (blue) achieves a superior trade-off compared to the computationally intensive Best-of-N (purple, fixed at 8 min) and bridges the gap towards expensive fine-tuning methods.

**Efficiency Analysis.** To evaluate the computational efficiency of Flow-TTRL and provide insights for future work, Tables 4, 5, 6 present a detailed wall-clock breakdown (in seconds, summed over all steps) averaged over 100 random prompts across various backbones. The latency is categorized into Rollout (SDE+ODE sampling), RM Eval (VAE decoding and reward model evaluation), Update (latent optimization), and Other (inherent pipeline overhead). Since the rollout branches are independent, they can be effectively parallelized when VRAM capacity permits, as indicated in the "+Batch" columns. Our analysis reveals that when hardware memory is sufficient, parallelization significantly reduces inference latency, demonstrating the scalability of Flow-TTRL and its potential for practical deployment. Unless otherwise specified, the settings follow Appendix B.1, conducted on a single NVIDIA A800 80G.

| Backbone | GPU | Rollout/+Batch | RM Eval/+Batch | Update | Other | Total/+Batch |
|---|---|---|---|---|---|---|
| SD 3.5-M+Flow-TTRL | A800 | 25.68 / 12.35 | 71.15 / 27.02 | 24.68 | 11.35 | 132.86 / 75.40 |
| | RTX 4090 | 22.47 / OOM | 56.36 / 24.12 | 27.15 | 8.89 | 114.87 / 82.63 |
| Flux.1 Dev+Flow-TTRL | A800 | 100.32 / 38.23 | 71.31 / 28.18 | 83.54 | 31.36 | 286.53 / 181.31 |
| | RTX 4090 | 93.19 / OOM | 48.72 / 19.80 | 82.67 | 25.71 | 250.29 / 221.37 |

*Table 4.* **Hardware Efficiency Comparison.** Detailed wall-clock breakdown (seconds) on NVIDIA A800 (80GB) and RTX 4090 (24G).

| Backbone | Steps | Rollout/+Batch | RM Eval/+Batch | Update | Other | Total/+Batch |
|---|---|---|---|---|---|---|
| SD 3.5-M+Flow-TTRL | 10 | 5.85 / 3.24 | 15.96 / 7.32 | 11.25 | 3.17 | 36.23 / 24.98 |
| | 20 | 12.37 / 7.08 | 37.69 / 13.26 | 18.81 | 5.11 | 73.98 / 44.26 |
| | 40 | 25.68 / 12.35 | 71.15 / 27.02 | 24.68 | 11.35 | 132.86 / 75.40 |
| Flux.1 Dev+Flow-TTRL | 10 | 16.51 / 8.96 | 17.15 / 7.14 | 34.49 | 10.01 | 78.16 / 60.60 |
| | 20 | 42.92 / 18.66 | 37.68 / 17.94 | 65.48 | 16.95 | 163.03 / 119.03 |
| | 40 | 100.32 / 38.23 | 71.31 / 28.18 | 83.54 | 31.36 | 286.53 / 181.31 |

*Table 5.* **Inference Steps Breakdown.** Efficiency comparison across different sampling steps.

| Backbone | Group | Rollout/+Batch | RM Eval/+Batch | Update | Other | Total/+Batch |
|---|---|---|---|---|---|---|
| SD 3.5-M+Flow-TTRL | 4 | 12.39 / 9.38 | 35.14 / 20.86 | 23.41 | 11.37 | 82.31 / 65.02 |
| | 8 | 25.68 / 12.35 | 71.15 / 27.02 | 24.68 | 11.35 | 132.86 / 75.40 |
| | 12 | 38.17 / 14.56 | 94.34 / 30.24 | 25.65 | 12.02 | 170.18 / 82.47 |
| Flux.1 Dev+Flow-TTRL | 4 | 51.79 / 31.67 | 34.76 / 23.55 | 81.78 | 30.71 | 199.04 / 167.71 |
| | 8 | 100.32 / 38.23 | 71.31 / 28.18 | 83.54 | 31.36 | 286.53 / 181.31 |
| | 12 | 123.42 / 43.55 | 90.26 / 30.32 | 84.62 | 31.19 | 329.49 / 189.68 |

*Table 6.* **Group Size Scaling.** Impact of the number of rollout branches on inference latency.

## D.2. Extended Experiments on PartiPrompts, Pick-a-Pic and DrawBench

To verify the model's performance regarding human preference alignment, aesthetic quality, and text-image alignment, we conducted evaluations on the PartiPrompts, Pick-a-Pic (Kirstain et al., 2023), and Drawbench (Saharia et al., 2022) datasets. Quantitative results (Table 7, Table 8, and Table 9) demonstrate that Flow-TTRL consistently outperforms both base models and the computation-heavy Best-of-N baseline under a 5-minute computational budget. Our method achieves a superior trade-off, effectively realizing a triple alignment of these objectives. Qualitative comparisons in Figure 11a, Figure 11b, and Figure 12 further confirm these findings. Visualizations demonstrate that Flow-TTRL produces images with superior visual fidelity and enhanced prompt adherence, effectively aligning with human aesthetic preferences.

| Model | Human Preference Alignment | | | Aesthetic Quality | T2I Alignment |
|---|---|---|---|---|---|
| | PickScore↑ | HPS v2↑ | ImageReward↑ | Aesthetic↑ | CLIP↑ |
| SD 3.5 Medium | 22.21 | 0.275 | 0.890 | 5.442 | 28.01 |
| Flux.1 Dev | 22.84 | 0.316 | 1.205 | 5.843 | 27.27 |
| SD 3.5 Medium + BoN (5min) | 22.50 | 0.289 | 1.112 | 5.532 | 28.84 |
| Flux.1 Dev + BoN (5min) | 23.02 | 0.318 | 1.308 | 5.960 | 27.98 |
| SD 3.5 Medium + Flow-TTRL | 22.69 | 0.301 | 1.365 | 5.541 | **29.27** |
| Flux.1 Dev + Flow-TTRL | **23.17** | **0.323** | **1.472** | **5.963** | 28.51 |

*Table 7.* **Evaluation results on PartiPrompts.** Comparison of Human Preference Alignment, Aesthetic Quality, and Semantic Alignment. Best results are shown in **bold**.

## D.3. KL Analysis

To investigate the regularization behavior of Flow-TTRL, we analyzed the KL losses during the optimization process. The experiments are conducted on the PartiPrompts dataset using ImageReward as the reward model. Although the KL constraint is inherent to the Flow-TTRL framework, the explicit KL loss is calculated over the 14 time steps of GRPO.

The evolution of KL loss for SD3.5 and Flux.1 Dev is illustrated in Figure 13. The initial non-zero KL value at $t = 0$ originates from the optimization of the latent variable by the PRDP mechanism. As the Flow-TTRL process advances, we observe a consistent upward trend in KL loss, indicating that the generated latents are progressively deviating from the reference latents. This deviation confirms the effectiveness of the search process, as the model explores the latent space to generate images with distinct styles that maximize the reward.

| Model | Human Preference Alignment | | | Aesthetic Quality | T2I Alignment |
|---|---|---|---|---|---|
| | PickScore↑ | HPS v2↑ | ImageReward↑ | Aesthetic↑ | CLIP↑ |
| SD 3.5 Medium | 21.89 | 0.285 | 0.727 | 5.631 | 26.92 |
| Flux.1 Dev | 22.35 | 0.316 | 0.929 | 6.086 | 26.02 |
| SD 3.5 Medium + BoN (5min) | 22.01 | 0.289 | 0.839 | 5.735 | 27.47 |
| Flux.1 Dev + BoN (5min) | 22.51 | 0.314 | 1.081 | 6.180 | 26.86 |
| SD 3.5 Medium + Flow-TTRL | 22.24 | 0.299 | 1.187 | 5.733 | **28.09** |
| Flux.1 Dev + Flow-TTRL | **22.67** | **0.322** | **1.340** | **6.185** | 27.37 |

*Table 8.* **Evaluation results on Pick-a-Pic.** Comparison of Human Preference Alignment, Aesthetic Quality, and Semantic Alignment. Best results are shown in **bold**.

| Model | Human Preference Alignment | | | Aesthetic Quality | T2I Alignment |
|---|---|---|---|---|---|
| | PickScore↑ | HPS v2↑ | ImageReward↑ | Aesthetic↑ | CLIP↑ |
| SD 3.5 Medium | 22.14 | 0.272 | 0.630 | 5.181 | 28.87 |
| Flux.1 Dev | 22.59 | 0.305 | 0.931 | 5.722 | 27.30 |
| SD 3.5 Medium + BoN (5min) | 22.27 | 0.275 | 0.757 | 5.199 | 29.82 |
| Flux.1 Dev + BoN (5min) | 22.77 | 0.308 | 1.048 | **5.812** | 28.03 |
| SD 3.5 Medium + Flow-TTRL | 22.50 | 0.288 | 1.082 | 5.257 | **30.12** |
| Flux.1 Dev + Flow-TTRL | **23.03** | **0.315** | **1.252** | 5.778 | 29.00 |

*Table 9.* **Evaluation results on Drawbench.** Comparison of Human Preference Alignment, Aesthetic Quality, and Semantic Alignment. Best results are shown in **bold**.

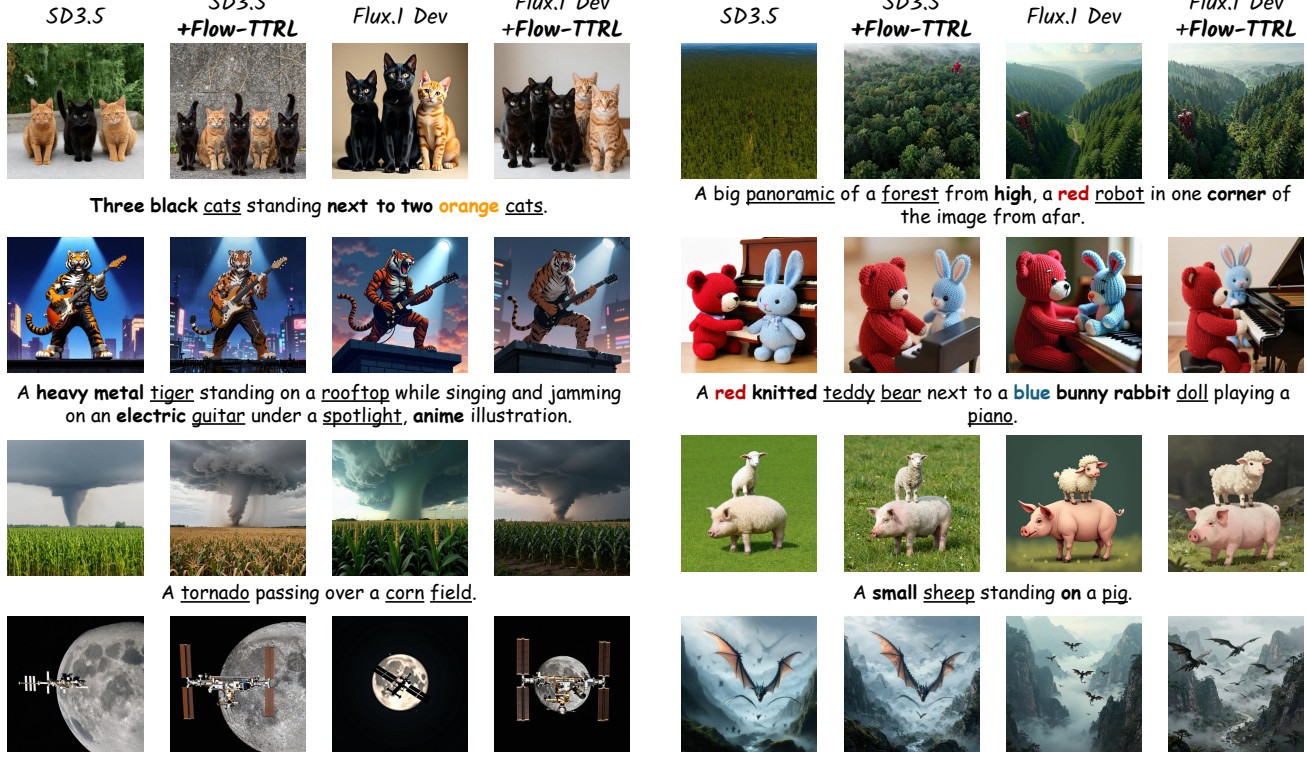

*(a)* **Qualitative comparison on PartiPrompts.** Flow-TTRL shows better prompt adherence and detail than base models.

*(b)* **Qualitative comparison on Pick-a-Pic.** Our method improves aesthetic quality and user preference alignment.

*Figure 11.* **Qualitative comparison results.** Visual examples from PartiPrompts (left) and Pick-a-Pic (right) benchmarks.

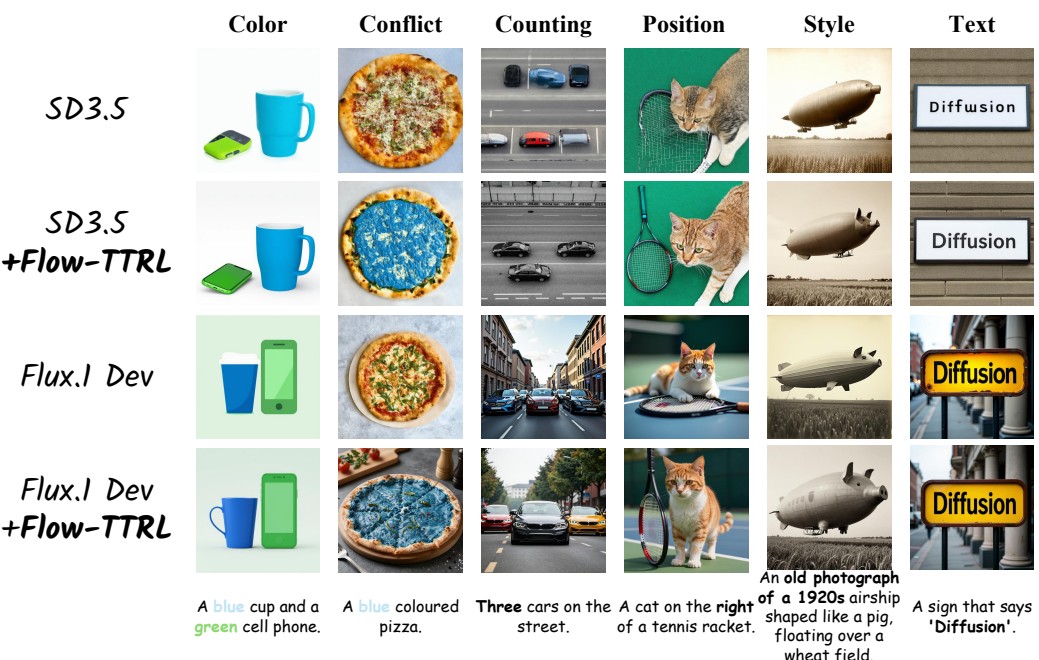

*Figure 12.* **Qualitative comparison on Drawbench.** Our method improves aesthetic quality and user preference alignment.

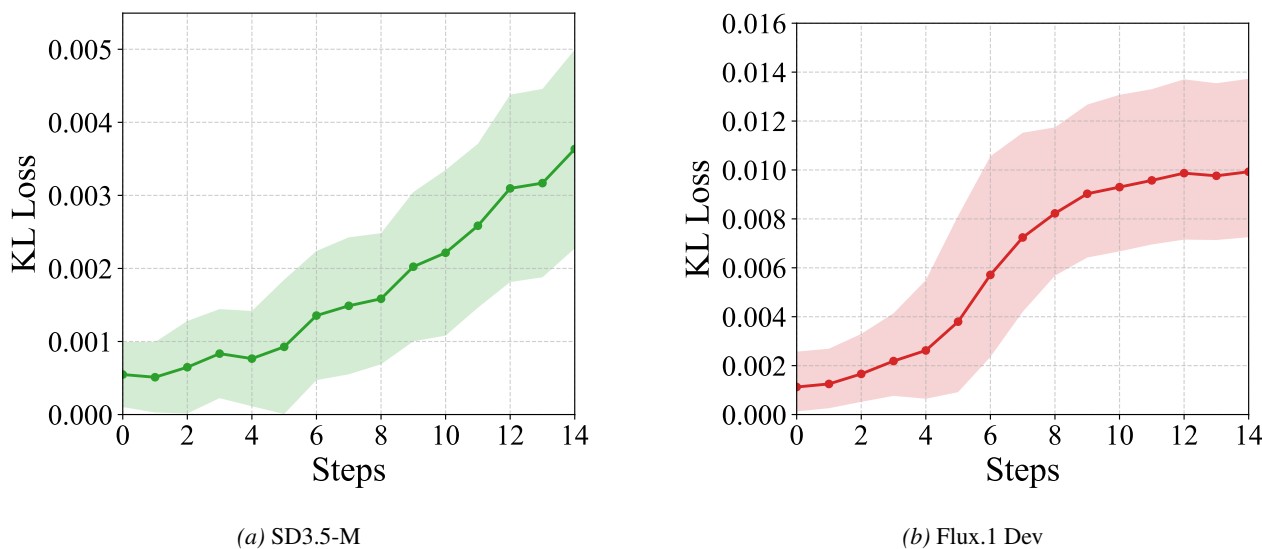

*(a)* SD3.5-M

*(b)* Flux.1 Dev

*Figure 13.* **Evolution of KL Loss during Flow-TTRL.** The value at $t = 0$ denotes the latent state following the completion of PRDP optimization. As the steps increase, the latent progressively deviates from the reference, indicating effective exploration for higher rewards. The widening variance and increasing KL values at later stages serve to counteract potential reward hacking.

Crucially, higher time steps correlate with more accurate reward estimation but also an increased risk of reward hacking. Consequently, the rising KL loss acts as a necessary counterforce to prevent excessive deviation and maintain generation quality. Comparing the two models, SD3.5 exhibits a relatively stable increase in KL loss. In contrast, Flux.1 Dev demonstrates a sharp surge in KL loss during specific mid-to-late time steps with significantly larger variance. Overall, Flux.1 Dev also exhibits a significantly larger KL loss magnitude compared to SD3.5-M. We attribute these distinct behavioral patterns and numerical scales to the inherent properties of the latent space and the generative priors of the respective base models. Despite these intrinsic differences, the KL constraint dynamically adapts to these characteristics, effectively suppressing instability and mitigating reward hacking.

### D.4. Extended Experiments on Verifiable Rewards

To further demonstrate the versatility and robustness of Flow-TTRL beyond learned human preference models, we conduct extended experiments on a suite of verifiable rewards derived from deterministic pixel statistics or objective algorithms. These include **Brightness** and **Darkness**, which are calculated based on the mean pixel intensity and its negative, respectively, to steer the model toward specific luminance levels. **Contrast** is measured via the variance of pixel intensities across the image to evaluate the optimization of dynamic range. We also utilize JPEG-based metrics as proxies for visual complexity: **JPEG Compressibility** (defined as the negative of the JPEG file size) incentivizes the generation of simpler patterns and smoother surfaces, while **JPEG Incompressibility** (direct file size) promotes high-frequency textures and information entropy. Finally, an **OCR Score** is employed to quantify prompt adherence in text rendering by measuring the overlap between the target prompt and text extracted from generated images via the PaddleOCR engine (Cui et al., 2025). Specifically, our evaluation is conducted on a curated test set comprising 100 randomly sampled prompts from PartiPrompts, Pick-a-Pic (Kirstain et al., 2023), and Drawbench (Saharia et al., 2022) datasets, while the OCR prompt is sourced from Flow-GRPO (Liu et al., 2025b). These verifiable benchmarks confirm that Flow-TTRL can effectively navigate diverse, non-linear, and even discrete reward landscapes while keeping the base model weights intact.

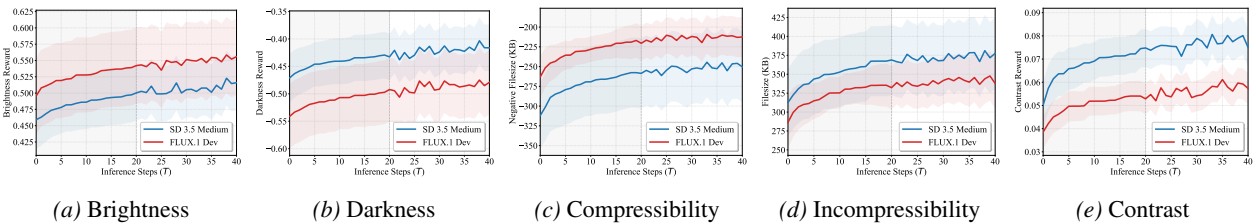

| *(a)* Brightness | *(b)* Darkness | *(c)* Compressibility | *(d)* Incompressibility | *(e)* Contrast |

*Figure 14.* **Quantitative reward optimization dynamics.** We plot the reward values across 40 inference steps for five deterministic objectives. The curves demonstrate the model's ability to maximize objective-specific rewards at test-time.

*Table 10.* **Quantitative comparison of OCR performance.** We report the OCR scores for base models and our Flow-TTRL framework. The best results are highlighted in **bold**.

| Metric | SD 3.5 | SD 3.5 + Flow-TTRL | Flux.1 Dev | Flux.1 Dev + Flow-TTRL | Flow-GRPO |
|---|---|---|---|---|---|
| OCR Score ↑ | 0.59 | 0.78 | 0.70 | 0.84 | **0.91** |

**Analysis of Statistical Verifiable Rewards.** We analyze the performance of Flow-TTRL on verifiable rewards through both quantitative reward curves (Figure 14) and qualitative visual samples (Figure 15). Specifically, we evaluate the model across five deterministic metrics: Brightness (Figures 14a, 15a), Darkness (Figures 14b, 15b), Compressibility (Figures 14c, 15c), Incompressibility (Figures 14d, 15d), and Contrast (Figures 14e, 15e). Our results reveal a consistent two-phase evolutionary pattern. During the initial 20 inference steps, the model demonstrates an exceptional ability to autonomously identify the optimal direction for reward maximization. This is evidenced by the rapid ascent in reward curves and a simultaneous, sharp visual transition toward the target features, such as the emergence of high-key luminance in brightness guidance or structural simplification in compressibility tasks. In the subsequent 20 steps, characterized by autonomous evolution, we observe more pronounced fluctuations in the quantitative trajectories. These oscillations are primarily attributed to the sensitive nature of ODE trajectories. Despite these local variances, the qualitative samples confirm that the model successfully resides within the high-reward regimes throughout the process. Furthermore, qualitative evidence reaffirms the indispensable role of the KL loss. It effectively prevents the model from converging toward degenerate solutions or image collapse, which are common when optimizing for non-semantic verifiable rewards. These results prove that Flow-TTRL can effectively search for and

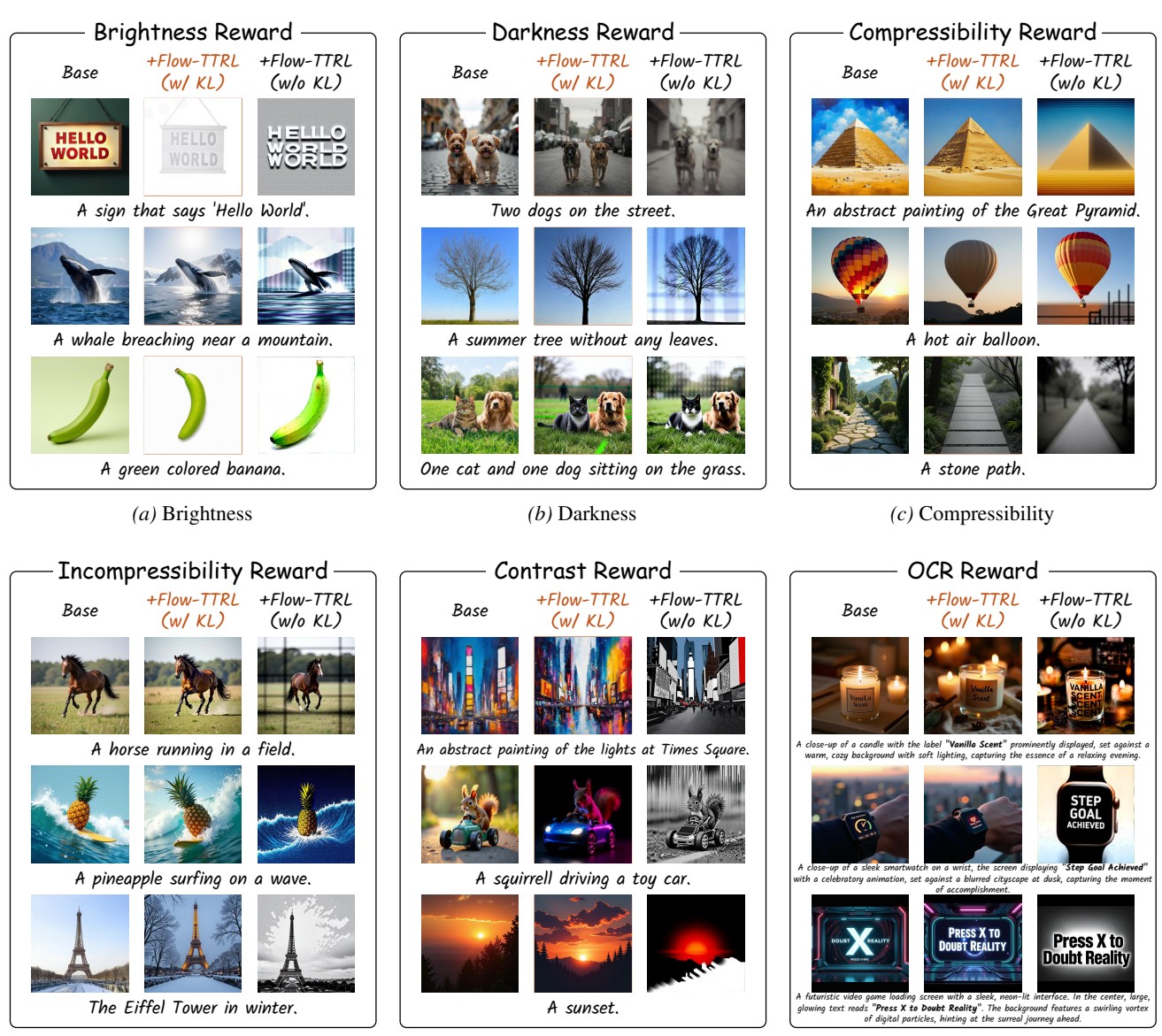

*Figure 15.* **Qualitative visualization of verifiable reward guidance.** Visual evidence showing the significant distribution shift induced by Flow-TTRL under different verifiable reward functions during inference.

maintain high-reward states for diverse, non-linear objectives at test-time without requiring any global weight modifications.

**Analysis of OCR Performance.** As demonstrated in Table 10, Flow-TTRL achieves significant improvements in OCR scores across both SD 3.5 and Flux.1 Dev models. To maintain visual integrity while optimizing for text legibility, we employ a hybrid reward objective that combines the OCR signal with HPS v2. This strategy is crucial for mitigating reward hacking, where the model might otherwise sacrifice global image structure to satisfy the sparse OCR metric. The necessity of the KL constraint is further highlighted in Figure 15f. We observe that removing this constraint leads to a catastrophic collapse in prompt alignment—a classic symptom of reward hacking where the latent optimization drifts into out-of-distribution regions that yield high rewards but nonsensical visuals. Despite these gains, a performance gap remains between our test-time approach and RL fine-tuning methods like Flow-GRPO. We attribute this to the inherent limitations of single-step SDE evolution in handling the complex, localized pixel rendering required for character synthesis. Generating legible text necessitates precise, large-area intensity adjustments within the latent space. We find that even with increased noise scales or relaxed KL constraints, the single-step SDE struggles to accurately estimate the trajectories required for correct character formation. We observe that in single-step SDE sampling, OCR rewards for different candidate latents often converge to a narrow range. This leads to an "averaged" reward assignment, where the reward advantage for potential character states is insufficiently distinctive. Consequently, the optimization process receives sparse heuristic guidance, making it challenging for the model to converge on the precise pixel configurations needed for high-fidelity text rendering.

### D.5. Effect of Iteration Numbers

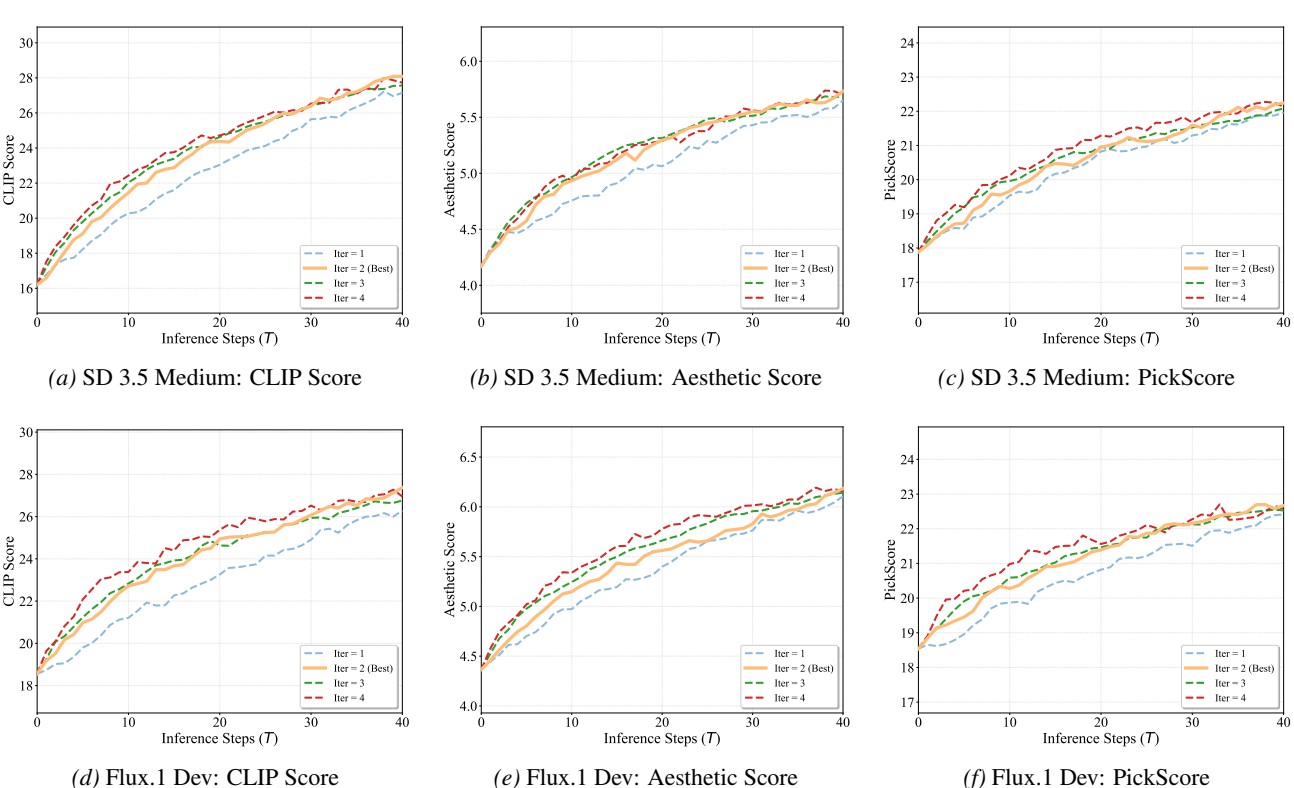

*(a)* SD 3.5 Medium: CLIP Score     *(b)* SD 3.5 Medium: Aesthetic Score     *(c)* SD 3.5 Medium: PickScore

*(d)* Flux.1 Dev: CLIP Score     *(e)* Flux.1 Dev: Aesthetic Score     *(f)* Flux.1 Dev: PickScore

*Figure 16.* **Study on the number of inner-loop iterations** ($K$). We evaluate the reward curves of CLIP Score, Aesthetic Score, and PickScore across different inference steps $T$. Results demonstrate that $K = 2$ consistently achieves the best balance between convergence speed and final performance, effectively avoiding the over-optimization issues observed in $K > 2$.

We investigate the sensitivity of Flow-TTRL to the inner-loop iteration count $K \in \{1, 2, 3, 4\}$. As illustrated in Figure 16, we can observe that: **1.Optimal Convergence with** $K = 2$**.** Across all evaluated metrics and base models, $K = 2$ (Proposed) consistently achieves the highest final rewards. This configuration provides the optimal balance: it offers sufficient gradient guidance to align the latent trajectory with human preferences while maintaining the structural integrity of the base generative model. In contrast, $K = 1$ exhibits clear under-optimization, failing to reach the potential of the reward landscape. **2. Over-optimization Trade-off in** $K > 2$**.** While $K > 2$ often demonstrates the fastest reward ascent

during the initial inference phase ($T < 15$), it subsequently suffers from the over-optimization or reward hacking effect. In later stages, the reward curves for $K > 2$ tend to saturate or experience slight oscillations, eventually falling below the $K = 2$ baseline. This suggests that excessive iterations may push the latent representations into out-of-distribution regions of the proxy reward models, leading to a marginal decline in both semantic alignment and aesthetic quality. Consequently, we fix $K = 2$ as the default setting for Flow-TTRL to ensure robust and superior alignment performance.

## D.6. Visualization of Different Stochasticity Levels

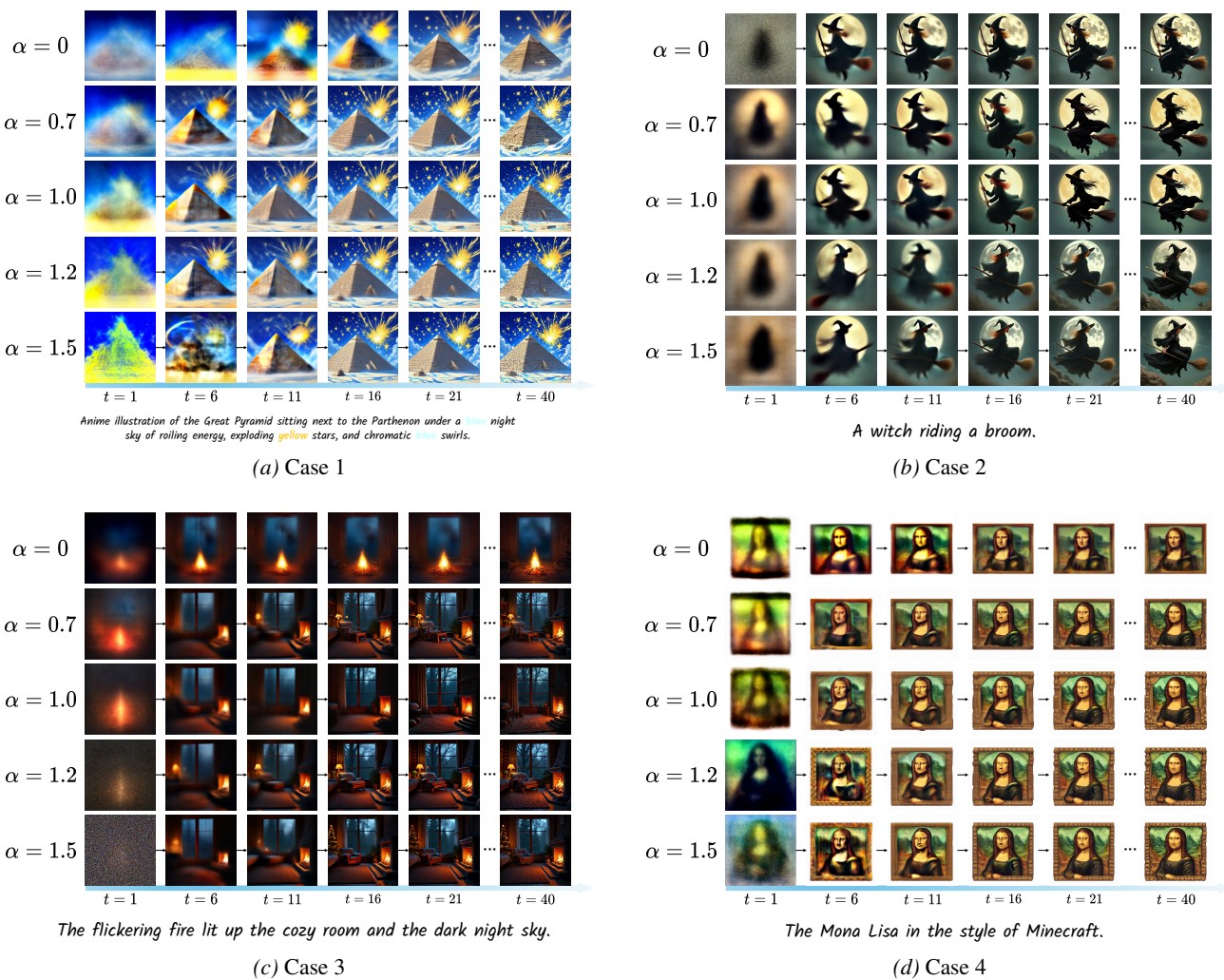

*Figure 17.* **Effect of stochastic noise level $\alpha$ on Flow-TTRL exploration space.** The baseline $\alpha = 0$ denotes the deterministic ODE. Increasing $\alpha$ enhances the stochasticity of the action search, enabling Flow-TTRL to escape sub-optimal trajectories and rectify semantic errors through reward-guided exploration.

Within the Flow-TTRL framework, the stochasticity coefficient $\alpha$ directly governs the breadth of the action search space during inference-time optimization. To isolate its influence, Figure 17 provides visualizations across four representative cases with $\alpha \in \{0, 0.7, 1.0, 1.2, 1.5\}$. The configuration $\alpha = 0$ represents the deterministic ODE baseline, which corresponds to a search space with zero stochasticity. As observed, this baseline often remains trapped in sub-optimal semantic basins, resulting in persistent alignment failures. By increasing $\alpha$, Flow-TTRL extends its exploration manifold, providing sufficient stochastic perturbation to decouple latent trajectories from biased ODE priors and steer them toward high-reward regimes. These static snapshots demonstrate that the stochastic component is instrumental in escaping local optima and rectifying complex compositional errors through reward-guided exploration. Consequently, our primary implementation employs a dynamic schedule that decays $\alpha$ from $1.5$ to $0.5$. This strategy yields superior results as the high initial stochasticity facilitates large-scale semantic rectification in the early stages, while the lower noise levels toward the end of the trajectory

allow for fine-grained aesthetic refinement and structural stabilization.

### D.7. Effect of Dynamic Stochasticity Scheduling Strategies

To investigate the influence of the stochasticity schedule on semantic alignment, we evaluate different decay strategies for the parameter $\alpha$ in Flow-TTRL. $\alpha$ controls the transition from broad exploration (high $\alpha$ in early stages) to fine-grained structural exploitation (low $\alpha$ in late stages). In addition to the default Linear Decay schedule, we implement a Cosine Decay schedule to provide a smoother transition, which maintains higher exploration stochasticity during the intermediate steps. Table 11 compares these two strategies across GenEval and T2I-CompBench (Complex) using Flux.1 Dev and SD 3.5-M backbones. The results indicate that while both scheduling strategies significantly outperform the base models, the Cosine Decay schedule consistently yields further improvements in complex semantic alignment, particularly for the Flux.1 Dev backbone. This suggests that a more expansive exploration phase in the mid-stage helps the model better escape local optima in semantic layout generation.

| Methods | GenEval (Overall)↑ | T2I-CompBench (Complex)↑ |
|---|---|---|
| SD 3.5 Medium | 0.63 | 0.3542 |
| Flux.1 Dev | 0.66 | 0.3703 |
| SD 3.5 Medium + Flow-TTRL (Linear) | **0.87** | 0.4045 |
| Flux.1 Dev + Flow-TTRL (Linear) | 0.83 | 0.4179 |
| SD 3.5 Medium + Flow-TTRL (Cosine) | **0.87** | 0.4051 |
| Flux.1 Dev + Flow-TTRL (Cosine) | 0.85 | **0.4207** |

*Table 11.* **Study on dynamic $\alpha$ scheduling.** Comparison of Linear and Cosine Decay strategies on GenEval and T2I-CompBench. Best results are shown in **bold**.

### D.8. Effect of Latent Optimization

To ensure a rigorous and fair evaluation, we first compare Flow-TTRL against latent selection methods (specifically Best-of-N and Beam Search) under a comparable inference time budget. As demonstrated in Table 12, we deliberately grant the baseline methods slightly more execution time to provide a conservative comparison. Under these constraints, Flow-TTRL significantly outperforms prior-restricted search methods, particularly in compositional reasoning (GenEval, Complex) and structural accuracy (OCR) (shown in Figure 18). The performance gap stems from fundamental differences in search strategies: latent selection is restricted to greedy searching within a pre-trained prior, whereas Flow-TTRL performs continuous heuristic optimization within the latent space to access statistically rare, high-reward regions. Furthermore, while the baselines frequently suffer from reward hacking (e.g., unnatural lighting and distorted structures) without rigid $L_2$ (w/o $L_2$) regularization, adding such constraints (w/ $L_2$) severely bottlenecks their alignment capabilities. In contrast, the MDP formulation of Flow-TTRL inherently accommodates KL-divergence constraints, gracefully preventing reward hacking while leveraging in-context guidance from future feedback to achieve superior visual aesthetics and alignment. Beyond execution time, we extend our analysis to an equal-NFE setting, directly comparing Flow-TTRL against the Base models and Best-of-N ($N = 8$) across the fine-grained sub-categories of GenEval and T2I-CompBench. As detailed in Table 13 and Table 14, Flow-TTRL consistently dominates across almost all evaluated dimensions, including counting, spatial positioning, color attribution, and texture alignment (as visualized in Figure 19 and Figure 20). This comparison confirms that mere latent selection provides only marginal performance gains; active latent optimization through sampling is indispensable for robustly handling complex and structured prompts. By seamlessly integrating in-context RL into a KL-regularized MDP, Flow-TTRL comprehensively perceives the reward landscape, effectively unlocking a superior trade-off between prompt alignment and generation speed.

### D.9. Multiple Reward Analysis

We investigate the individual contributions of the HPS v2 and CLIP Score rewards to the performance of Flow-TTRL, concluding that their integration is essential for achieving balanced generative quality. As demonstrated by the quantitative results in Table 15 and the visual evidence in Figure 21, optimizing for a single reward signal frequently triggers the reward hacking phenomenon. For instance, over-reliance on HPS v2 often drives the model toward unnaturally oversaturated color profiles and artificial textures, while CLIP-centric optimization tends to force the explicit rendering of specific textual tokens

| Backbone | Method | Time (s) | GenEval↑ | Complex↑ | PickScore↑ | HPSv2↑ | ImageReward↑ | OCR↑ |
|---|---|---|---|---|---|---|---|---|
| SD 3.5-M | Best-of-N ($w/o\ L_2$) | 132.16 | 0.73 | 0.3683 | 22.04 | **0.299** | 0.932 | 0.62 |
| | Best-of-N ($w/\ L_2$) | 134.84 | 0.73 | 0.3702 | 21.88 | 0.292 | 1.066 | 0.62 |
| | Beam Search ($w/o\ L_2$) | 138.39 | 0.73 | 0.3674 | 21.99 | 0.295 | 0.879 | 0.61 |
| | Beam Search ($w/\ L_2$) | 141.81 | 0.72 | 0.3718 | 22.07 | 0.293 | 1.142 | 0.62 |
| | **Flow-TTRL** | 130.48 | **0.87** | **0.4045** | **22.24** | **0.299** | **1.187** | **0.78** |
| Flux.1 Dev | Best-of-N ($w/o\ L_2$) | 301.67 | 0.70 | 0.3821 | 22.59 | 0.323 | 1.191 | 0.72 |
| | Best-of-N ($w/\ L_2$) | 303.92 | 0.74 | 0.3847 | 22.48 | **0.324** | 1.315 | 0.73 |
| | Beam Search ($w/o\ L_2$) | 324.28 | 0.72 | 0.3869 | 22.70 | 0.322 | 1.287 | 0.71 |
| | Beam Search ($w/\ L_2$) | 328.60 | 0.73 | 0.3843 | **22.73** | **0.324** | 1.154 | 0.72 |
| | **Flow-TTRL** | 285.97 | **0.83** | **0.4179** | 22.67 | 0.322 | **1.340** | **0.84** |

*Table 12.* **Similar Time Comparison.** Quantitative evaluation of Flow-TTRL against latent selection baselines under similar execution time budgets. Best results are shown in **bold**.

| Backbone | Method | Overall↑ | Single↑ | Two↑ | Counting↑ | Colors↑ | Position↑ | Color Attrib.↑ |
|---|---|---|---|---|---|---|---|---|
| SD 3.5-M | Base | 0.63 | 0.98 | 0.78 | 0.50 | 0.81 | 0.24 | 0.52 |
| | Best-of-N ($w/o\ L_2$) | 0.71 | 0.98 | 0.89 | 0.70 | 0.80 | 0.31 | 0.57 |
| | Best-of-N ($w\ L_2$) | 0.70 | 0.98 | 0.91 | 0.63 | 0.83 | 0.28 | 0.55 |
| | **Flow-TTRL** | **0.87** | **1.00** | **0.99** | **0.95** | **0.90** | **0.55** | **0.85** |
| Flux.1 Dev | Base | 0.66 | 0.98 | 0.81 | 0.74 | 0.79 | 0.22 | 0.45 |
| | Best-of-N ($w/o\ L_2$) | 0.69 | 0.98 | 0.84 | 0.77 | 0.79 | 0.23 | 0.53 |
| | Best-of-N ($w\ L_2$) | 0.70 | 0.98 | 0.90 | 0.79 | 0.84 | 0.22 | 0.49 |
| | **Flow-TTRL** | **0.83** | **1.00** | **0.97** | **0.94** | **0.90** | **0.41** | **0.77** |

*Table 13.* **Equal NFE Comparison on GenEval.** Quantitative results comparing Flow-TTRL against base models and Best-of-N ($N = 8$) across fine-grained categories. Best results within the selected methods are shown in **bold**.

| Backbone | Method | Color↑ | Shape↑ | Texture↑ | 2D-Spatial↑ | Non-Spatial↑ | Complex↑ |
|---|---|---|---|---|---|---|---|
| SD 3.5-M | Base | 0.7994 | 0.5669 | 0.7338 | 0.2850 | 0.3146 | 0.3542 |
| | Best-of-N ($w/o\ L_2$) | 0.8127 | 0.5691 | 0.7459 | 0.2912 | 0.3107 | 0.3671 |
| | Best-of-N ($w\ L_2$) | 0.8113 | 0.5744 | 0.7396 | 0.2977 | 0.3152 | 0.3697 |
| | **Flow-TTRL** | **0.9042** | **0.7361** | **0.8261** | **0.4414** | **0.3319** | **0.4045** |
| Flux.1 Dev | Base | 0.7407 | 0.5718 | 0.6922 | 0.2863 | 0.3127 | 0.3703 |
| | Best-of-N ($w/o\ L_2$) | 0.7556 | 0.5896 | 0.7068 | 0.2924 | 0.3101 | 0.3798 |
| | Best-of-N ($w\ L_2$) | 0.7682 | 0.5832 | 0.7015 | 0.2899 | 0.3115 | 0.3784 |
| | **Flow-TTRL** | **0.8804** | **0.6717** | **0.7958** | **0.4390** | **0.3229** | **0.4179** |

*Table 14.* **Equal NFE Comparison on T2I-CompBench.** Quantitative results comparing Flow-TTRL against base models and Best-of-N ($N = 8$) across structural constraints. Best results within the selected methods are shown in **bold**.

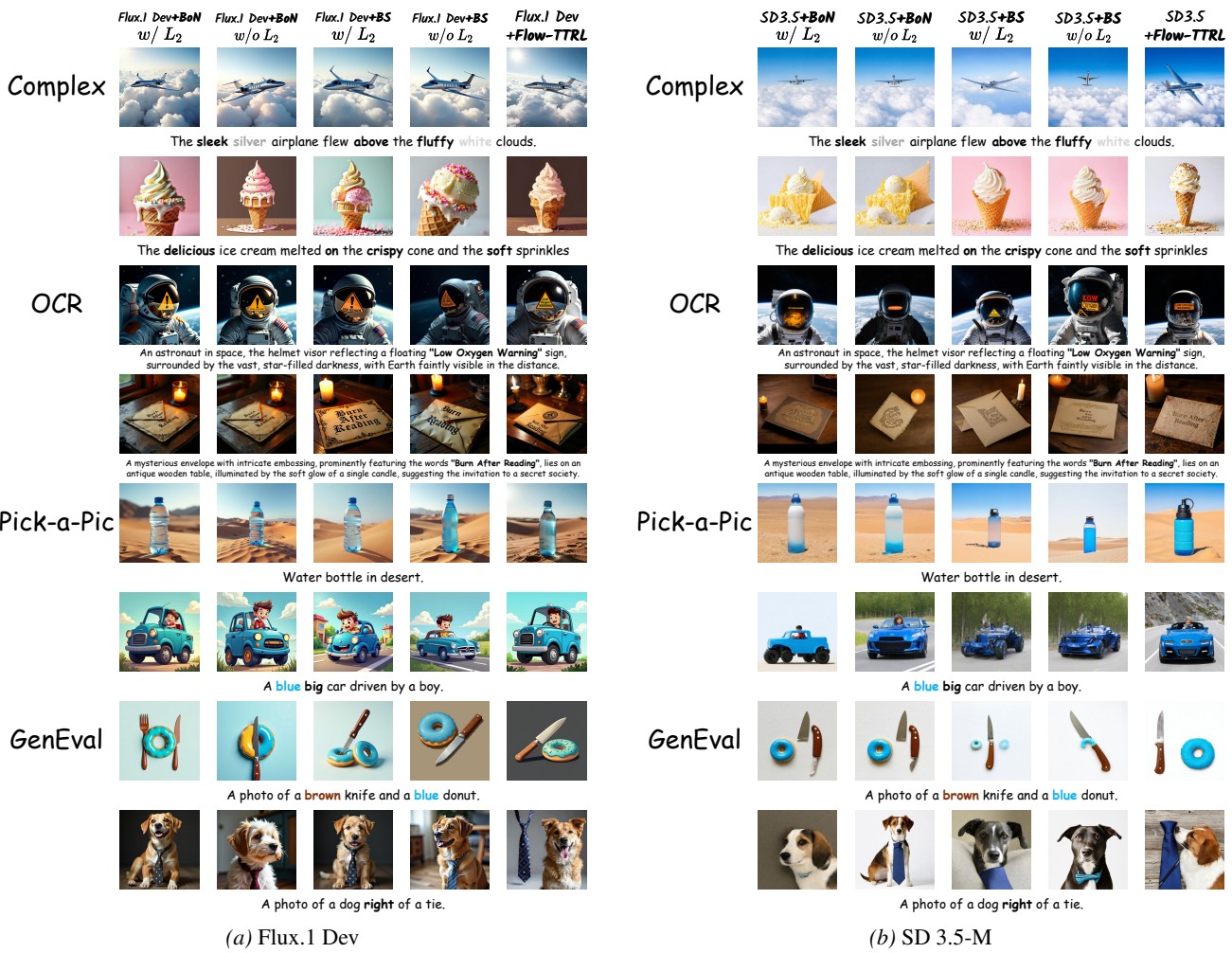

*(a)* Flux.1 Dev                    *(b)* SD 3.5-M

*Figure 18.* **Comparison against latent selection methods.** Flow-TTRL demonstrates superior alignment compared to Best-of-N and Beam Search under comparable time budgets.

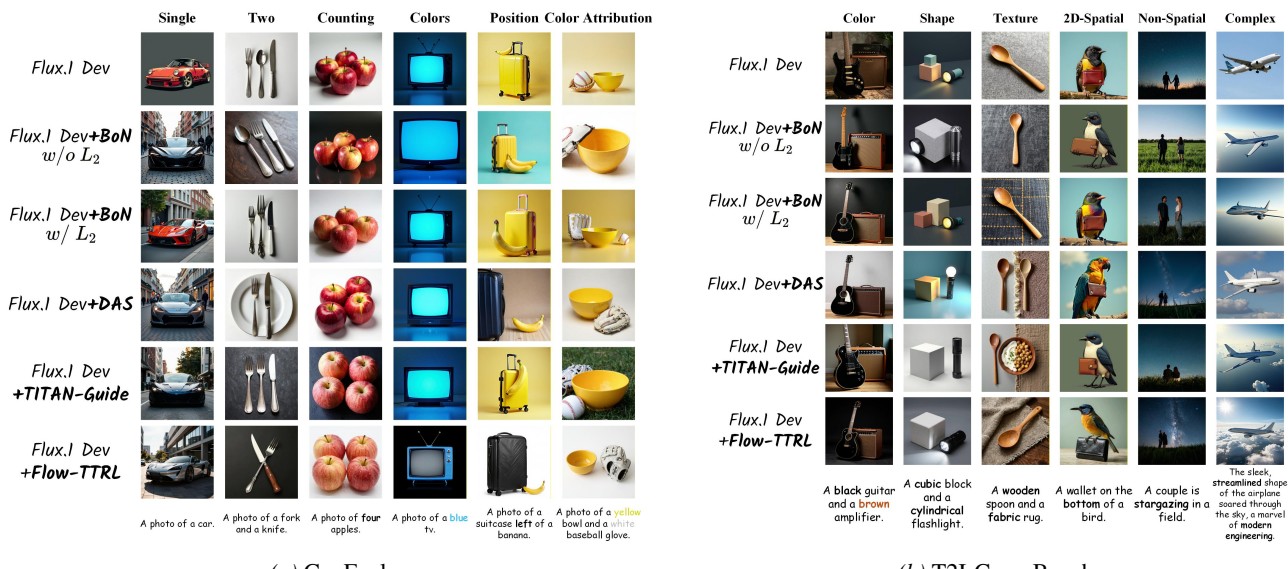

*(a)* GenEval          *(b)* T2I-CompBench

*Figure 19.* **Comparison against test-time alignment methods on Flux.1 Dev.** Flow-TTRL achieves more precise semantic alignment under equal NFE.

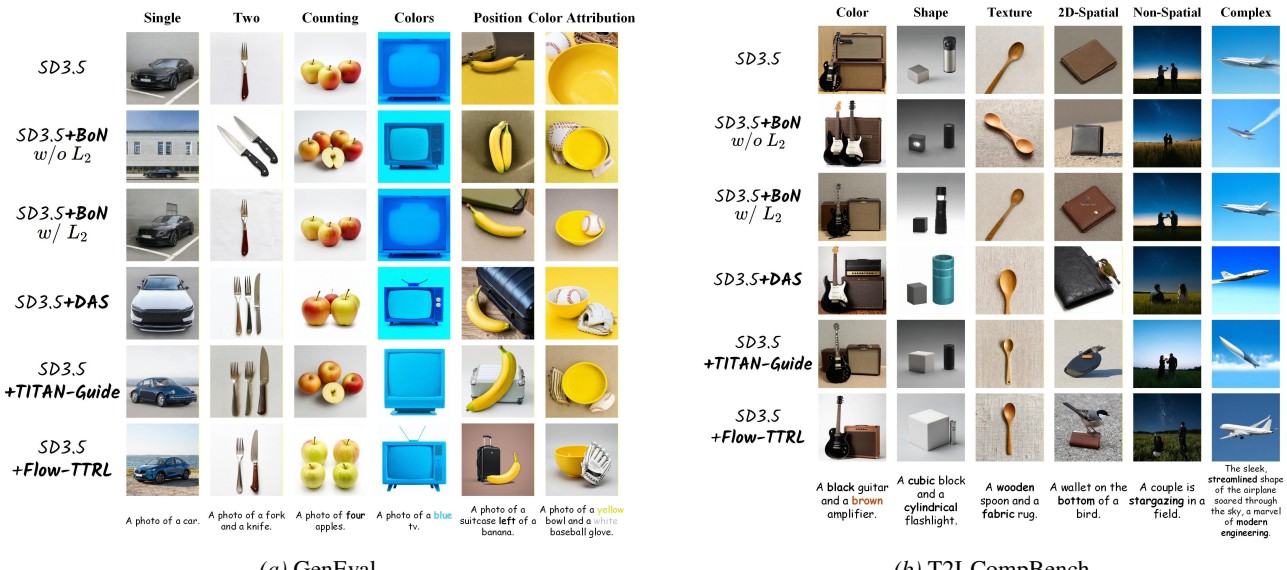

*(a)* GenEval          *(b)* T2I-CompBench

*Figure 20.* **Comparison against test-time alignment methods on SD 3.5 Medium.** Flow-TTRL achieves more precise semantic alignment under equal NFE.

at the expense of global compositional harmony and layout coherence. Beyond simple combination, Flow-TTRL enables a plug-and-play fusion of diverse reward signals. As illustrated in Figure 22, our framework allows for seamless interpolation between various reward pairs—such as HPS, ImageReward (IR), and PickScore (PS) with Aesthetic (AES) or CLIP scores. This flexibility provides a powerful mechanism for personalized generation, where users can dynamically calibrate the mixture ratios of different semantic and aesthetic priors to align with specific preferences without any parameter updates. Ultimately, our framework harnesses the synergy of multiple rewards to stabilize the latent search process, effectively mitigating the trade-offs associated with isolated reward signals. This multi-reward configuration ensures that the resulting scientific illustrations maintain high aesthetic refinement while strictly adhering to complex textual constraints, proving that diverse, multi-objective guidance is critical for robust test-time optimization.

| Model | Human Preference Alignment | | | Aesthetic Quality | T2I Alignment |
|---|---|---|---|---|---|
| | PickScore↑ | HPS v2↑ | ImageReward↑ | Aesthetic↑ | CLIP↑ |
| SD 3.5 Medium | 22.14 | 0.272 | 0.630 | 5.181 | 28.87 |
| + Flow-TTRL | **22.50** | **0.288** | **1.082** | 5.257 | 30.12 |
| + Flow-TTRL w/o HPS v2 | 22.17 | 0.268 | 0.723 | 5.093 | **30.59** |
| + Flow-TTRL w/o CLIPScore | 22.39 | 0.287 | 0.981 | **5.347** | 28.91 |
| Flux.1 Dev | 22.59 | 0.305 | 0.931 | 5.722 | 27.30 |
| + Flow-TTRL | **23.03** | **0.315** | 1.252 | **5.778** | **29.00** |
| + Flow-TTRL w/o HPS v2 | 22.62 | 0.304 | 0.912 | 5.683 | 28.83 |
| + Flow-TTRL w/o CLIPScore | 22.83 | 0.314 | **1.325** | 5.707 | 27.43 |

*Table 15.* **Ablation study of reward components on Drawbench.** We compare the baseline performance against our full Flow-TTRL framework and its variants by selectively removing HPS v2 and CLIPScore rewards.

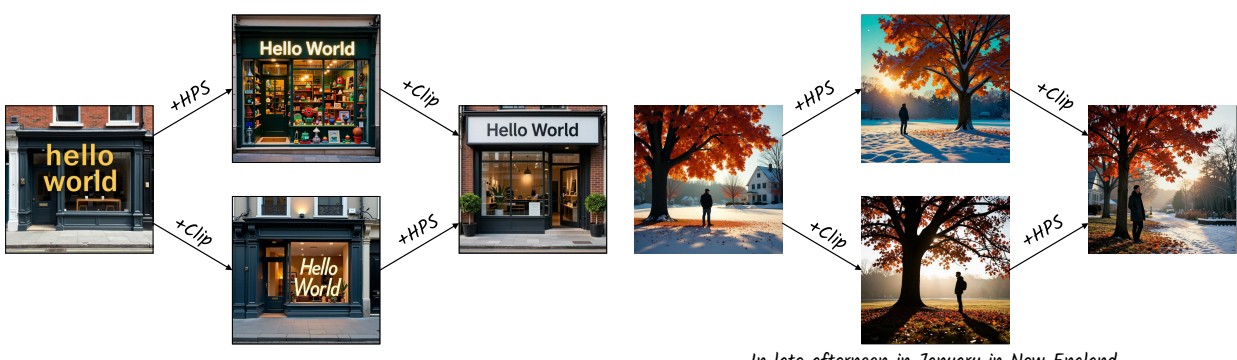

A storefront with '**Hello World**' written on it.

In late afternoon in January in New England, a man stands in the shadow of a maple tree.

*Figure 21.* **Qualitative evidence of reward balancing.** Visual comparison of generations guided by different reward combinations. The full Flow-TTRL (multi-reward) demonstrates superior balance, whereas single-reward versions often exhibit signs of reward hacking.

## D.10. Visualization of Attention Patterns

This section provides qualitative evidence of how reward-driven optimization reshapes the interplay between textual prompts and latent representations. To investigate the impact of the reward model on cross-modal alignment, we visualize the interaction attention maps at the final convergence state ($T = 40$). Rather than relying on the base ODE prior, our framework utilizes reward signals to modulate the coupling between prompt tokens and latent features, ensuring that the final latent state accurately reflects high-reward objectives. For both SD 3.5 and Flux.1 Dev, these interaction scores are averaged across all attention heads to provide a robust representation of the resulting cross-modal correspondence. By isolating the interactive components within the joint attention architectures, we demonstrate that reward guidance effectively reallocates attention toward previously neglected linguistic concepts. The final optimized latents thus achieve a state of enhanced semantic fidelity, where latent features are precisely steered to align with the intricate constraints of the input prompt. As demonstrated by the six cases in Figure 23, vanilla SD3.5 and Flux.1 Dev models often suffer from semantic leakage or suboptimal rendering quality. By optimizing the latent space via reward functions, our Flow-TTRL not only generates

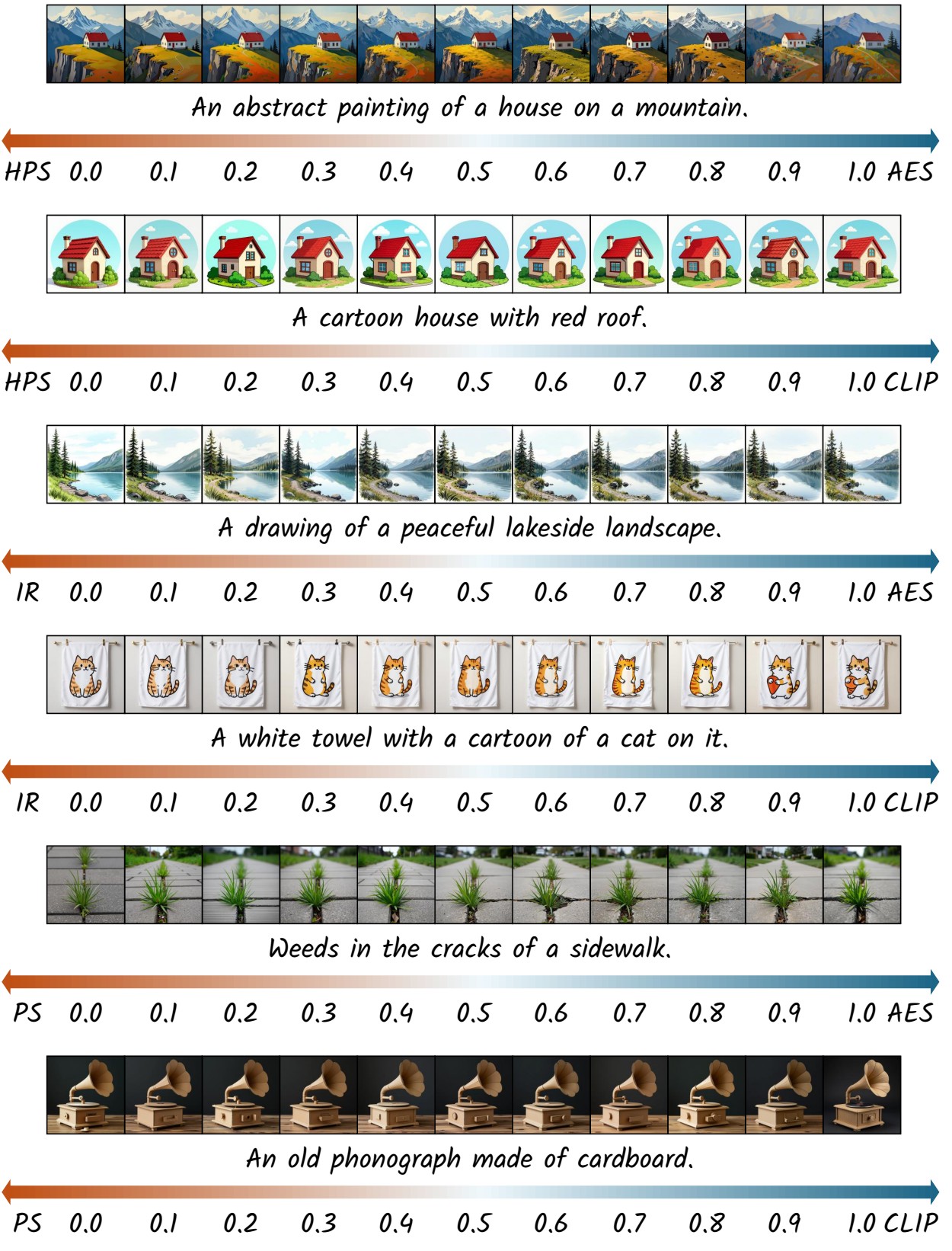

*Figure 22.* **Qualitative results of multi-reward interpolation.** Each row demonstrates a smooth transition between two distinct reward objectives. The numerical labels indicate the interpolation ratio; for instance, a value of 0.4 represents a weighted combination of 0.4 from the left-most reward—such as **HPS**, **ImageReward (IR)**, or **PickScore (PS)**—and 0.6 from the right-most reward, typically **Aesthetic Score (AES)** or **CLIP Score**. This illustrates the framework's plug-and-play flexibility in balancing semantic alignment and aesthetic quality.

images that exhibit superior alignment with both textual prompts and human preferences but also yields significantly clearer attention correspondences between linguistic tokens and latent representations.

### D.11. Additional Qualitative Comparison Results

To further validate the performance of our framework, we present additional qualitative comparisons in this section. The provided examples are primarily sourced from the T2I-CompBench benchmark, with a particular focus on its "Complex" category. As demonstrated in Figure 24a, Flow-TTRL consistently achieves superior generative quality compared to the base models on standard benchmarks. Furthermore, in the more challenging scenarios involving intricate attribute binding and spatial relationships (Figure 24b), Flow-TTRL demonstrates a robust ability to maintain semantic alignment where base models often fail. These visual improvements serve as strong evidence that our test-time latent optimization effectively navigates the generative distribution toward high-preference regions without the need for parameter fine-tuning.

### D.12. Failure Cases and Calibration

It is a well-recognized challenge in the test-time optimization that different prompts often require distinct optimal hyperparameter configurations to achieve the best results (Chefer et al., 2023). Otherwise, it can lead to noticeable performance degradation or failure cases as qualitatively illustrated in Figure 25. We observe a similar phenomenon in Flow-TTRL. Although our default settings (i.e., $\beta = 0.00015, \eta = 750$) are designed to be conservative and provide a robust baseline across diverse datasets, they may not consistently yield the "theoretical peak" performance for every individual prompt. As shown in Figure 25a, on certain samples, the default KL coefficient $\beta$ may be too small to sufficiently constrain the latent search within the model's prior distribution, triggering reward hacking and a loss of generative priors that manifest as unnatural artifacts. While on some other samples, the default $\beta$ may be too high, imposing a constraint so rigid that the reward signal is hindered from effectively guiding the generation trajectory or achieving prompt alignment. Furthermore, as illustrated in Figure 25b, on certain samples, the default learning rate $\eta$ may be too small to achieve sufficient optimization or escape local optima, resulting in persistent artifacts and blurry textures. While on some other samples, the default $\eta$ may be too high, causing structural instability and jagged edges that disrupt the global compositional harmony. Crucially, because Flow-TTRL operates entirely at inference time without requiring parameter updates, these failure modes are not permanent deficiencies. Instead, they can be efficiently rectified via manual hyperparameter calibration, allowing the model to recover high-quality, aligned results within a single inference pass.

## E. Limitations

Despite its benefits, Flow-TTRL faces several limitations s similar to all other test-time alignment methods: (1) It requires additional inference-time computational overhead primarily from the iterative DiT forward passes, VAE decoding, and reward scoring processes. (2) Despite the empirical stability of our default hyperparameter settings, achieving optimal performance often requires manual calibration of hyperparameters, since the best settings for test-time learning are often prompt-specific. (3) The method remains dependent on the quality of reward models, which means it does not fundamentally eliminate reward hacking if the underlying models have inherent biases or perceptual flaws.

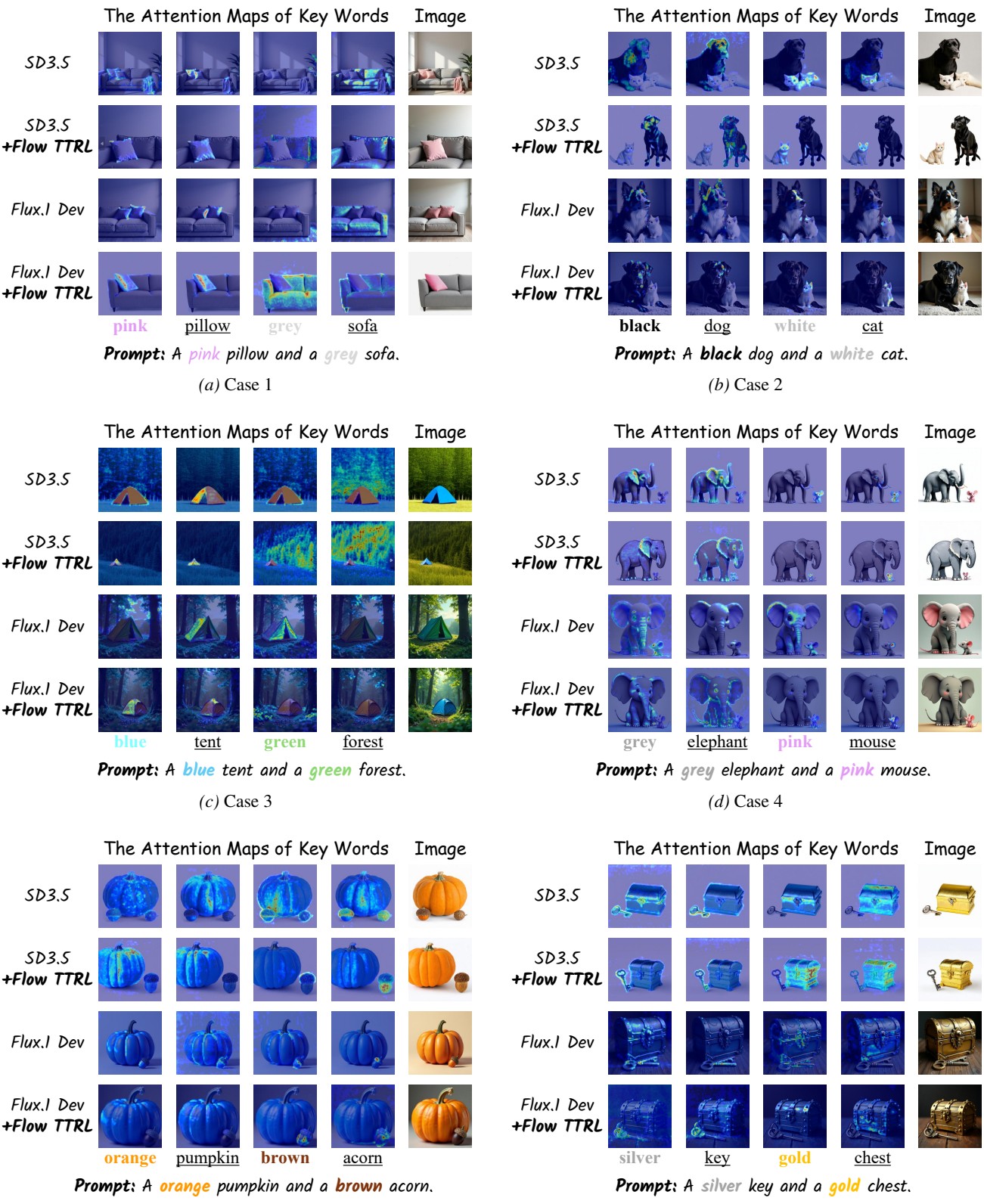

*Figure 23.* Visualization of interactive attention maps across six cases.

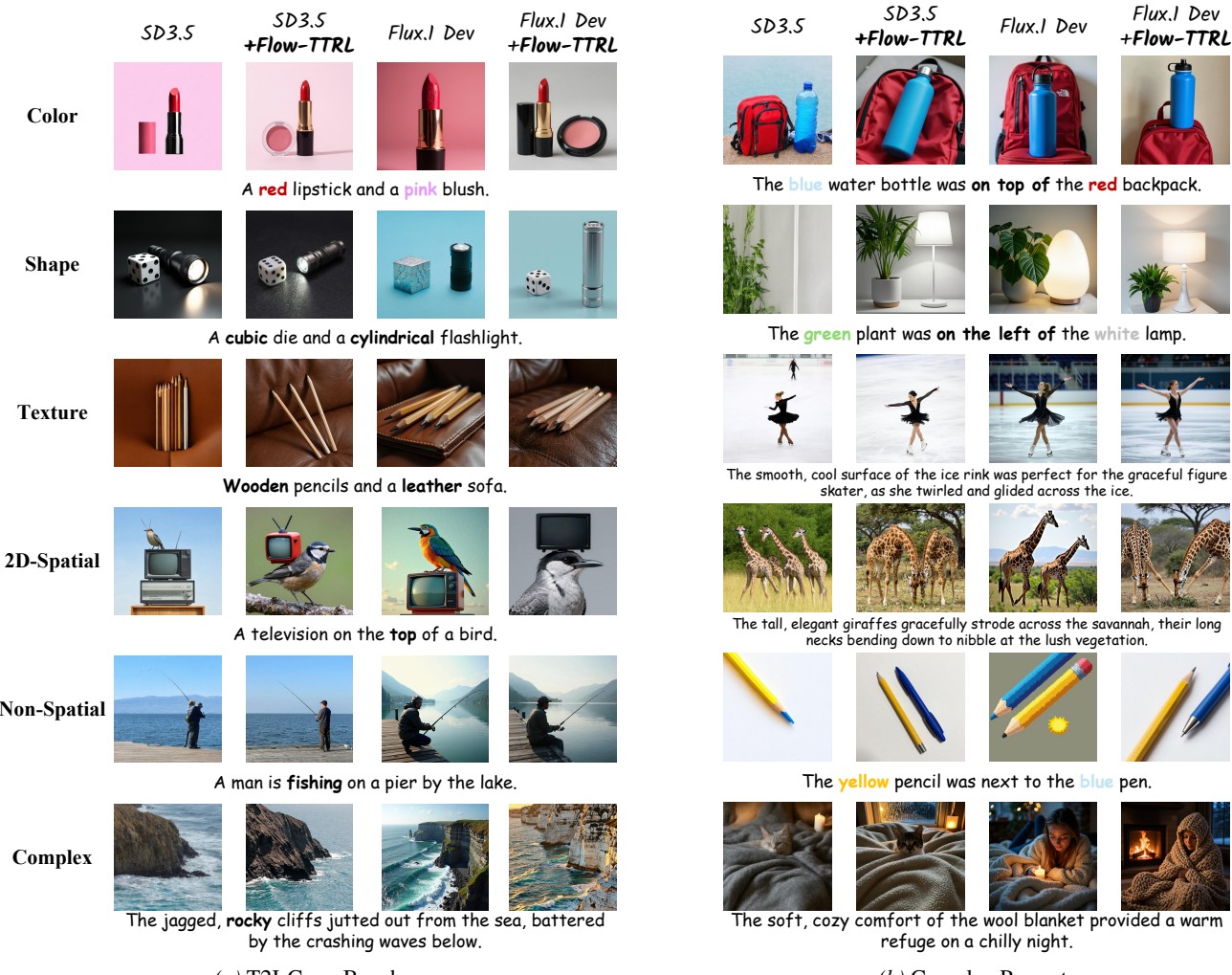

*(a)* T2I-CompBench                    *(b)* Complex Prompts

*Figure 24.* **Extended qualitative comparisons.** Qualitative results on (a) the T2I-CompBench dataset and (b) complex prompts. Flow-TTRL consistently demonstrates higher aesthetic appeal and more precise semantic alignment compared to base models by effectively navigating the latent distribution at test-time.

## Appendix References

Bertsekas, D. P. Nonlinear programming. *Journal of the Operational Research Society*, 48(3):334–334, 1997.

Chefer, H., Alaluf, Y., Vinker, Y., Wolf, L., and Cohen-Or, D. Attend-and-excite: Attention-based semantic guidance for text-to-image diffusion models. *ACM transactions on Graphics (TOG)*, 42(4):1–10, 2023.

Cui, C., Sun, T., Lin, M., Gao, T., Zhang, Y., Liu, J., Wang, X., Zhang, Z., Zhou, C., Liu, H., et al. Paddleocr 3.0 technical report. *arXiv preprint arXiv:2507.05595*, 2025.

Kirstain, Y., Polyak, A., Singer, U., Matiana, S., Penna, J., and Levy, O. Pick-a-pic: An open dataset of user preferences for text-to-image generation. *Advances in neural information processing systems*, 36:36652–36663, 2023.

Na, B., Park, M., Sim, G., Shin, D., Bae, H., Kang, M., Kwon, S. J., Kang, W., and Moon, I.-C. Diffusion adaptive text embedding for text-to-image diffusion models. *arXiv e-prints*, pp. arXiv–2510, 2025.

Radford, A., Kim, J. W., Hallacy, C., Ramesh, A., Goh, G., Agarwal, S., Sastry, G., Askell, A., Mishkin, P., Clark, J., et al. Learning transferable visual models from natural language supervision. In *International conference on machine learning*, pp. 8748–8763. PmLR, 2021.

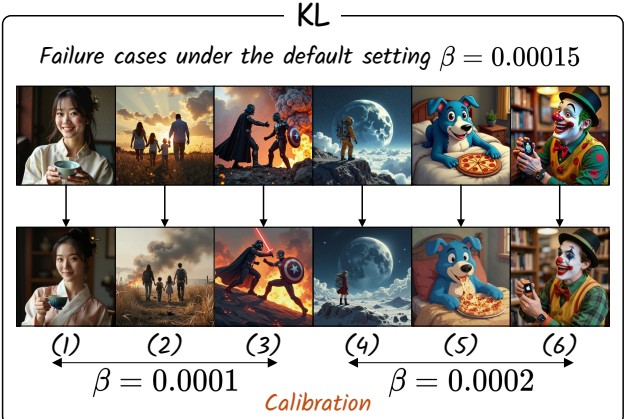

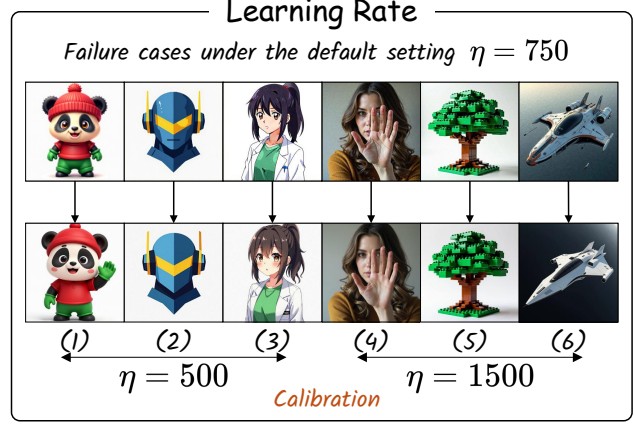

*(a)* **Calibration of KL coefficient** $\beta$. On certain samples, the default $\beta$ may be too small to prevent reward hacking and the loss of model priors. While on some other samples, the default $\beta$ may be too high so that the reward is hindered from effectively guiding alignment with the prompt.

*(b)* **Calibration of learning rate** $\eta$. On certain samples, the default $\eta$ may be too small to achieve sufficient optimization or escape local optima where persistent artifacts emerge. While on some other samples, the default $\eta$ may be too high, resulting in jagged edges and structural instability.

*Figure 25.* **Failure cases and their calibration.** Although our default hyperparameters are robust for most scenarios, specific edge cases can be rectified through test-time calibration. This process effectively resolves trade-offs between reward guidance and generative stability.

Saharia, C., Chan, W., Saxena, S., Li, L., Whang, J., Denton, E. L., Ghasemipour, K., Gontijo Lopes, R., Karagol Ayan, B., Salimans, T., et al. Photorealistic text-to-image diffusion models with deep language understanding. *Advances in neural information processing systems*, 35:36479–36494, 2022.

Tang, Z., Peng, J., Tang, J., Hong, M., Wang, F., and Chang, T.-H. Inference-time alignment of diffusion models with direct noise optimization. *arXiv preprint arXiv:2405.18881*, 2024.

Wu, X., Hao, Y., Sun, K., Chen, Y., Zhu, F., Zhao, R., and Li, H. Human preference score v2: A solid benchmark for evaluating human preferences of text-to-image synthesis. *arXiv preprint arXiv:2306.09341*, 2023.

Xu, J., Liu, X., Wu, Y., Tong, Y., Li, Q., Ding, M., Tang, J., and Dong, Y. Imagereward: Learning and evaluating human preferences for text-to-image generation. *Advances in Neural Information Processing Systems*, 36:15903–15935, 2023.

Xue, Z., Wu, J., Gao, Y., Kong, F., Zhu, L., Chen, M., Liu, Z., Liu, W., Guo, Q., Huang, W., et al. Dancegrpo: Unleashing grpo on visual generation. *arXiv preprint arXiv:2505.07818*, 2025.

