# OpenReview forum: "Test-Time Reinforcement Learning for Flow Matching"
_ICML.cc/2026/Conference — ICML 2026 regular_

### Official Review · Reviewer_16WY · 2026-03-06

**Soundness:** 3
**Presentation:** 3
**Significance:** 3
**Originality:** 3
**Overall Recommendation:** 5
**Confidence:** 4

**Summary:**

This paper proposes Flow-TTRL, a test-time reinforcement learning framework for flow-matching text-to-image modes. Unlike prior approaches that require expensive parameter fine-tuning, Flow-TTRL operates entirely at inference time by treating latent trajectories as implicit policies and optimizing them using reward signals.
The paper introduces several key components:A latent-space reinforcement learning formulation. A reference latent trajectory with KL regularization to prevent reward hacking. An SDE-based rollout strategy to enable exploration. A two-stage optimization strategy: PRDP for early structure stabilization and GRPO for preference refinement.
Meanwhile, experiments on SD3.5-Medium and FLUX.1 Dev demonstrate substantial improvements across multiple benchmarks, without any parameter fine-tuning.
Overall, the paper proposes a novel paradigm for aligning generative models at test time.

**Compliance With Llm Reviewing Policy:**

Affirmed.

**Final Justification:**

I have raised the score since my concerns have been resolved.

**Key Questions For Authors:**

- Since the paper’s main selling point is avoiding offline fine-tuning, could the authors provide a more detailed wall-clock analysis of inference cost, ideally decomposed by backbone, number of inference steps, group size, reward-model evaluation cost, and hardware? A stronger answer here would improve my confidence in the practical impact of the method.
- Could the authors clarify more explicitly when they believe Flow-TTRL should be preferred over offline RL fine-tuning: lower total compute budget, greater flexibility, easier deployment, personalization, or other settings?
- How sensitive is the method to the choice of reward model and reward scaling? Have the authors observed reward-model-specific artifacts or reward hacking, and if so, how much does the KL/reference-trajectory mechanism mitigate them?

**Limitations:**

yes

**Strengths And Weaknesses:**

# Pros
- The paper introduces a fundamentally new alignment paradigm by performing reinforcement learning at inference time rather than training time. Treating latent trajectories as policies is a novel and influential idea that may influence future work in diffusion, flow matching, and generative model alignment.
- The formulation is well justified theoretically. The use of reference latent trajectories and KL regularization effectively prevents reward hacking and trajectory collapse. The use of a  two-stage optimization design is particularly well motivated:PRDP stabilizes early noisy stages and GRPO improves preference and aesthetic alignment.This staged approach improves both stability and performance.
- The separation between an early PRDP stage for structural stabilization and a later GRPO stage for finer preference optimization matches the intuition that early denoising mainly controls global composition while later stages refine detail and aesthetics. The ablation results support that both stages contribute meaningfully. In particular, removing PRDP or GRPO causes visible drops across GenEval / Complex / preference metrics.
- The method requires no training, no dataset, and no parameter updates, making it highly practical for deployment. It can be applied to many other fields in future such as closed-source models and personalized generation This significantly improves usability compared to training-based alignment methods.

# Cons
- Although the method outperforms the baseline model, its gains remain limited compared with training-based methods such as Flow-GRPO. On GenEval, its performance is still inferior to Flow-GRPO. On T2I-CompBench, while it appears to outperform Flow-GRPO on the surface, the Flow-GRPO results reported in the table are obtained by training on the GenEval dataset and then evaluating on T2I-CompBench. I am not sure whether this comparison is entirely fair. I therefore think the paper should present itself primarily as a flexible and practical alternative, rather than a general replacement for RL fine-tuning. This does not invalidate the contribution, but it slightly tempers the significance claim.
- The paper’s main appeal is avoiding parameter updates, but this comes at the cost of repeated candidate rollout, reward evaluation, and latent optimization inside a single sampling run. The paper does report a computation-efficiency figure and notes that the method is a form of test-time scaling, but the latency analysis remains somewhat coarse. For example, I would have liked a more explicit wall-clock breakdown across inference steps, group size, reward model cost, and GPU setup, since practical usability is central to the paper’s claim.
- The paper does not analyze convergence guarantees or stability properties of latent optimization.Such analysis would strengthen the theoretical foundation.

---

> ### Author Rebuttal · Authors · 2026-03-31
>
> Dear Reviewer 16WY,
>
> **Thank you for providing such detailed and thorough feedback!**
>
> **W1:** Thank you for pointing this out. As stated in L.362 and L.849, the positioning of this paper is to provide a **high-performing and efficient alternative** to training-based paradigms. We agree with your assessment and will improve our statements.
>
> **W2:** Tables 1–3 provide a detailed wall-clock breakdown (s, summed over steps) on 100 random prompts across various backbones. The rollout branches are independent, thus they can be **parallelized** when VRAM allows, as indicated in the **"+Batch"** columns. The latency includes Rollout (SDE+ODE sampling), RM Eval (VAE decoding and model evaluation), Update (latent optimization), and Other (inherent pipeline overhead with the reference latent). Unless otherwise specified, the settings follow Appendix B.1, conducted on NVIDIA A800 80G.
>
> **Table 1: Hardware**
>
> Settings: GPU (A800 80G, RTX 4090 24G).
>
>
> |Backbone|GPU|Rollout/+Batch|RM Eval/+Batch|Update|Other|Total/+Batch|
> |---|---|---|---|---|---|---|
> |SD 3.5-M+Flow-TTRL|A800|25.68/12.35|71.15/27.02|24.68|11.35|132.86/75.40|
> ||RTX 4090|22.47/OOM|56.36/24.12|27.15|8.89|114.87/82.63|
> |Flux.1 Dev+Flow-TTRL|A800|100.32/38.23|71.31/28.18|83.54|31.36|286.53/181.31|
> ||RTX 4090|93.19/OOM|48.72/19.80|82.67|25.71|250.29/221.37|
>
> **Table 2: Inference Steps**
>
> |Backbone|Steps|Rollout/+Batch|RM Eval/+Batch|Update|Other|Total/+Batch|
> |---|---|---|---|---|---|---|
> |SD 3.5-M+Flow-TTRL|10|5.85/3.24|15.96/7.32|11.25|3.17|36.23/24.98|
> ||20|12.37/7.08|37.69/13.26|18.81|5.11|73.98/44.26|
> ||40|25.68/12.35|71.15/27.02|24.68|11.35|132.86/75.40|
> |Flux.1 Dev+Flow-TTRL|10|16.51/8.96|17.15/7.14|34.49|10.01|78.16/60.60|
> ||20|42.92/18.66|37.68/17.94|65.48|16.95|163.03/119.03|
> ||40|100.32/38.23|71.31/28.18|83.54|31.36|286.53/181.31|
>
> **Table 3: Group Size**
>
>
> |Backbone|Group| Rollout/+Batch |RM Eval/+Batch|Update|Other|Total/+Batch|
> |---|---|---|---|---|---|---|
> |SD 3.5-M+Flow-TTRL|4|12.39/9.38|35.14/20.86|23.41|11.37|82.31/65.02|
> ||8|25.68/12.35|71.15/27.02|24.68|11.35|132.86/75.40|
> ||12|38.17/14.56|94.34/30.24|25.65|12.02|170.18/82.47|
> |Flux.1 Dev+Flow-TTRL|4|51.79/31.67|34.76/23.55|81.78|30.71|199.04/167.71|
> ||8| 100.32/38.23| 71.31/28.18| 83.54| 31.36 | 286.53/181.31|
> ||12|123.42/43.55|90.26/30.32|84.62|31.19|329.49/189.68|
>
> Flow-TTRL meets practical deployment requirements and offers **a superior performance-efficiency trade-off over other test-time optimization (TTO) methods (please refer to reviewer GMCn W3).**
>
> **W3:** The convergence analysis is analogous to [1] (Appendix B), briefly outlined here:
>
> Flow-TTRL optimizes the latent $x_t$ by maximizing Eq.6, we denote the objective as $J(x_t)$.
>
> We assume $J(x_t)$ is $L$-smooth in the latent space, meaning its gradient $\nabla J$ is $L$-Lipschitz continuous. This is a standard assumption in latent spaces (Theorem 1 in [1]). The Descent Lemma ensures that for any step $x_t^{(k)}$ and $x_t^{(k+1)}$:
>
> $$J(x_t^{(k+1)}) \geq J(x_t^{(k)}) + \langle \nabla J(x_t^{(k)}), x_t^{(k+1)} - x_t^{(k)} \rangle - \frac{L}{2} \|x_t^{(k+1)} - x_t^{(k)}\|^2$$
>
> Substituting our update rule $x_t^{(k+1)} = x_t^{(k)} + \eta \nabla J(x_t^{(k)})$, we obtain:
>
> $$J(x_t^{(k+1)}) - J(x_t^{(k)}) \geq \eta \left(1 - \frac{L \eta}{2}\right) \|\nabla J(x_t^{(k)})\|^2$$
>
> Provided $0 < \eta < 2/L$, the objective value is monotonically increasing at each iteration.
>
> ---
>
> **Q1:** Please refer to W2.
>
> **Q2:** Flow-TTRL should be preferred in settings requiring:
>
> **1.Personalization:** Flow-TTRL enables flexible control over alignment intensity and reward interpolation (see Fig. 19) within a single inference pass.
>
> **2.Greater Flexibility:** The in-context reinforcement learning of Flow-TTRL enables an "All-in-One" solution where a single frozen base model handles diverse tasks via modular reward functions.
>
> **3.Lower Total Compute Budget:** Flow-TTRL serves as a powerful alternative when data or computational resources are insufficient for fine-tuning.
>
> **Q3:**
>
> **1.Sensitivity:** Flow-TTRL is **highly responsive** to the direction of different reward signals and their interpolation weights (see Fig. 19). However, it remains **robust** to the absolute reward scaling thanks to the group-wise advantage normalization in GRPO and the KL regularization.
>
> **2.Reward Hacking:** Without the KL constraint, we indeed observed reward-model-specific artifacts, such as **over-saturation (HPSv2)** or **distorted textures (verifiable rewards)** (see Figs. 15, 18).
>
> **3.Mitigation:** KL constraints improve alignment while guiding the generated images to reside within the intersection of the pre-trained manifold and high-reward regions, whereas omitting them sacrifices the prior (see Figs. 4,5,13,15).
>
> References:
>
> 1. Inference-Time Alignment of Diffusion Models with Direct Noise Optimization
>
> **We hope the responses can help solve your questions. Thanks again for your thorough review and looking forward to your reply!**

---

> > ### Author Rebuttal · Reviewer_16WY · 2026-04-02
> >
> > All my concerns are resolved.

---

> > > ### Author Response · Authors · 2026-04-02
> > >
> > > Thank you for acknowledging our clarifications and responses, as well as for your positive assessment of our work.
> > >
> > > Your constructive comments and suggestions have been instrumental in helping us further enhance the clarity and quality of our paper. We truly appreciate your time and support throughout the review process.

---

### Official Review · Reviewer_GMCn · 2026-03-08

**Soundness:** 2
**Presentation:** 1
**Significance:** 2
**Originality:** 2
**Overall Recommendation:** 3
**Confidence:** 4

**Summary:**

The paper, Flow-TTRL, proposes a test-time RL framework for flow-based models instead of fine-tuning model parameters. The framework performs stochastic rollouts using an SDE formulation and evaluates one-step denoised candidates with the reward function. The optimization consists of two strategies: an early PRDP to stabilize structural semantics, followed by GRPO to refine fine-grained details. The approach maintains a reference trajectory through a KL constraint. Experiments on flow-matching backbones show improvements on benchmarks including GenEval and T2I-CompBench.

**Compliance With Llm Reviewing Policy:**

Affirmed.

**Final Justification:**

All of my concerns have been addressed. Specifically, my main concern regarding the lack of baseline experiments was resolved by including additional test-time optimization methods (e.g., DAS, TITAN-Guide).

**Key Questions For Authors:**

1. Typo in Eq. (4): period should be replaced with a comma.
2. Typo in L.274: a period should be added at the end of the sentence.
3. Typo in Eq. (10): the prompt variable c should be typeset in bold for consistency in the paper.

**Limitations:**

Yes

**Strengths And Weaknesses:**

Strengths:

**1. Test-time adaptation of RL objectives.**

The paper adapts RL objectives such as PRDP and GRPO typically used for training to an inference-time. This may offer a practical alternative when fine-tuning large generative models is computationally infeasible.

**2.Strong improvements across benchmarks.**

The experimental results demonstrate substantial improvements in text-image alignment and human preference. These results suggest that the proposed test-time optimization can reach performance comparable to RL-based fine-tuning methods without requiring additional training.

**3. Additional experiments with multiple rewards.**

Although not emphasized in the main text, the appendix includes experiments using multiple reward signals. These experiments indicate that the method can combine several reward objectives, which is relevant for real-world alignment scenarios.

Weaknesses:

**1. Inconsistency between theoretical formulation and reward computation.**

In the PRDP derivation (Eq. 8), the reward is defined on the final generated sample $x_0$ through denoising trajectory. The appendix also appears to derive the objective using $x_0$. However, the algorithm computes rewards using a single-step projection $\hat{x}_0$. This mismatch raises concerns about whether the theoretical derivation remains valid under the approximation. The issue may be particularly significant in early denoising steps where the latent state is highly noisy and the $\hat{x}_0$  may poorly approximate the final sample.

**2. Claim of statement.**

The paper claims to introduce Flow-TTRL as ''the first test-time RL framework'' for flow-based model (L.105). While the combination of RL objectives with flow-matching models is novel, the broader concept of test-time optimization already exists (refer to Weakness 3). Prior work has proposed guidance methods that modify sampling trajectories using gradients during inference. A comparison with these methods would help clarify the novelty of the proposed approach.

**3. Lack of comparison with existing test-time guidance methods.**

The experiments mainly compare Flow-TTRL with RL-based fine-tuning methods or baseline models. However, since the main advantage of the proposed framework is training-free alignment, comparisons with other inference-time guidance approaches would be more appropriate. For example, methods such as gradient-based energy guidance (FreeDoM [1]), inference-time optimization (TITAN-Guide [2], which is already cited by the authors), non-gradient guidance (SCG[3]), or SMC-based approach [4]. Empirical comparisons in terms of both performance and runtime would strengthen the paper.

[1] Yu, Jiwen, et al. "Freedom: Training-free energy-guided conditional diffusion model." ICCV. 2023.

[2] Simon, Christian, et al. "TITAN-Guide: Taming Inference-Time AligNment for Guided Text-to-Video Diffusion Models." ICCV. 2025.

[3] Huang, Yujia, et al. "Symbolic music generation with non-differentiable rule guided diffusion." ICML. 2024

[4] Kim, Sunwoo, et al. "Test-time alignment of diffusion models without reward over-optimization." ICLR. 2025.

---

> ### Author Rebuttal · Authors · 2026-03-31
>
> Dear Reviewer GMCn,
>
> **Many thanks for your valuable comments, which have greatly helped us improve our work!**
>
> **W1:**
>
> **1.Theoretical-Reward Consistency:** Estimating $x_0$ via single-step projection to $\hat{x}_0$ (analogous to Tweedie’s formula [1-2]) is a **widely adopted** and **empirically validated practice** in test-time optimization (TTO) [3-9] and RL [10-12] to simplify the computation graph or gain dense rewards. **Approximation error analysis is analogous to [6] (Appendix A.4).** Therefore, the theoretical formulation and reward computation can be regarded as mutually **matched and consistent**.
>
> **2.Early Structural Stability via PRDP:** PRDP utilizes pairwise reward differences $\Delta r_{ij}$ to mitigate estimation biases under the KL constraint (as confirmed by Table 3 and Fig. 6) . Aligning global structure, rather than fine-grained aesthetics, is highly feasible even under high-noise conditions (see Figs. 8, 9, 14, 17).
>
> **3.Implicit Search for Clean Projections:** Reward models naturally favor latent trajectories that yield cleaner and more recognizable $\hat{x}_0$ projections. This preference enables Flow-TTRL to perform an implicit search for clean projections, explaining the high-fidelity results even in few-step settings (see Fig. 10).
>
> **W2:** From an application perspective, Flow-TTRL serves as a training-free alternative to RL. The following table highlights the differences between Flow-TTRL and TTO.
>
> ||TTO|Flow-TTRL|
> |-|-|-|
> |Perspective|Pure Denoising ODE|Unified ODE+SDE into MDP|
> |Reward Compatibility|Differentiable rewards only|General (Non-differentiable, verifiable, etc.; see Appendix D.4) |
> |Reward Hacking|KL-Incompatible (Susceptible to reward hacking) |KL-Compatible (Robust alignment)|
>
> We provide a detailed comparison with TTO methods (please refer to W3).
>
> **W3:** We conducted additional experiments on GenEval, T2I-CompBench (Complex), and OCR. All settings follow those specified in Appendix B.1. For the OCR task, an additional OCR reward term is introduced with a weight of 0.5. The reported time reflects the average generation latency per sample. The non-gradient SCG [9] is methodologically similar to latent selection (please see responses to Reviewer UEyp Q1 and 4d2g W2). Results:
>
> |Models|Time(s)|GenEval↑|Complex↑|OCR↑|
> |---|---|---|---|---|
> |SD3.5-M|11.14|0.63|0.3542|0.59|
> |FreeDoM[3] (ICCV,2023)|253.43|0.72|0.3689|0.62|
> |DAS[13] (ICLR,2025)|279.78|0.83|0.3847|0.68|
> |TITAN-Guide[14] (ICCV,2025)|221.24|0.74|0.3791|0.62|
> |DATE[6] (NeurIPS,2025)|104.58|0.79|0.3759|0.64|
> |**Flow-TTRL**|132.86|**0.87**|**0.4045**|**0.78**|
>
> Conclusion:
>
> 1. **Superior Overall Alignment:** Flow-TTRL achieves the best performance and consistently improves semantic and textual alignment across GenEval, Complex, and OCR tasks.
> 2. **Effective Optimization of Non-differentiable Rewards:** A major challenge in OCR tasks is that the reward functions are typically **non-differentiable**, making it difficult for baselines to provide effective supervision. While DAS uses a differentiable surrogate model (CLIP+MLP) to approximate OCR scores, this approximation remains suboptimal and time-consuming. Flow-TTRL demonstrates a clear advantage.
> 3. **Better Performance-Efficiency Balance:** Flow-TTRL achieves high efficiency via single-step projection and fewer iterations. FreeDoM and DAS require multiple gradient-based iterations at each timestep, and TITAN-Guide suffers from slow inference due to its complex computation graph. While DATE is slightly faster because it avoids the extensive rollout process and also utilizes single-step projection to $\hat{x}_0$, our results show that the rollout is essential for achieving higher alignment quality. Parallelization drastically speeds up rollout (please see response to 16WY W2).
>
> ---
>
> **Q1-3:** Thanks for spotting the typos. All fixed now.
>
> **References:**
>
> 1. An empirical Bayes approach to statistics
> 2. Tweedie’s formula and selection bias
> 3. Freedom: Training-free energy-guided conditional diffusion model
> 4. Universal guidance for diffusion models
> 5. Prompt-tuning latent diffusion models for inverse problems
> 6. Diffusion Adaptive Text Embedding for Text-to-Image Diffusion Models
> 7. ReNeg: Learning Negative Embedding with Reward Guidance
> 8. Dynamic prompt optimizing for text-to-image generation
> 9. Symbolic Music Generation with Non-Differentiable Rule Guided Diffusion
> 10. Imagereward: Learning and evaluating human preferences for text-to-image generation
> 11. Directly Fine-Tuning Diffusion Models on Differentiable Rewards
> 12. Directly Aligning the Full Diffusion Trajectory with Fine-Grained Human Preference
> 13. Test-time Alignment of Diffusion Models without Reward Over-optimization
> 14. TITAN-Guide: Taming Inference-Time AligNment for Guided Text-to-Video Diffusion Models
>
> **We appreciate your guidance! Feel free to let us know if you have any further questions. If you feel your concerns are addressed, we respectfully hope you could reevaluate our work!**

---

> > ### Author Rebuttal · Reviewer_GMCn · 2026-04-01
> >
> > I acknowledge the additional baseline comparisons provided during the limited rebuttal period. The proposed method exhibits performance improvements over the baselines presented in the author response. However, establishing a definitive comparison with existing test-time optimization methods or equal-NFE baselines necessitates more extensive experiments. The evaluation protocols for these baselines must parallel the scope of Main Tables 1 and 2, extending beyond the limited metrics provided in the rebuttal. Instead of comparing all the baselines, it would be more interesting to know the relative advantages of one or two representative approaches.
> >
> > Regarding the theoretical formulation, the criticism was directed at the omission of the approximation context within the manuscript. While the empirical use of the single-step projection may be mathematically acceptable, the text lacks any explicit justification connecting the terminal state in the derivation to the implemented approximation. This should be revised in the manuscript. In addition, it remains unclear how the reward model encourages cleaner and more recognizable latent trajectories.
> >
> > Based on the requirement for more comprehensive baseline evaluations and the descriptive gap in the theoretical section of the manuscript, my initial recommendation is maintained. However, if the baseline comparison issues are adequately addressed, I am open to reconsidering my score. I would be glad if the authors could reply back regarding this matter.

---

> > > ### Author Response · Authors · 2026-04-04
> > >
> > > Dear Reviewer GMCn,
> > >
> > > **We sincerely appreciate your constructive and insightful feedback!**
> > >
> > > > **Q1: Baseline comparison**
> > >
> > > Extend the experiments on GenEval and T2I-CompBench with TTO methods (**DAS [1] and TITAN-Guide [2]**) and equal-NFE sampling baselines (**Best-of-N, $N=8$**). The quantitative results are shown in **Tables 1-2**, with qualitative comparisons available at: [`SD3.5-GenEval`](https://anonymous.4open.science/r/pic-E3BB/SD35-geneval_TTO.png), [`FLUX-GenEval`](https://anonymous.4open.science/r/pic-E3BB/FLUX-geneval_TTO.png), [`SD3.5-T2ICompBench`](https://anonymous.4open.science/r/pic-E3BB/SD35-T2ICompBench_TTO.png), and [`FLUX-T2ICompBench`](https://anonymous.4open.science/r/pic-E3BB/FLUX-T2ICompBench_TTO.png). **These results will be incorporated into the revised experimental section.**
> > >
> > > ### Table1:GenEval
> > >
> > > |Backbone|Method|Overall↑|Single↑|Two↑|Counting↑|Colors↑|Position↑|Color Attrib.↑|Time(s)|
> > > |-|-|-|-|-|-|-|-|-|-|
> > > |SD 3.5-M|Base|0.63|0.98|0.78|0.50|0.81|0.24|0.52|10.57|
> > > ||Best-of-N($w/o\ L_2$)|0.71|0.98|0.89|0.70|0.80|0.31|0.57|99.04|
> > > ||Best-of-N($w\ L_2$)|0.70|0.98|0.91|0.63|0.83|0.28|0.55|100.81|
> > > ||DAS|0.83|**1.00**|0.98|0.89|0.86|**0.57**|0.67|281.20|
> > > ||TITAN-Guide|0.74|**1.00**|0.91|0.76|0.85|0.32|0.60|222.37|
> > > ||Flow-TTRL|**0.87**|**1.00**|**0.99**|**0.95**|**0.90**|0.55|**0.85**|130.85|
> > > |FLUX.1 Dev|Base|0.66|0.98|0.81|0.74|0.79|0.22|0.45|27.61|
> > > ||Best-of-N($w/o\ L_2$)|0.69|0.98|0.84|0.77|0.79|0.23|0.53|191.95|
> > > ||Best-of-N($w\ L_2$)|0.70|0.98|0.90|0.79|0.84|0.22|0.49|197.72|
> > > ||DAS|0.80|**1.00**|0.96|0.81|**0.90**|0.37|0.73|609.15|
> > > ||TITAN-Guide|0.76|0.99|0.90|0.83|0.88|0.34|0.64|463.32|
> > > ||Flow-TTRL|**0.83**|**1.00**|**0.97**|**0.94**|**0.90**|**0.41**|**0.77**|284.07|
> > >
> > > ### Table2:T2I-CompBench
> > >
> > > |Backbone|Method|Color↑|Shape↑|Texture↑|2D-Spatial↑|Non-Spatial↑|Complex↑|Time(s)|
> > > |-|-|-|-|-|-|-|-|-|
> > > |SD 3.5-M|Base|0.7994|0.5669|0.7338|0.2850|0.3146|0.3542|11.86|
> > > ||Best-of-N($w/o\ L_2$)|0.8127|0.5691|0.7459|0.2912|0.3107|0.3671|101.34|
> > > ||Best-of-N($w\ L_2$)|0.8113|0.5744|0.7396|0.2977|0.3152|0.3697|103.58|
> > > ||DAS|0.8561|0.6482|0.7717|0.3679|0.3294|0.3847|284.61|
> > > ||TITAN-Guide|0.8468|0.6235|0.7689|0.3725|0.3276|0.3791|223.64|
> > > ||Flow-TTRL|**0.9042**|**0.7361**|**0.8261**|**0.4414**|**0.3319**|**0.4045**|130.86|
> > > |FLUX.1 Dev|Base|0.7407|0.5718|0.6922|0.2863|0.3127|0.3703|27.82|
> > > ||Best-of-N($w/o\ L_2$)|0.7556|0.5896|0.7068|0.2924|0.3101|0.3798|188.52|
> > > ||Best-of-N($w\ L_2$)|0.7682|0.5832|0.7015|0.2899|0.3115|0.3784|197.14|
> > > ||DAS|0.8371|**0.6779**|0.7689|0.3662|0.3161|0.3951|614.71|
> > > ||TITAN-Guide|0.8217|0.6511|0.7673|0.3786|0.3154|0.3853|461.57|
> > > ||Flow-TTRL|**0.8804**|0.6717|**0.7958**|**0.4390**|**0.3229**|**0.4179**|285.13|
> > >
> > > **Observations:**
> > >
> > > + **Optimization vs. Selection**: Comparing TTO with Best-of-N methods confirms that while selection provides marginal gains, **latent optimization** is essential for handling complex and structured prompts.
> > > + **Necessity of Sampling**: The performance gap between sampling-aware methods (DAS, Flow-TTRL) and TITAN-Guide underscores that **comprehensive perception of the reward landscape through sampling** is necessary.
> > > + **Efficiency and Effectiveness**: By integrating **In-context RL into a KL-regularized MDP**, Flow-TTRL effectively achieves a superior trade-off between alignment and speed compared to traditional TTO.
> > > + **Findings align with Tables 1-2 in the manuscript.**
> > >
> > > > **Q2: theory gap**
> > >
> > > We will incorporate the following revisions:
> > >
> > > 1. **L.231 (Justifications):** Complement the **justifications (please see the previous rebuttal), references, and the intuition (please refer to Q3)** for single-step projection.
> > > 2. **Appendix (Proof):** Provide a theoretical proof of **approximation error analysis** analogous to the derivation in [3] (Appendix A.4).
> > > 3. **Section 4.2 (PRDP):** Elaborate on how PRDP helps alleviate estimation biases using $\Delta r_{ij}$.
> > >
> > > > **Q3: how the reward model encourages cleaner trajectories**
> > >
> > > 1. **Artifact suppression as a core preference dimension:** The reward model  (typically trained on human preferences) [4] inherently rates images with fewer artifacts higher. This preference is baked into the scalar reward it outputs for any trajectory.
> > >
> > > 2. Suppose we have two trajectories for the same prompt: At the critical early stage (t=0.9) **Trajectory A scores much higher (0.85) than Trajectory B (0.45) due to fewer artifacts**.
> > >
> > > 3. Flow-TTRL computes the advantage for steps in A vs. B. Through policy updates (e.g., latent optimization via gradient ascent in Eq. 6), it **increases the likelihood of the latents that produced Trajectory A**. This achieves implicit search for artifact suppression.
> > >
> > > **References:**
> > >
> > > 1. Kim et al. (DAS, ICLR 2025)
> > > 1. Simon et al. (TITAN-Guide, ICCV 2025)
> > > 1. Na et al. (DATE, NeurIPS 2025)
> > > 1. Xu et al. (Imagereward, NeurIPS 2023)
> > >
> > > **Thank you again for your patient and detailed feedback. We hope our response has addressed your concerns; if so, we respectfully hope you could re-evaluate our work.**

---

### Official Review · Reviewer_UEyp · 2026-03-13

**Soundness:** 3
**Presentation:** 2
**Significance:** 3
**Originality:** 3
**Overall Recommendation:** 4
**Confidence:** 4

**Summary:**

This paper proposes Flow-TTRL, a test-time reinforcement learning framework for flow-matching models to achieve high alignment with human preferences during inference without requiring model fine-tuning. It reinterprets intermediate latent representations as an implicit policy and employs a coarse-to-fine optimization strategy, using Proximal Reward Difference Prediction (PRDP) at the inital sampling steps followed by Group Relative Policy Optimization (GRPO) for aesthetic refinement at later steps. Experimental results demonstrate that Flow-TTRL significantly improves text-image alignment and aesthetic quality across multiple benchmarks.

**Compliance With Llm Reviewing Policy:**

Affirmed.

**Final Justification:**

The authors addressed my concerns. I am inclined to acceptance after seeing that the authors' method outperforms the best-of-N baselines and other test time methods like TITAN-Guide and DAS under the same computational budget.

**Key Questions For Authors:**

- Given the conceptual overlap between your method and iterative selection strategies, it is recommended that you include a "best-of-N latent selection" baseline in your experiments. Specifically, comparing Flow-TTRL against a version that simply selects the best candidate from the group $G$ at each step (with and without a KL/distance-based constraint) would more clearly isolate the specific value added by the gradient-based optimization (PRDP/GRPO) versus simple increased sampling density in the latent space. It would be good to discuss the differences of your method's intuition and this "best-of-N latent selection" baseline's intuition.

**Limitations:**

Yes.

**Strengths And Weaknesses:**

## Strengths
- By reinterpreting intermediate latents as a mutable implicit policy, the paper enables alignment without modifying the underlying flow-matching model's weights. This approach allows the system to "steer" trajectories toward high-reward regions of the existing vector field using SDE-based rollouts, effectively bridging the gap between fixed generative models and dynamic reinforcement learning.
- The method delivers significant performance gains, such as increasing GenEval accuracy from 63% to 87% for SD 3.5-Medium, while remaining compatible with consumer-grade hardware. These results demonstrate that Flow-TTRL achieves alignment quality comparable to state-of-the-art fine-tuning methods.
- The authors provide detailed analyses for the two-stage PRDP-to-GRPO strategy, the impact of inner-loop iterations ($K$), the necessity of SDE-based stochasticity ($\alpha$), and the role of KL-divergence constraints in maintaining generative stability. The results seem to support the necessity of different components in the Flow-TTRL framework.

## Weaknesses
- The primary drawback of Flow-TTRL is the substantial computational overhead it introduces during the inference process. Unlike standard sampling using a fine-tuned model, this method requires performing iterative latent optimization for many timesteps. For each of these steps, the framework must generate a group of candidate trajectories ($G=8$), decode them to the pixel space, evaluate them using external reward models, and then perform multiple gradient-based updates ($K=2$) on the latent representation. While this "plug-and-play" approach avoids the massive one-time cost of model fine-tuning, it significantly increases the latency for generating each individual image, shifting the burden of computation directly to the user's sampling time.
- During the early warm-up stages of denoising, the latent state is highly fragile, and external reward models can be inaccurate due to the high levels of noise. This makes the optimization use unreliable reward signals that can potentially lead to trajectory collapse.

### Minor issues
- There is an extra ')' in Eq. 4.
- Why is the expectation over the condition $c ~ p(c)$ used in Eq. 4? Why is $p(c)$ involved here if you are optimizing the latents (where the condition should be already given)?

---

> ### Author Rebuttal · Authors · 2026-03-31
>
> Dear Reviewer UEyp,
>
> **Thank you for your insightful comments!**
>
> **W1:**
>
> **1.Parallelization drastically speeds up rollout and decoding with enough memory** (please refer to reviewer 16WY W2).
>
> **2.Better performance-efficiency balance than other Test-Time Optimization (TTO) methods** (please refer to reviewer GMCn W3).
>
> **3.Scalability and Robustness:** While Flow-TTRL achieves optimal performance at $G=8$ and $K=2$, Figs. 7b, 10 demonstrate that it consistently leads traditional methods even with smaller group sizes or fewer timesteps, ensuring **adaptability** to diverse real-world needs.
>
> **4.Agnostic and Extensible Design:** Flow-TTRL is **algorithm-agnostic** and **possesses immense future** potential to integrate with emerging efficient RL techniques designed to eliminate group rollouts or the VAE decoding process.
>
> **W2:**
>
> **1.Necessity and Feasibility**: Since global structural features emerge and differentiate early (see Figs. 8, 9, 14, 17), aligning structure, rather than aesthetics, becomes feasible with the KL constraint.
>
> **2.PRDP Mitigates Biases**: PRDP employs pairwise reward differences $\Delta r_{ij}$ to mitigate inaccurate reward signals (as confirmed by Table 3 and Fig. 6).
>
> **3.Implicit Search for Clean Projections:** Reward models favor trajectories with cleaner $\hat{x}_0$ projections, enabling Flow-TTRL to perform an implicit search for clean projections. This explains its effectiveness in few-step settings (see Fig. 10).
>
> **4.Approximation error analysis is analogous to [1] (Appendix A.4).**
>
> **5.More details (please refer to reviewer GMCn W1).**
>
> ---
>
> **Minor issue 1**: Corrected, thanks!
>
> **Minor issue 2**: Thank you for your sharp observation. This is a formal oversight inherited from conventional RL and does not compromise the validity of Flow-TTRL. The corrected Eq. 4 is:
>
> $$x_t^* = x_t + \eta \nabla_{x_t} R(\Phi_\theta(x_{T:0}, \mathbf{c}))$$
>
> ---
>
> **Q1:**
>
> We conducted a comparative analysis against **Best-of-N** ($N=8$) and **Beam Search** ($N=8, B=2$) on GenEval, T2I-CompBench (Complex), Pick-a-Pic, and OCR. $N$ denotes the number of candidate branches generated from each parent branch at each timestep, while $B$ represents the number of top-performing branches retained for the subsequent step. $w/\ L_2$ signifies that the optimization is constrained by the $L_2$ distance to the original ODE trajectory at each step, while $w/o\ L_2$ does not. All other experimental configurations follow Appendix B.1. Results:
>
> |Backbone|Method|Time (s)|GenEval↑|Complex↑|PickScore↑|HPSv2↑|ImageReward↑|OCR↑|
> |-|-|-|-|-|-|-|-|-|
> |SD3.5-M|Best-of-N($w/o\ L_2$)|100.76|0.71|0.3671|21.96|0.289|0.840|0.61|
> ||Best-of-N($w/ L_2$)|102.94|0.70|0.3697|21.89|0.291|1.014|0.61|
> ||Beam Search($w/o\ L_2$)|185.41|0.73|0.3712|22.11|0.296|0.961|0.61|
> ||Beam Search($w/\ L_2$)|187.73|0.73|0.3751|22.07|0.293|1.039|0.63|
> ||**Flow-TTRL**|130.48|**0.87**|**0.4045**|**22.24**|**0.299**|**1.187**|**0.78**|
> |Flux.1 Dev|Best-of-N($w/o\ L_2$)|189.67|0.69|0.3798|22.34|0.318|1.148|0.72|
> ||Best-of-N($w/\ L_2$)|193.15|0.70|0.3784|22.32|0.317|1.072|0.72|
> ||Beam Search($w/o\ L_2$)|324.28|0.72|0.3869|22.70|0.322|1.287|0.71|
> ||Beam Search($w/\ L_2$)|328.60|0.73|0.3843|**22.73**|**0.324**|1.154|0.72|
> ||**Flow-TTRL**|285.97|**0.83**|**0.4179**|22.67|0.322|**1.340**|**0.84**|
>
> Observations:
>
> - **Performance Advantage:** While latent searching methods improve simple alignment, Flow-TTRL significantly outperforms them in compositional (GenEval, Complex) and structural (OCR) tasks.
> - **Regularization Robustness:** $w/o\ L_2$ suffers from reward hacking, which results in unnatural lighting and distorted structures. $w/\ L_2$ fails to effectively balance alignment and generation quality. Flow-TTRL maintains superior visual integrity.
> - **Better Trade-Off:** Flow-TTRL achieves higher generation quality with a better latency-to-performance trade-off than latent search methods.
>
> Differences between Flow-TTRL and latent selection:
>
> + **Continuous Heuristic Optimization**: Flow-TTRL performs heuristic optimization within the continuous latent space to reach statistically rare high-reward regions, while latent selection is restricted to searching according to the prior, which may be ineffective under verifiable rewards (Appendix D.4).
> + **Proactive Guidance**: Flow-TTRL optimizes the current latent state using context derived from future feedback analogous to in-context reinforcement learning [2], while latent selection cannot.
> + **Inherent Regularization**: The MDP of Flow-TTRL is naturally compatible with KL-divergence to prevent reward hacking, while latent selection is not.
>
> **References:**
>
> 1. Diffusion Adaptive Text Embedding for Text-to-Image Diffusion Models
> 2. In-Context Reinforcement Learning through Bayesian Fusion of Context and Value Prior
>
> **Thank you again for helping us improve the paper! Please let us know if you have any further questions. If you feel your concerns are addressed, please consider reevaluating our work!**

---

> > ### Author Rebuttal · Reviewer_UEyp · 2026-04-04
> >
> > Thanks for the response. The Flow-TTRL used more time than Best-of-N in the optimization, which makes the results less convincing. I would like to see the performance of Best-of-N / Beam Search using the same time budget as the Flow-TTRL (also using the same GPU).

---

> > > ### Author Response · Authors · 2026-04-06
> > >
> > > Dear Reviewer UEyp,
> > >
> > > **Thank you for raising this important point. We have addressed it as follows:**
> > >
> > > > **Q1: Comparison with Latent Selection**
> > >
> > > We conducted **additional experiments** on the **same GPU (A800 40G, as specified in Appendix B)** to ensure a fair comparison. Specifically, for **SD3.5-M**, we compared Flow-TTRL against **Best-of-N ($N=12$)** and **Beam Search ($B=2,N=6$)**. For **Flux.1 Dev**, we used **Best-of-N ($N=16$)** and **Beam Search ($N=8, B=2$)**. The definitions of hyperparameters are consistent with the previous rebuttal. **Since perfectly matching execution time is difficult, we ensured the baselines were granted slightly more inference time than Flow-TTRL** to provide a more convincing and conservative comparison. **Table 1** presents the quantitative results. Qualitative results are available at: [`SD3.5`](https://anonymous.4open.science/r/img-651D/sd35.png) and [`Flux.1 Dev`](https://anonymous.4open.science/r/img-651D/flux1.0.png) (click to view).
> > >
> > > ### Table 1
> > >
> > > | Backbone | Method | Time (s) | GenEval↑ | Complex↑ | PickScore↑ | HPSv2↑ | ImageReward↑ | OCR↑ |
> > > |-|-|-|-|-|-|-|-|-|
> > > | SD3.5-M | Best-of-N($w/o\ L_2$) | 132.16 | 0.73 | 0.3683 | 22.04 | **0.299** | 0.932 | 0.62 |
> > > | | Best-of-N($w/ L_2$) | 134.84 | 0.73 | 0.3702 | 21.88 | 0.292 | 1.066 | 0.62 |
> > > | | Beam Search($w/o\ L_2$) | 138.39 | 0.73 | 0.3674 | 21.99 | 0.295 | 0.879 | 0.61 |
> > > | | Beam Search($w/\ L_2$) | 141.81 | 0.72 | 0.3718 | 22.07 | 0.293 | 1.142 | 0.62 |
> > > | | **Flow-TTRL** | 130.48 | **0.87** | **0.4045** | **22.24** | **0.299** | **1.187** | **0.78** |
> > > | Flux.1 Dev | Best-of-N($w/o\ L_2$) | 301.67 | 0.70 | 0.3821 | 22.59 | 0.323 | 1.191 | 0.72 |
> > > | | Best-of-N($w/\ L_2$) | 303.92 | 0.74 | 0.3847 | 22.48 | **0.324** | 1.315 | 0.73 |
> > > | | Beam Search($w/o\ L_2$) | 324.28 | 0.72 | 0.3869 | 22.70 | 0.322 | 1.287 | 0.71 |
> > > | | Beam Search($w/\ L_2$) | 328.60 | 0.73 | 0.3843 | **22.73** | **0.324** | 1.154 | 0.72 |
> > > | | **Flow-TTRL** | 285.97 | **0.83** | **0.4179** | 22.67 | 0.322 | **1.340** | **0.84** |
> > >
> > > **The results align with our previous findings. Main Observations:**
> > >
> > > 1. **Flow-TTRL delivers superior performance under both equal-NFE (please see our previous rebuttal) and equal-time budgets (please see Table 1).**
> > >
> > > 2. **Superior Performance in Complex Alignment:** While latent searching methods improve basic alignment, **Flow-TTRL significantly outperforms them** in compositional reasoning (GenEval, Complex) and structural accuracy (OCR) **even with lower inference latency**. Specifically,
> > >
> > >    + **GenEval:** Flow-TTRL achieves **0.87 / 0.83** (SD3.5 / Flux) vs. baselines' **0.73 / 0.74**.
> > >
> > >    + **Complex:** Flow-TTRL reaches **0.4045 / 0.4179** (SD3.5 / Flux) vs. baselines' **0.3718 / 0.3869**.
> > >
> > >    + **OCR:** Flow-TTRL scores **0.78 / 0.84** (SD3.5 / Flux) vs. baselines' **0.62 / 0.73**.
> > >
> > > 3. **More Pronounced Scaling Effect:** **Flow-TTRL exhibits a more pronounced scaling effect as the rollout group size increases. (please see Figure 7b)** In contrast, latent selection fails to demonstrate similar scaling characteristics, with the accuracy gains from an expanded search space remaining **marginal**.
> > >
> > > 4. **Robustness to Reward Hacking:** $w/o\ L_2$ frequently leads to **reward hacking**, resulting in unnatural lighting and distorted spatial structures. Even with $L_2$ regularization, these baselines struggle to balance prompt alignment with generative quality. In contrast, **the MDP formulation of Flow-TTRL is inherently compatible with KL-divergence constraints**, allowing it to maintain superior visual aesthetics and naturalism while achieving higher alignment scores.
> > >
> > > **Why Flow-TTRL is Better:**
> > >
> > > 1. **Continuous Heuristic Optimization vs. Prior-Restricted Search:**
> > >    Flow-TTRL performs **heuristic optimization within the continuous latent space to reach statistically rare high-reward regions**. In contrast, latent selection is restricted to searching within the pre-trained prior, which is often ineffective for verifiable rewards (see Appendix D.4).
> > > 2. **In-context Guidance vs. Greedy Search:**
> > >    Flow-TTRL optimizes **the current latent state using context derived from future feedback**, analogous to In-Context reinforcement learning [1]. However, latent selection methods rely on greedy search, which cannot guarantee global optimality and fails to incorporate such trajectory-level guidance.
> > > 3. **Inherent Regularization vs. Rigid Constraints:**
> > >    **The MDP formulation of Flow-TTRL is naturally compatible with KL-divergence**, preventing reward hacking while maintaining generative quality. Latent selection lacks this inherent mechanism and relies on rigid $L_2$ distance constraints.
> > >
> > > **References:**
> > >
> > > 1. In-Context Reinforcement Learning through Bayesian Fusion of Context and Value Prior
> > >
> > > **Thank you again for your time and effort throughout the review process. We hope our responses adequately address your concerns. If you feel your concerns are addressed, please consider re-evaluating our work.**

---

### Official Review · Reviewer_4d2G · 2026-03-13

**Soundness:** 3
**Presentation:** 4
**Significance:** 3
**Originality:** 3
**Overall Recommendation:** 4
**Confidence:** 4

**Summary:**

This paper proposes Flow-TTRL, a test-time RL method for flow matching models in terms of text-to-image generation. Instead of optimizing the model itself, Flow-TTRL aims at optimizing the latent variable at the inference time, avoiding the computation overhead. Flow-TTRL consists of 2 stages, the high-noise warm up stage and the low-noised fine-grained stage, using different rewards, respectively. The experiments show a satisfied performance improvement on text-to-image generation.

**Compliance With Llm Reviewing Policy:**

Affirmed.

**Final Justification:**

All concerns are solved. And I keep my positive score.

**Key Questions For Authors:**

See weakness

**Limitations:**

Yes

**Strengths And Weaknesses:**

Strengths:

(1) The idea of applying RL at the inference time on flow matching is novel.

(2) The two stage division makes sense.

(3) This paper is well-presented, the best in my batch.

Weaknesses:

(1) While SDE-ODE transform is well applied in the flow matching RL community, the authors are encouraged to try the following: applying different SDE parameter (the one controls the randomness) on different stage, or even different steps. This may make the candidate groups in different steps more aligned.

(2) As this approach will "of course" be effective, is there any comparison with pure Best-of-N test time scaling? As SDE rollout is as same computation-costed as Best-of-N, and takes more overhead in the following RL training, is there a big result difference between these two?

(3) It is encouraged to cite the following paper： 【1】Luo, Yifu, Penghui Du, Bo Li, Sinan Du, Tiantian Zhang, Yongzhe Chang, Kai Wu, Kun Gai, and Xueqian Wang. "Sample By Step, Optimize By Chunk: Chunk-Level GRPO For Text-to-Image Generation." arXiv preprint arXiv:2510.21583 (2025).

I appreciate the work this paper presents, and I believe future exploration following this work is worthwhile.

---

> ### Author Rebuttal · Authors · 2026-03-31
>
> Dear Reviewer 4d2G,
>
> **We sincerely thank you for your insightful suggestion and recognition!**
>
> **W1:** This aligns perfectly with the core design philosophy of Flow-TTRL and [1]. As analyzed in Fig 7(a) and Appendix B.1, D.6, Flow-TTRL employs a **dynamic randomness schedule** controlled by the parameter $\alpha$:
>
> 1. **Early Stages:** We apply a larger $\alpha$ to facilitate broader exploration of search space, allowing the model to explore diverse semantic layouts and avoid local optima.
> 2. **Late Stages:** We reduce $\alpha$ to focus on intra-group comparisons of texture details and fine-grained structural alignment.
>
> To further address your concern and explore the optimal scheduling, we have revised our manuscript and extended our original **Linear Decay** implementation to a **Cosine Decay** schedule. The cosine schedule provides a smoother transition, maintaining higher exploration energy in the mid-stage while ensuring stable exploitation at the end. The experimental results on GenEval and T2I-CompBench (Complex) demonstrate that the Cosine Decay schedule further enhances the model's semantic alignment:
>
>
> | | GenEval (Overall)↑ | T2I-CompBench(Complex)↑ |
> |--------------------------|--------------------|----------------------|
> | FLUX.1 Dev | 0.66 | 0.3703 |
> | SD 3.5-M | 0.63 | 0.3542 |
> | FLUX.1 Dev + Flow-TTRL(Linear Decay) | 0.83 | 0.4179 |
> | SD 3.5-M + Flow-TTRL(Linear Decay) | **0.87** | 0.4045 |
> | FLUX.1 Dev + Flow-TTRL(Cosine Decay) | 0.85 | **0.4207** |
> | SD 3.5-M + Flow-TTRL(Cosine Decay) | **0.87** | 0.4051 |
>
> **W2:** We compared Flow-TTRL against inference-time scaling baselines: **Best-of-N** ($N=8$) and **Beam Search** ($N=8, B=2$) [2] across GenEval, T2I-CompBench (Complex), Pick-a-Pic, and OCR. Specifically, $N$ denotes the number of candidate branches generated from each parent branch at each timestep, while $B$ represents the number of top-performing branches retained to proceed to the next timestep. All other settings are fully consistent with those described in the Appendix B.1. In the revised manuscript, we will incorporate a detailed comparison between our method and latent selection approaches to highlight their key distinctions. Results:
>
>
> | Backbone | Method | Time(s) | GenEval↑ | Complex↑ | PickScore↑ | HPS v2↑ | ImageReward↑ | OCR↑ |
> |--------------|------------|----------|--------------|--------------|----------------|-------------|------------------|----------|
> | SD 3.5-M | Best-of-N | 100.76 | 0.71 | 0.3671 | 21.96 | 0.289 | 0.840 | 0.61 |
> || Beam Search | 185.41 | 0.73 | 0.3712 | 22.11 | 0.296 | 0.961 | 0.61 |
> || **Flow-TTRL** | 130.48 | **0.87** | **0.4045** | **22.24** | **0.299** | **1.187** | **0.78** |
> | Flux.1 Dev | Best-of-N | 189.67 | 0.69 | 0.3798 | 22.34 | 0.318 | 1.148 | 0.72 |
> || Beam Search | 324.28 | 0.72 | 0.3869 | **22.70** | **0.322** | 1.287 | 0.71 |
> || **Flow-TTRL** | 285.97 | **0.83** | **0.4179** | 22.67 | **0.322** | **1.340** | **0.84** |
>
>
> - **Performance Advantage:** The experimental results clearly show that latent searching methods struggle with Compositional (GenEval, Complex) and Structural (OCR) tasks. These methods are inherently limited by the discrete prior distribution. **By utilizing SDE rollouts as context, Flow-TTRL operates as a form of in-context reinforcement learning, navigating toward continuous high-reward regions that latent selection methods often overlook [3].** Regarding the experimental results with $L_2$ distance constraint and differences between Flow-TTRL and the latent searching methods, please refer to Reviewer UEyp Q1.
> - **Reward Hacking in Latent Searching:** Latent searching methods can achieve competitive results in simple alignment metrics like PickScore or HPS v2 through a greedy SDE solver. They often lead to unnatural visual artifacts or a sharp decline in image diversity. Even when augmented with $L_2$ constraints relative to the vanilla ODE trajectory, they still struggle to effectively balance reward maximization with the original generative prior.
> - **Significant Computational Efficiency:** Flow-TTRL operates as an efficient heuristic search that achieves higher generation quality with a better latency-to-performance trade-off than complex beam search. This demonstrates that gradient-guided trajectory steering is far more compute-efficient than managing multiple candidate branches.
>
> **W3:** We appreciate the suggestion and will cite [4] as a significant advancement in our revised manuscript.
>
> **References:**
>
> 1. Tempflow-grpo: When timing matters for grpo in flow models
> 2. Test-Time Scaling of Diffusion Models via Noise Trajectory Search
> 3. In-Context Reinforcement Learning through Bayesian Fusion of Context and Value Prior
> 4. Sample By Step, Optimize By Chunk: Chunk-Level GRPO For Text-to-Image Generation
>
> **Lastly, thanks again for your comments and suggestions, feel free to let us know if you have any further questions or concerns.**

---

> > ### Author Rebuttal · Reviewer_4d2G · 2026-04-03
> >
> > I appreciate the authors' effort answering my questions, and they are all solved. I am glad to maintain my positive score. Good luck!

---

> > > ### Author Response · Authors · 2026-04-03
> > >
> > > We sincerely thank Reviewer 4d2G for the positive acknowledgment and for the time invested in reviewing our work. We are delighted that our responses effectively addressed all your concerns, and we truly appreciate your support and encouragement!

---

### Decision · Program_Chairs · 2026-04-30

**Decision:**

Accept (regular)

**Comment:**

This paper proposes Flow-TTRL, a test-time reinforcement learning framework that aligns flow-matching text-to-image models with human preferences at inference time without any parameter fine-tuning. The method reinterprets intermediate latent representations as an implicit policy within a KL-regularized MDP and introduces a two-stage optimization strategy: Proximal Reward Difference Prediction (PRDP) for structural stabilization under high noise, followed by Group Relative Policy Optimization (GRPO) for fine-grained aesthetic refinement. Experiments on SD 3.5-Medium and Flux.1 Dev demonstrate substantial gains on GenEval, T2I-CompBench, and human preference benchmarks.

All reviewers acknowledged the novelty of the core idea—performing RL-based alignment at inference time rather than through costly model fine-tuning. The main concerns raised during review were: (1) computational overhead at inference time, (2) novelty relative to existing test-time guidance/optimization methods, (3) a gap between the theoretical formulation (rewards on final x₀) and the implementation (single-step projection x̂₀), and (4) the lack of comparisons against test-time optimization baselines. Through an extensive rebuttal, the authors provided detailed wall-clock breakdowns across hardware, inference steps, and group sizes; equal-time-budget comparisons against Best-of-N, Beam Search, DAS, TITAN-Guide, FreeDoM, and DATE; and convergence analysis. The results convincingly demonstrated that Flow-TTRL achieves a superior performance-efficiency trade-off over latent selection and gradient-based guidance methods, particularly on compositional and structural tasks. The authors also committed to clarifying the single-step projection approximation in the manuscript. All four reviewers confirmed their concerns were resolved after the rebuttal and follow-up exchanges.

The consensus in the post-rebuttal discussion is clearly positive: the paper presents a novel and well-executed paradigm for training-free alignment of flow-matching models, with strong empirical support. The inference-time cost, while non-trivial, is justified by the training-free nature, parallelizability, and favorable trade-offs demonstrated against comparable test-time methods.

Therefore, the AC is happy to recommend acceptance. Congratulations! Please well incorporate the reviewers' feedback to prepare for a a solid camera-ready version.